# ActionEQA: Action Interface for Embodied Question Answering

**Tianwei Bao\*[1], Qineng Wang\*[1], Kangrui Wang[1], Mingkai Deng[2], Guangyi Liu[5], Jiayuan Mao[3], Larry Birnbaum[1], Zhiting Hu[4], Eric P. Xing[2,5], Zhaoran Wang[1], Manling Li[1]**

[1]Northwestern University   [2]Carnegie Mellon University   [3]Massachusetts Institute of Technology
[4]University of California, San Diego   [5]Mohamed bin Zayed University of Artificial Intelligence

*{tianweibao, qinengw, kangrui.wang}@u.northwestern.edu, mingkai.deng@cs.cmu.edu, jiayuanm@mit.edu*
*{l-birnbaum, zhaoran.wang, manling.li}@northwestern.edu, {guangyi.liu, eric.xing}@mbzuai.ac.ae, zhh019@ucsd.edu*

\* Equal contribution.

**Reviewed on OpenReview:** *https://openreview.net/forum?id=HY2ruqdMt4*

## Abstract

While Vision-Language Models (VLMs) are increasingly integral to embodied intelligence, a significant action understanding bottleneck persists in translating high-level semantic instructions into precise low-level physical actions. However, current benchmarks for embodied agents primarily focus on high-level perception and planning, failing to capture the depth and nature of this semantic-to-physical gap. To address this, we introduce ACTIONEQA, the first Embodied Question Answering (EQA) benchmark designed to methodically evaluate the ability of VLMs to bridge this critical yet underexplored semantic-physical divide. Grounded in real-world robotics data, ACTIONEQA thoroughly analyzes VLMs' grasp of the action interface using a dual-tier design: (1) a Three-Tiered Action Hierarchy for pinpointing the depth at which VLMs' action reasoning collapses. (2) Bidirectional Reasoning Tasks for testing whether VLMs struggle more to predict action outcomes or infer the actions that led to them. Our key findings reveal: (1) The primary bottleneck in action understanding occurs at the mid-level, arising from the challenge of grounding compositional language in 3D physical geometry. (2) VLMs are more adept at inferring past actions than predicting their future outcomes. (3) Richer visual inputs require greater spatial reasoning from VLMs to map actions to physical geometry. (4) Within the action hierarchy, model failures shift from predominantly perceptual errors at the high level to flawed geometric and physical reasoning at the low level.

## 1 Introduction

Vision-Language Models (VLMs) (Anthropic, 2025; Comanici et al., 2025; OpenAI, 2025c; Bai et al., 2025b; Zhu et al., 2025) have shown remarkable capabilities in visual perception and high-level planning, suggesting strong potential for embodied intelligence (Driess et al., 2023; Zitkovich et al., 2023; Bousmalis et al., 2023; Li et al., 2023). Yet, a critical gap remains: do VLMs truly understand actions and possess the ability to bridge the semantic-to-physical gap between abstract plans and concrete execution? The central challenge lies in translating semantic-level instructions, such as *"Close the fridge"*, into precise, physical-level commands, such as 7-degree-of-freedom (7DoF) action vectors representing desired state changes (Brohan et al., 2022; 2023; O'Neill et al., 2024; Kim et al., 2024; Black et al., 2024). Despite its importance, the interface between language and control remains poorly understood and insufficiently evaluated.

Existing benchmarks for embodied agents primarily focus on perception, task completion rates, or language-guided manipulation (Majumdar et al., 2024; Das et al., 2018; Yu et al., 2019; Zheng et al., 2022; Gong et al., 2023; Li et al., 2024a; Yang et al., 2025c). This approach treats the agent's reasoning as a black box, overlooking the hierarchical nature of action itself and making it impossible to diagnose where, along the

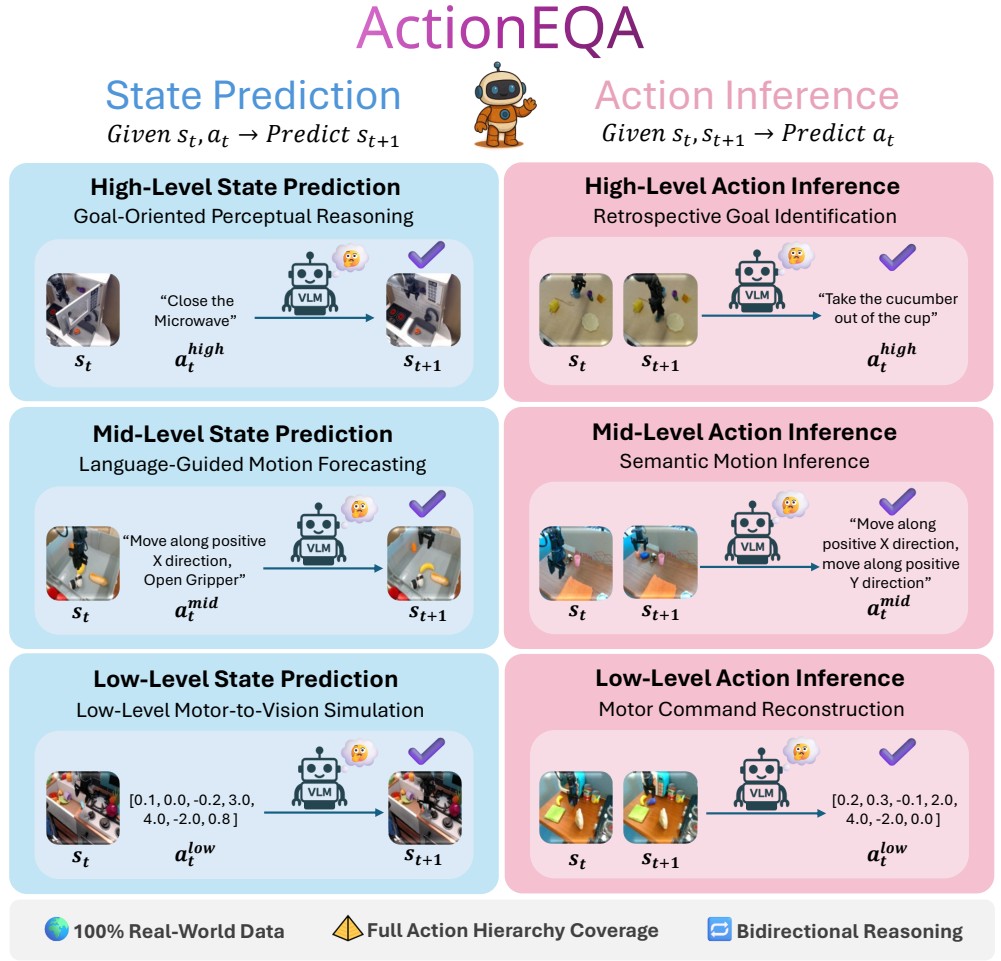

Figure 1: ACTIONEQA assesses the full action hierarchy, from high-level goal-oriented commands to low-level motor commands, through bidirectional action reasoning tasks: State Prediction, which predicts the visual outcome of an action, and Action Inference, which infers the action that caused a visual change.

semantic-to-physical gap, a model's understanding fails. While recent end-to-end Vision-Language-Action (VLA) models (Brohan et al., 2022; Zitkovich et al., 2023; Black et al., 2024; Intelligence et al., 2025) attempt to address this gap directly, their internal reasoning processes are often opaque, making it difficult to interpret failures and know how to generalize from them. Thus, a fine-grained tool is pressingly needed to help us diagnose model deficiencies in reasoning actions at different abstractions.

To tackle this fundamental problem, we introduce ACTIONEQA, the first *action-centric* diagnostic EQA benchmark designed to systematically dissect the semantic-to-physical gap in VLMs. Built entirely on real-world robotics data to ground our analysis in physical complexity, ACTIONEQA serves as a diagnostic tool through two core design principles. First, it evaluates VLMs across a **Three-Tiered Action Hierarchy**, from high-level semantic actions, to mid-level motion trend descriptions, and down to low-level motor commands, to precisely locate where VLMs' action reasoning falters. Second, it employs a **Bidirectional Reasoning Framework** to probe VLMs' grasp of action-based cause and effect by testing both forward state prediction and backward action explanation abilities.

Our comprehensive evaluation of 34 leading VLMs with ACTIONEQA reveals a significant gap between model and human performance (58.4% vs. 95.6%) and provides several critical insights into the semantic-to-physical gap:

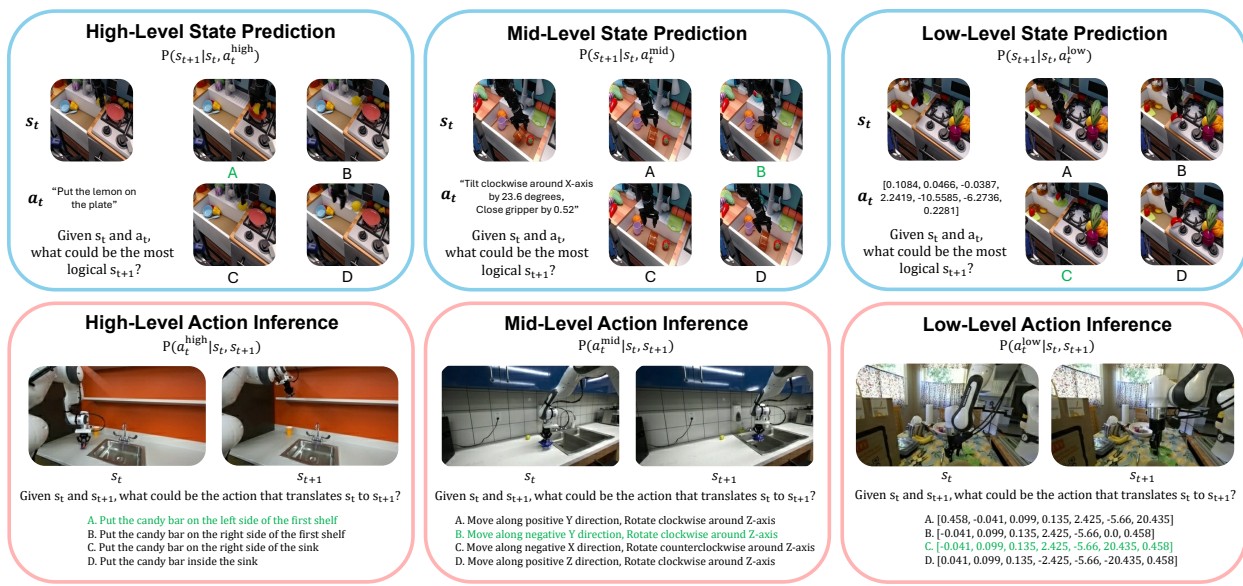

Figure 2: Representative Question-Answering (QA) samples from the six main categories of ACTIONEQA

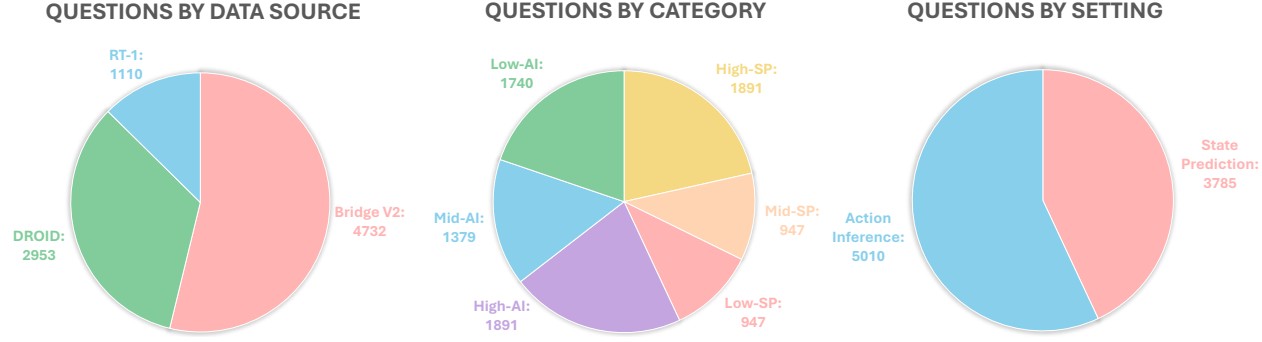

Figure 3: ActionEQA data statistics.

1. **A surprising mid-level bottleneck in the action hierarchy is driven by rotational commands.** Rather than a simple abstract-to-concrete performance decline along the action hierarchy, VLMs struggle most with mid-level semantic motion descriptions, particularly **rotational actions** (e.g., *Tilt clockwise around X-axis, Rotate clockwise around Z-axis*), highlighting the grounding of compositional language into 3D spatial transformations as a key challenge currently faced by VLMs.

2. **VLMs consistently demonstrate a higher aptitude for Action Inference over State Prediction.** They find it easier to explain what action caused an observed outcome than to predict the future outcome of a given action, underscoring a fundamental weakness in predictive physical reasoning.

3. **The Egocentric view dominates in single-view settings.** Camera perspective plays a crucial role in bidirectional action understanding. The egocentric (first-person) viewpoint consistently outperforms the exocentric (third-person) viewpoint, with the performance gap reaching a notable 23.8% on high-level inference tasks.

4. **The nature of model failures shifts with the action hierarchy.** At the high level, errors are predominantly *perceptual*. As actions become more granular, the failure source shifts to flawed *geometric and physical reasoning*, which accounts for the majority of errors at the low level.

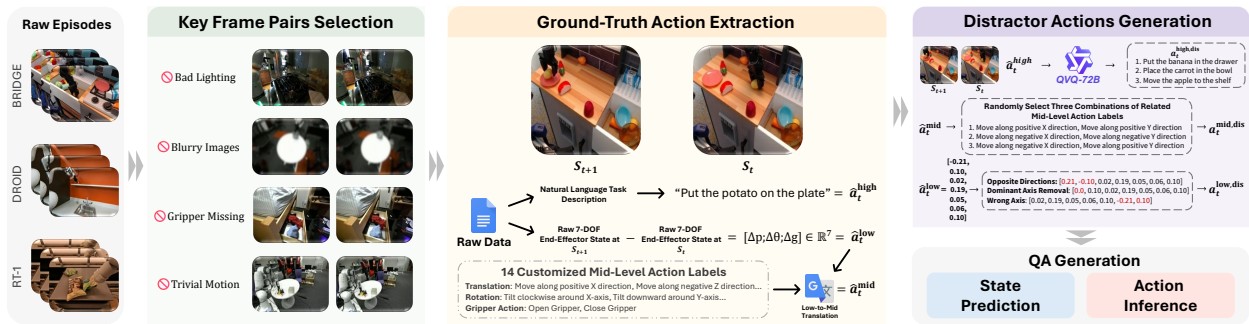

Figure 4: The ActionEQA benchmark curation pipeline.

## 2 ActionEQA Benchmark

### 2.1 A Unified Framework for Evaluating Bidirectional Multi-Level Action Understanding

A pivotal challenge for embodied agents powered by foundation models is bridging the *semantic-to-physical gap*: translating abstract goals (the "what") into the precise motor commands required for physical interaction (the "how"). Existing benchmarks often fail to pinpoint where a model's understanding breaks down across this full spectrum of action. To address this, we introduce ACTIONEQA, the first action-centric EQA benchmark designed to systematically analyze this gap using real-world robotics data. ACTIONEQA serves as a diagnostic tool built on two core principles: a *three-tiered action hierarchy* that localizes failures at specific levels of abstraction, from high-level semantics to low-level physics, and a *bidirectional reasoning framework* that tests action-based cause and effect understanding by requiring both forward predictive and backward inferential reasoning. ACTIONEQA thus provides a comprehensive tool for evaluating how well existing VLMs bridge this critical divide to function as embodied agents.

**Three-Tiered Action Hierarchy.** To analyze a model's capabilities at different levels of abstraction, we decompose actions into a three-tiered hierarchy. The High-Level Action $a_t^{high}$ represents the task's intent or "what," expressed as a natural language goal (e.g., *"Close the Microwave"*). The Mid-Level Action $a_t^{mid}$ describes the "semantic how," providing a language-based description of motion (e.g., *"Move along positive X direction, rotate clockwise around Z-axis"*). Finally, the Low-Level Action $a_t^{low}$ defines the "physical how" through a raw 7-DoF end-effector command vector, $a_t^{low} = [\Delta x, \Delta y, \Delta z, \Delta roll, \Delta pitch, \Delta yaw, \Delta gripper]$, which specifies the precise physical state change.

**Bidirectional Reasoning.** A model's understanding of the action-outcome relationship is incomplete if it cannot reason about this connection from both forward and backward perspectives. We therefore introduce a bidirectional evaluation composed of two complementary reasoning tasks. The first is *State Prediction*, which assesses forward-looking capabilities by asking: "Given an initial state $s_t$ and an action $a_t$, what is the resulting state $s_{t+1}$?" This task directly tests the model's grasp of an action's consequences. The second is *Action Inference*, which evaluates backward-looking, explanatory ability by asking: "Given an initial state $s_t$ and a final state $s_{t+1}$, what action $a_t$ was taken?" Together, these tasks provide a holistic assessment of a model's ability to understand the bidirectional nature of action-based cause and effect.

**EQA Task Formulation.** By applying our bidirectional reasoning framework across the three-tiered action hierarchy, we formulate the six unique action reasoning tasks that comprise the ACTIONEQA benchmark, as illustrated in Figure 1. Together, these tasks provide a comprehensive evaluation of a VLM's ability to reason about the intricate interplay between actions and their physical outcomes. Detailed examples for each task category are presented in Figure 2.

### 2.2 Curation of ActionEQA

ACTIONEQA is built exclusively using data from three large-scale, in-the-wild robotics manipulation datasets: DROID (Khazatsky et al., 2024), BridgeData V2 (Walke et al., 2023), and RT-1 (Brohan et al., 2022). The

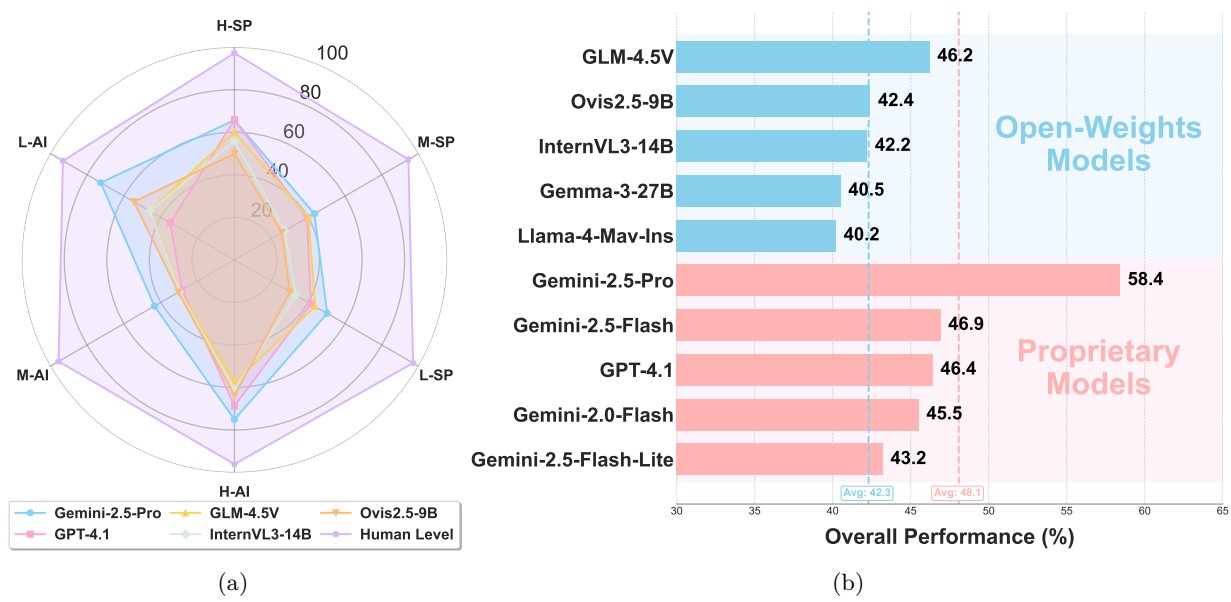

Figure 5: (a) Overall performance of 5 representative models vs. human across six main categories of ActionEQA. (b) Overall performance of the five top-performing open-source and proprietary models.

creation of our benchmark follows a five-stage pipeline, as illustrated in Figure 4. The first stage involves extracting the **Raw Episodes** from DROID, BridgeData V2, and RT-1 datasets. These episodes then go through **Key Frame Pairs Selection** to filter out low-quality data such as episodes with bad lighting, blurry images, a missing gripper, or trivial motion. The third stage is **Ground-Truth Action Extraction**, where the natural language task description, such as "*Put the potato on the plate,*" is extracted from the raw episode data to be used as the high-level ground-truth action $\hat{a}_t^{high}$. In this same stage, aligning with common practice in recent vision-language-action model literature (Brohan et al., 2022; 2023; O'Neill et al., 2024; Belkhale et al., 2024; Kim et al., 2024), the raw 7-DoF end-effector states at $S_{t+1}$ and $S_t$ are used through subtraction to generate the ground-truth low-level action $a_t^{low}$. This is represented by a 7-dimensional vector $[\Delta x, \Delta y, \Delta z, \Delta roll, \Delta pitch, \Delta yaw, \Delta g] = [\Delta \mathbf{p}; \Delta \boldsymbol{\theta}; \Delta g] \in \mathbb{R}^7$ describing desired state changes, which provides a simple, fixed-size regression target suitable for end-to-end learning. Then, we used 14 customized mid-level action labels to translate those low-level actions into their corresponding mid-level actions $\hat{a}_t^{mid}$. The fourth stage is **Distractor Actions Generation**, which involves creating distractor actions for the high, mid, and low action levels based on the ground-truth actions. High-level distractors $a_t^{high,dis}$ are generated using QVQ-72B (Qwen Team, 2024) followed by human verification, mid-level distractors $a_t^{mid,dis}$ are generated by randomly selecting three combinations of related mid-level action labels, and low-level distractors $a_t^{low,dis}$ are generated using techniques like opposite directions, dominant axis removal, and wrong axis. The average Ambiguity Score of 1.15/5.00 and the average Plausibility Score of 4.55/5.00 detailed in Appendix C.4 confirm the exceptional quality of the distractors in ACTIONEQA. The final stage is **QA Generation**, where the ground-truth and distractor actions are used to create multiple-choice questions for State Prediction and Action Inference tasks. A more detailed breakdown of the curation process and dataset statistics is available in the appendix B.

## 3  Experiments

In this section, we begin by conducting an extensive evaluation of the semantic-to-physical gap in VLMs using the ACTIONEQA benchmark in Section 3.2, which reveal three key insights: a surprising bottleneck at the mid-level of the action hierarchy, a pronounced asymmetry in bidirectional reasoning capabilities, and a significant performance gap between both VLMs and humans, as well as between proprietary and open-weights models. Following this, we delve into more detailed analyses that explore the underlying factors

| Type | Model | DROID | | | | | | | BridgeData V2 | | | | | | | RT-1 | | | Overall Perf. |
|---|---|---|---|---|---|---|---|---|---|---|---|---|---|---|---|---|---|---|---|
| | | H-SP | M-SP | L-SP | H-AI | M-AI | L-AI | Avg. | H-SP | M-SP | L-SP | H-AI | M-AI | L-AI | Avg. | H-SP | H-AI | Avg. | |
| Open-Weights | Gemma-3-27B | 45.9 | 34.4 | 29.5 | 51.3 | 26.0 | 32.6 | 36.6 | 63.9 | 31.3 | 35.8 | 71.4 | 24.5 | 40.4 | 44.6 | 54.2 | 61.1 | 57.7 | 40.5 |
| | Gemma-3-12B | 39.1 | 29.3 | 26.1 | 48.4 | 24.1 | 49.2 | 36.0 | 56.0 | 34.0 | 40.6 | 52.3 | 16.2 | 43.2 | 40.4 | 44.7 | 53.7 | 49.2 | 38.2 |
| | Gemma-3-4B | 27.8 | 25.5 | 27.7 | 45.3 | 29.0 | 40.5 | 32.6 | 28.4 | 29.2 | 24.9 | 50.9 | 23.5 | 38.2 | 32.6 | 31.9 | 42.3 | 37.1 | 32.5 |
| | GLM-4.5V 🏆 | 35.4 | 39.3 | 37.5 | 53.0 | 30.3 | 43.1 | 39.8 | 74.1 | 40.9 | 50.2 | 54.5 | 29.2 | 48.8 | 49.6 | 70.8 | 65.2 | 68.0 | 46.2 |
| | GLM-4.1V | 35.1 | 20.6 | 16.3 | 39.1 | 23.7 | 40.7 | 29.3 | 65.4 | 29.0 | 40.4 | 50.7 | 35.3 | 40.7 | 43.6 | 50.3 | 59.3 | 54.8 | 37.2 |
| | GLM-4V | 26.3 | 23.6 | 24.4 | 49.9 | 25.8 | 50.4 | 33.4 | 40.3 | 24.4 | 27.4 | 49.9 | 18.3 | 49.3 | 34.9 | 38.2 | 53.7 | 46.0 | 34.7 |
| | ERNIE-4.5-VL-28B | 43.3 | 30.1 | 24.8 | 41.6 | 30.5 | 31.6 | 33.7 | 51.2 | 33.3 | 32.2 | 44.6 | 31.9 | 27.6 | 36.8 | 55.7 | 48.3 | 52.0 | 36.0 |
| | Llama-4-Scout-17B-16E | 30.9 | 32.8 | 28.1 | 37.4 | 30.5 | 50.4 | 35.0 | 41.3 | 28.8 | 28.8 | 44.5 | 31.3 | 54.2 | 38.2 | 33.5 | 46.5 | 40.0 | 36.7 |
| | Llama-4-Mav-17B-128E | 34.0 | 27.9 | 32.8 | 42.8 | 26.4 | 60.3 | 37.4 | 49.4 | 35.6 | 40.2 | 43.5 | 26.5 | 61.8 | 42.8 | 39.8 | 47.7 | 43.8 | 40.2 |
| | InternVL3-14B 🏆 | 46.5 | 26.7 | 28.5 | 59.5 | 27.2 | 45.0 | 38.9 | 58.5 | 27.9 | 38.1 | 63.7 | 32.5 | 46.8 | 44.6 | 62.9 | 59.3 | 61.1 | 42.2 |
| | InternVL3-8B | 39.1 | 29.7 | 30.5 | 46.5 | 25.4 | 35.0 | 34.4 | 49.6 | 26.0 | 37.0 | 62.6 | 39.0 | 38.4 | 42.1 | 49.0 | 60.9 | 55.0 | 38.8 |
| | InternVL3-2B | 30.3 | 29.5 | 27.5 | 41.1 | 28.2 | 22.3 | 29.8 | 33.1 | 35.6 | 30.6 | 51.4 | 29.4 | 27.8 | 34.7 | 31.0 | 51.7 | 41.4 | 32.5 |
| | InternVL2.5-8B-MPO | 34.8 | 22.8 | 26.5 | 45.3 | 26.0 | 30.7 | 31.0 | 61.6 | 23.1 | 39.3 | 59.7 | 30.8 | 33.2 | 41.3 | 61.1 | 59.3 | 60.2 | 37.2 |
| | InternVL2.5-2B-MPO | 24.9 | 23.6 | 25.1 | 30.6 | 18.6 | 18.6 | 23.6 | 25.4 | 24.4 | 28.5 | 42.4 | 17.6 | 24.6 | 27.2 | 27.2 | 30.8 | 29.0 | 25.2 |
| | InternVL2.5-8B | 30.3 | 27.3 | 28.9 | 47.9 | 29.9 | 37.3 | 33.4 | 55.8 | 24.0 | 37.0 | 58.2 | 33.1 | 43.3 | 41.9 | 54.1 | 59.8 | 57.0 | 38.7 |
| | InternVL2.5-2B | 26.9 | 23.6 | 25.1 | 30.6 | 17.8 | 21.4 | 24.2 | 35.1 | 24.4 | 27.6 | 44.8 | 22.7 | 24.8 | 29.9 | 25.6 | 33.2 | 29.4 | 26.5 |
| | Qwen3-VL-8B-Ins | 40.5 | 29.9 | 32.0 | 56.7 | 28.4 | 33.1 | 36.8 | 41.4 | 32.0 | 29.9 | 58.3 | 33.3 | 32.6 | 37.9 | 47.7 | 78.0 | 62.9 | 38.9 |
| | Qwen2.5-VL-72B-Ins | 38.8 | 32.0 | 27.3 | 41.1 | 28.6 | 46.4 | 35.7 | 52.6 | 31.7 | 33.1 | 49.8 | 21.8 | 46.4 | 39.2 | 41.3 | 76.0 | 58.7 | 38.9 |
| | Qwen2.5-VL-32B-Ins | 32.0 | 27.1 | 31.2 | 33.1 | 33.1 | 45.3 | 33.6 | 50.5 | 30.4 | 37.9 | 42.5 | 21.7 | 45.8 | 38.1 | 50.6 | 71.0 | 60.8 | 38.2 |
| | Qwen2.5-VL-7B-Ins | 35.4 | 25.9 | 23.8 | 36.0 | 29.9 | 45.9 | 32.8 | 46.1 | 26.7 | 27.4 | 39.7 | 34.8 | 45.2 | 36.7 | 51.5 | 72.1 | 61.8 | 37.2 |
| | Qwen2.5-VL-3B-Ins | 23.5 | 23.6 | 25.1 | 37.1 | 27.8 | 36.2 | 28.9 | 24.3 | 24.4 | 23.1 | 36.0 | 32.2 | 36.7 | 29.5 | 25.0 | 63.1 | 44.1 | 30.7 |
| | Qwen2-VL-7B-Ins | 36.0 | 24.2 | 25.0 | 43.6 | 27.2 | 35.5 | 31.9 | 54.3 | 30.1 | 29.0 | 45.1 | 31.0 | 31.0 | 36.8 | 36.9 | 60.7 | 48.8 | 34.8 |
| | Qwen2-VL-2B-Ins | 24.6 | 23.6 | 27.3 | 32.3 | 28.8 | 20.4 | 26.2 | 25.9 | 24.4 | 23.7 | 35.3 | 24.7 | 19.3 | 25.6 | 27.4 | 40.5 | 34.0 | 26.4 |
| | Ovis2.5-9B 🏆 | 42.8 | 29.5 | 29.5 | 61.5 | 28.6 | 55.1 | 41.2 | 49.2 | 21.9 | 30.8 | 63.5 | 32.2 | 54.1 | 42.0 | 57.7 | 66.3 | 62.0 | 42.4 |
| | Ovis2.5-2B | 38.2 | 25.1 | 28.1 | 59.8 | 20.2 | 26.6 | 38.0 | 37.8 | 32.2 | 33.6 | 62.8 | 12.0 | 29.7 | 34.7 | 31.5 | 57.5 | 44.5 | 33.3 |
| | MiniCPM-V-4.5 | 34.8 | 34.6 | 32.0 | 56.9 | 23.9 | 51.6 | 39.0 | 36.5 | 22.4 | 33.3 | 59.9 | 30.7 | 55.4 | 39.7 | 44.1 | 57.7 | 50.9 | 39.9 |
| Proprietary | Gemini-2.5-Pro 🏆 | 52.7 | 44.6 | 48.9 | 70.3 | 42.5 | 73.9 | 55.5 | 75.6 | 42.2 | 51.8 | 77.1 | 44.4 | 71.3 | 60.4 | 68.8 | 77.8 | 73.3 | 58.4 |
| | Gemini-2.5-Flash | 43.3 | 33.2 | 40.3 | 68.6 | 33.5 | 50.9 | 45.0 | 68.9 | 27.9 | 33.6 | 70.8 | 33.1 | 49.5 | 47.3 | 69.2 | 71.0 | 70.1 | 46.9 |
| | Gemini-2.5-Flash-Lite | 45.0 | 24.6 | 31.6 | 58.9 | 30.5 | 56.5 | 41.2 | 54.0 | 25.8 | 39.3 | 64.0 | 24.5 | 58.6 | 44.4 | 57.5 | 61.8 | 59.7 | 43.2 |
| | Gemini-2.0-Flash | 47.6 | 24.8 | 32.6 | 56.7 | 29.0 | 58.0 | 41.5 | 69.6 | 24.7 | 34.7 | 64.3 | 31.8 | 59.7 | 47.5 | 70.6 | 68.1 | 69.4 | 45.5 |
| | GPT-4.1 🏆 | 50.7 | 30.3 | 36.3 | 62.9 | 30.9 | 35.7 | 41.1 | 80.4 | 48.4 | 46.8 | 71.6 | 26.0 | 34.0 | 51.2 | 66.5 | 71.4 | 69.0 | 44.2 |
| | o4-mini | 38.0 | 29.5 | 34.0 | 52.1 | 31.5 | 30.1 | 35.9 | 48.3 | 26.7 | 29.2 | 52.2 | 36.0 | 30.9 | 37.2 | 59.3 | 65.4 | 62.4 | 38.2 |
| | Claude-Opus-4 | 27.8 | 31.0 | 29.5 | 44.2 | 28.6 | 71.9 | 38.8 | 44.0 | 32.9 | 35.8 | 52.2 | 24.8 | 74.2 | 44.0 | 42.3 | 57.1 | 49.7 | 42.3 |
| | Claude-Sonnet-4 | 28.3 | 27.1 | 26.3 | 39.9 | 24.9 | 65.3 | 35.3 | 51.1 | 35.4 | 42.7 | 41.6 | 25.7 | 67.4 | 44.0 | 43.6 | 43.4 | 43.5 | 40.0 |
| | Human Performance | 95.7 | 91.2 | 96.7 | 92.9 | 95.3 | 92.6 | 94.1 | 99.6 | 97.7 | 97.3 | 99.4 | 95.7 | 93.7 | 97.2 | 96.4 | 96.0 | 96.2 | 95.6 |

Table 1: This table shows how accurately models handle bidirectional reasoning of State Prediction (SP) and Action Inference (AI) with High-Level, Mid-Level, and Low-Level Actions across three different datasets (H-, M-, L-). Overall Performance (Overall Perf.) of each model is calculated using the hierarchical averaging method introduced in Section 3. Color highlights identify the top three performing models for each metric (column), ranked separately within the Open-Weight and Proprietary models. Open-Weight (1st, 2nd, 3rd): ■, ■, ■. Proprietary (1st, 2nd, 3rd): ■, ■, ■.

contributing to these gaps. Specifically, we investigate how the richness of perceptual information affects reasoning by varying the number of exocentric camera views in Section 3.3, and we examine the impact of different camera perspectives by comparing egocentric, exocentric, and fused viewpoints in Section 3.4.

## 3.1 Experimental Setup

We evaluated a diverse suite of 34 state-of-the-art VLMs, comprising 26 open-weights models from eight families—Gemma-3 (Team et al., 2025), GLM (Hong et al., 2025; Wang et al., 2024c), ERNIE-4.5-VL (Baidu ERNIE Team, 2025), Llama-4 (Meta, 2025), InternVL (Zhu et al., 2025; Wang et al., 2024d; Chen et al., 2024), QwenVL (Bai et al., 2025b; Yang et al., 2025a), MiniCPM-V (Yu et al., 2025), and Ovis 2.5 (Lu et al., 2025)—and 8 proprietary models from OpenAI (OpenAI, 2025a;c), Google (Comanici et al., 2025), and Anthropic (Anthropic, 2025), with a full list provided in Table 20. To ensure deterministic results, all models were configured with a temperature of 0 and evaluated under the zero-shot setting. We used **accuracy** as our primary metric and calculated overall performance using a **hierarchical averaging method** detailed in Appendix C.2 to ensure each of the six action-centric reasoning tasks contributes equally to the final score. Further details on image resolutions are available in Sections 3.3 and 3.4. For the human evaluation, we recruited two trained annotators to answer ACTIONEQA questions using our custom Gradio-based annotation interface, ensuring that they received exactly the same visual context and instructions as the VLMs. We measured inter-annotator agreement (IAA) using a uniformly stratified subset of 120 questions across six QA settings, independently annotated by both annotators. The resulting raw agreement rate = 95.8% and Cohen's Kappa ($\kappa = 0.944$) indicate near-perfect agreement. Full details are in Appendix C.3.

| QA | Ovis2.5-9B | | | | InternVL3-14B | | | | Qwen2.5-VL-72B | | | | GLM-4.5V | | | | Gemini-2.5-Flash | | | |
|---|---|---|---|---|---|---|---|---|---|---|---|---|---|---|---|---|---|---|---|---|
| | 1V | 2V | 3V | 4V | 1V | 2V | 3V | 4V | 1V | 2V | 3V | 4V | 1V | 2V | 3V | 4V | 1V | 2V | 3V | 4V |
| H-SP | 49.2$_{\pm3.1}$ | 55.2$_{\pm3.1}$ | 52.1$_{\pm3.1}$ | 48.6$_{\pm3.1}$ | 58.5$_{\pm3.1}$ | 62.8$_{\pm3.0}$ | 65.6$_{\pm3.0}$ | 63.0$_{\pm3.0}$ | 52.6$_{\pm3.1}$ | 65.1$_{\pm3.0}$ | 55.6$_{\pm3.1}$ | 44.9$_{\pm3.1}$ | 74.1$_{\pm2.7}$ | 71.8$_{\pm2.8}$ | 71.9$_{\pm2.8}$ | 73.6$_{\pm2.8}$ | 68.9$_{\pm2.9}$ | 68.9$_{\pm2.9}$ | 63.1$_{\pm3.0}$ | 62.9$_{\pm3.0}$ |
| H-AI | 63.5$_{\pm3.0}$ | 58.5$_{\pm3.1}$ | 60.8$_{\pm3.1}$ | 64.5$_{\pm3.0}$ | 63.7$_{\pm3.0}$ | 70.0$_{\pm2.9}$ | 72.9$_{\pm2.8}$ | 75.0$_{\pm2.7}$ | 49.8$_{\pm3.1}$ | 54.7$_{\pm3.1}$ | 63.8$_{\pm3.0}$ | 67.7$_{\pm2.9}$ | 54.5$_{\pm3.1}$ | 56.9$_{\pm3.1}$ | 58.5$_{\pm3.1}$ | 62.4$_{\pm3.0}$ | 70.8$_{\pm2.8}$ | 68.4$_{\pm2.9}$ | 56.4$_{\pm3.1}$ | 56.8$_{\pm3.1}$ |
| M-SP | 21.9$_{\pm3.9}$ | 20.5$_{\pm3.8}$ | 18.7$_{\pm3.7}$ | 19.0$_{\pm3.7}$ | 27.9$_{\pm4.2}$ | 31.7$_{\pm4.4}$ | 32.9$_{\pm4.4}$ | 35.8$_{\pm4.5}$ | 31.7$_{\pm4.4}$ | 39.0$_{\pm4.6}$ | 37.2$_{\pm4.5}$ | 33.3$_{\pm4.4}$ | 40.9$_{\pm4.6}$ | 41.8$_{\pm4.6}$ | 34.7$_{\pm4.5}$ | 39.3$_{\pm4.6}$ | 27.9$_{\pm4.2}$ | 29.0$_{\pm4.3}$ | 29.9$_{\pm4.3}$ | 27.9$_{\pm4.2}$ |
| M-AI | 32.2$_{\pm3.1}$ | 33.0$_{\pm3.1}$ | 34.0$_{\pm3.1}$ | 35.6$_{\pm3.1}$ | 32.5$_{\pm3.1}$ | 31.7$_{\pm3.1}$ | 35.7$_{\pm3.1}$ | 35.4$_{\pm3.1}$ | 21.8$_{\pm2.7}$ | 27.0$_{\pm2.9}$ | 31.7$_{\pm3.1}$ | 34.6$_{\pm3.1}$ | 29.2$_{\pm3.0}$ | 29.7$_{\pm3.0}$ | 28.0$_{\pm3.0}$ | 30.1$_{\pm3.0}$ | 33.1$_{\pm3.1}$ | 33.6$_{\pm3.1}$ | 28.2$_{\pm3.0}$ | 27.6$_{\pm2.9}$ |
| L-SP | 30.8$_{\pm4.3}$ | 35.2$_{\pm4.5}$ | 32.0$_{\pm4.4}$ | 34.0$_{\pm4.4}$ | 38.1$_{\pm4.5}$ | 34.5$_{\pm4.4}$ | 35.2$_{\pm4.5}$ | 43.2$_{\pm4.6}$ | 33.1$_{\pm4.4}$ | 41.1$_{\pm4.6}$ | 41.8$_{\pm4.6}$ | 32.6$_{\pm4.4}$ | 50.2$_{\pm4.7}$ | 53.4$_{\pm4.7}$ | 52.5$_{\pm4.7}$ | 56.6$_{\pm4.6}$ | 33.6$_{\pm4.4}$ | 32.6$_{\pm4.4}$ | 33.6$_{\pm4.4}$ | 32.4$_{\pm4.4}$ |
| L-AI | 54.1$_{\pm3.1}$ | 56.1$_{\pm3.1}$ | 58.5$_{\pm3.1}$ | 60.2$_{\pm3.0}$ | 46.8$_{\pm3.1}$ | 46.6$_{\pm3.1}$ | 47.7$_{\pm3.1}$ | 46.5$_{\pm3.1}$ | 46.4$_{\pm3.1}$ | 52.6$_{\pm3.1}$ | 57.7$_{\pm3.1}$ | 59.0$_{\pm3.0}$ | 48.8$_{\pm3.1}$ | 53.0$_{\pm3.1}$ | 52.7$_{\pm3.1}$ | 51.5$_{\pm3.1}$ | 49.5$_{\pm3.1}$ | 48.5$_{\pm3.1}$ | 48.4$_{\pm3.1}$ | 45.0$_{\pm3.1}$ |
| Avg | 42.0$_{\pm1.4}$ | 43.1$_{\pm1.4}$ | 42.7$_{\pm1.4}$ | 43.7$_{\pm1.4}$ | 44.6$_{\pm1.4}$ | 46.2$_{\pm1.4}$ | 48.3$_{\pm1.4}$ | 49.8$_{\pm1.4}$ | 39.2$_{\pm1.4}$ | 46.6$_{\pm1.4}$ | 48.0$_{\pm1.4}$ | 45.4$_{\pm1.4}$ | 49.6$_{\pm1.4}$ | 51.1$_{\pm1.4}$ | 49.7$_{\pm1.4}$ | 52.3$_{\pm1.4}$ | 47.3$_{\pm1.4}$ | 46.8$_{\pm1.4}$ | 43.3$_{\pm1.4}$ | 42.1$_{\pm1.4}$ |

Table 2: Ablation study on the impact of the number of exocentric views using BridgeData V2. Each model is evaluated with 1, 2, 3, and 4 exocentric views (1V-4V). For each model and task row, the highest score is highlighted in Light gray to indicate the optimal number of views. Values represent accuracy percentages ± 95% Wald Confidence Intervals.

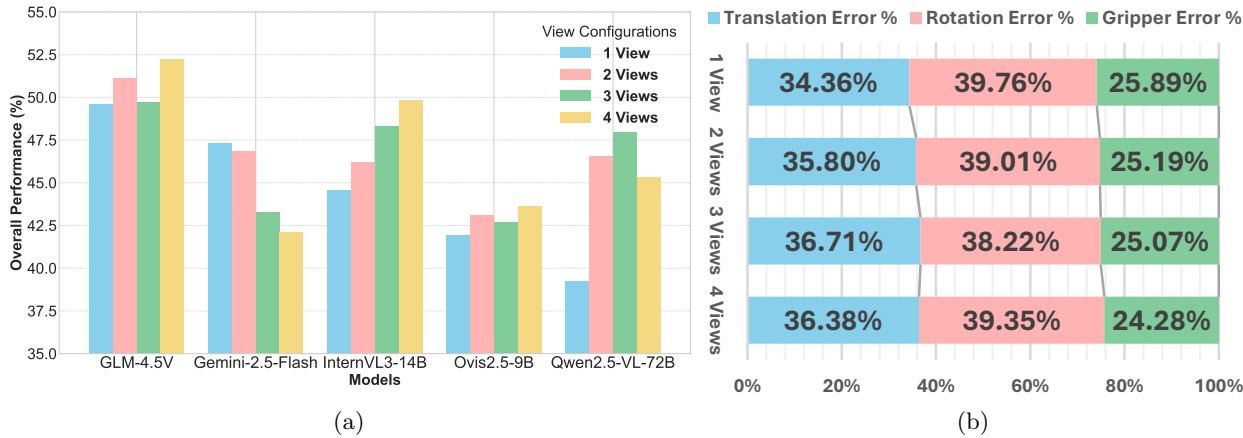

(a)                                                    (b)

Figure 6: (a) Overall performance of five representative VLMs across four exocentric view configurations using BridgeData V2. (b) Breakdown of average normalized error rates by view configuration on Bridge V2.

## 3.2 Semantic-Physical Gap in Action Reasoning

**The Unexpected Mid-Level Bottleneck in the Action Hierarchy.** Contrary to the expectation of a monotonic performance decline from abstract to concrete actions, our analysis pinpoints a surprising and significant bottleneck at the mid-level of the action hierarchy. As evidenced in Table 1, while VLMs demonstrate strong high-level goal-oriented reasoning and regain some footing at low-level kinematic interpretation, their *performance plummets when tasked with understanding mid-level semantic motion descriptions*. We statistically verified this V-shaped performance trend across all 34 evaluated models in Appendix C.11. Paired $t$-tests confirm that mid-level accuracy is significantly lower than both high-level ($p < 0.001$, mean diff. +18.93%) and low-level ($p < 0.001$, mean diff. +8.66%) performance. This trend is further exemplified by the top-performing model Gemini-2.5-Pro, which scores 70.4% on high-level tasks and 61.5% on low-level tasks. However, its performance drops sharply to just 43.4% on mid-level tasks. This indicates that the primary challenge lies not in understanding the ultimate goal (the "what") or the final physical command (the "physical how"), but in the crucial intermediate step of grounding compositional language motion (the "semantic how") such as "*Move along positive X-axis, Rotate clockwise around Z-axis*" into the geometry of physical movement. This grounding of semantic motion appears to be a core, unresolved challenge for current VLMs.

**VLMs Exhibit Asymmetric Aptitude in Bidirectional Action Reasoning.** Our bidirectional framework reveals a striking asymmetry in how VLMs reason about cause and effect of actions. Across the board, *VLMs consistently demonstrate a higher aptitude for Action Inference than for State Prediction*. This capability gap is statistically significant across the 34 evaluated models ($p < 0.001$, paired $t$-test), with Action Inference outperforming State Prediction by an average margin of 4.31% (see Appendix C.12). In other words, VLMs find it easier to interpret a finished action by inferring its preceding

| | Egocentric View | | | | | | Exocentric View | | | | | | Egocentric + Exocentric View | | | | | |
| | State Prediction | | | Action Inference | | | State Prediction | | | Action Inference | | | State Prediction | | | Action Inference | | |
| Model | H | M | L | H | M | L | H | M | L | H | M | L | H | M | L | H | M | L |
| Gemini-2.5-Pro | $60.3_{\pm5.1}$ | $42.4_{\pm4.3}$ | $43.6_{\pm4.3}$ | $85.6_{\pm3.7}$ | $45.4_{\pm4.4}$ | $77.4_{\pm3.0}$ | $52.7_{\pm5.2}$ | $44.6_{\pm4.3}$ | $48.9_{\pm4.4}$ | $70.3_{\pm4.8}$ | $42.5_{\pm4.4}$ | $73.9_{\pm3.2}$ | $60.1_{\pm5.1}$ | $42.8_{\pm4.3}$ | $44.2_{\pm4.3}$ | $85.0_{\pm3.7}$ | $41.7_{\pm4.4}$ | $72.7_{\pm3.2}$ |
| Gemini-2.5-Flash | $63.7_{\pm5.0}$ | $42.6_{\pm4.3}$ | $42.8_{\pm4.3}$ | $78.5_{\pm4.3}$ | $42.7_{\pm4.4}$ | $58.4_{\pm3.6}$ | $43.3_{\pm5.2}$ | $33.2_{\pm4.1}$ | $40.3_{\pm4.3}$ | $68.6_{\pm4.8}$ | $33.5_{\pm4.2}$ | $50.9_{\pm3.6}$ | $58.6_{\pm5.1}$ | $41.5_{\pm4.3}$ | $41.1_{\pm4.3}$ | $83.3_{\pm3.9}$ | $36.4_{\pm4.3}$ | $52.8_{\pm3.6}$ |
| GLM-4.5V | $57.8_{\pm5.2}$ | $36.5_{\pm4.2}$ | $36.7_{\pm4.2}$ | $80.7_{\pm4.1}$ | $41.1_{\pm4.4}$ | $49.6_{\pm3.6}$ | $35.4_{\pm5.0}$ | $39.3_{\pm4.3}$ | $37.5_{\pm4.3}$ | $53.0_{\pm5.2}$ | $30.3_{\pm4.1}$ | $43.1_{\pm3.6}$ | $54.1_{\pm5.2}$ | $39.3_{\pm4.3}$ | $32.6_{\pm4.1}$ | $79.9_{\pm4.2}$ | $35.4_{\pm4.2}$ | $49.1_{\pm3.6}$ |
| Gemma-3-27B | $51.3_{\pm5.2}$ | $35.6_{\pm4.2}$ | $33.0_{\pm4.1}$ | $78.8_{\pm4.3}$ | $32.7_{\pm4.2}$ | $34.2_{\pm3.6}$ | $45.9_{\pm5.2}$ | $34.4_{\pm4.1}$ | $29.5_{\pm4.0}$ | $51.3_{\pm5.2}$ | $26.0_{\pm3.9}$ | $32.6_{\pm3.4}$ | $42.8_{\pm5.2}$ | $35.8_{\pm4.2}$ | $34.6_{\pm4.1}$ | $68.6_{\pm4.8}$ | $29.0_{\pm4.0}$ | $35.1_{\pm3.4}$ |
| InternVL3-14B | $54.7_{\pm5.2}$ | $29.9_{\pm4.0}$ | $28.5_{\pm3.9}$ | $80.2_{\pm4.2}$ | $33.9_{\pm4.2}$ | $42.8_{\pm3.6}$ | $46.5_{\pm5.2}$ | $26.7_{\pm3.9}$ | $28.5_{\pm3.9}$ | $59.5_{\pm5.1}$ | $27.2_{\pm3.9}$ | $45.0_{\pm3.6}$ | $45.6_{\pm5.2}$ | $27.9_{\pm3.9}$ | $26.1_{\pm3.8}$ | $76.5_{\pm4.4}$ | $32.3_{\pm4.1}$ | $48.5_{\pm3.6}$ |
| Qwen2.5-VL-7B | $44.5_{\pm5.2}$ | $28.1_{\pm3.9}$ | $28.5_{\pm3.9}$ | $78.2_{\pm4.3}$ | $27.2_{\pm3.9}$ | $46.9_{\pm3.6}$ | $35.4_{\pm5.0}$ | $25.9_{\pm3.8}$ | $23.8_{\pm3.7}$ | $36.0_{\pm5.0}$ | $29.9_{\pm4.1}$ | $45.9_{\pm3.6}$ | $30.6_{\pm4.8}$ | $25.7_{\pm3.8}$ | $23.6_{\pm3.7}$ | $68.6_{\pm4.8}$ | $27.6_{\pm3.9}$ | $52.3_{\pm3.6}$ |
| Average | $55.4_{\pm5.2}$ | $35.9_{\pm4.2}$ | $35.5_{\pm4.2}$ | $80.3_{\pm4.1}$ | $37.2_{\pm4.3}$ | $51.6_{\pm3.6}$ | $43.2_{\pm5.2}$ | $34.0_{\pm4.1}$ | $34.8_{\pm4.1}$ | $56.5_{\pm5.2}$ | $31.6_{\pm4.1}$ | $48.6_{\pm3.6}$ | $48.6_{\pm5.2}$ | $35.5_{\pm4.2}$ | $33.7_{\pm4.1}$ | $77.0_{\pm4.3}$ | $33.7_{\pm4.2}$ | $51.8_{\pm3.6}$ |

Table 3: Ablation study comparing model performance across Egocentric, Exocentric, and Combined View configurations on the DROID dataset. Values include ± 95% Wald Confidence Intervals. H/M/L denote High/Mid/Low-Level action reasoning tasks. Light gray highlights the optimal view configuration for each model and task.

| | | Viewpoint Performance (%) | | | Performance Comparison | |
| Task | Action Level | Egocentric | Exocentric | Combined | Best Single View | Combined vs. Best |
|---|---|---|---|---|---|---|
| State Prediction | High-Level | 55.4 | 43.2 | 48.6 | Egocentric (+12.2) | -6.8 (Degradation) |
| | Mid-Level | 35.9 | 34.0 | 35.5 | Egocentric (+1.9) | -0.4 (Degradation) |
| | Low-Level | 35.5 | 34.8 | 33.7 | Egocentric (+0.7) | -1.8 (Degradation) |
| Action Inference | High-Level | 80.3 | 56.5 | 77.0 | Egocentric (+23.8) | -3.3 (Degradation) |
| | Mid-Level | 37.2 | 31.6 | 33.7 | Egocentric (+5.6) | -3.5 (Degradation) |
| | Low-Level | 51.6 | 48.6 | 51.8 | Egocentric (+3.0) | +0.2 (Marginal Gain) |

Table 4: Key insights into performance changes for Egocentric, Exocentric, and Combined views from Table 3.

causes than to anticipate the visual consequences of an intended action. This pattern is exemplified by the top-performing model, Gemini-2.5-Pro, which achieves an average score of 63.7% across all Action Inference tasks, yet only scores 53.2% on State Prediction tasks. This substantial gap underscores a fundamental weakness in their ability to perform predictive action reasoning concerning physical dynamics, which currently lags far behind their explanatory and retrospective capabilities.

**Performance Landscape of VLMs on Action Reasoning.** Our evaluation reveals *a profound gap between current VLM capabilities and human performance in action understanding*. As shown in Table 1 and Figure 5a, the top-performing model, Gemini-2.5-Pro, achieves an overall score of 58.4%, which, while leading the pack, falls dramatically short of the 95.6% accuracy demonstrated by human evaluators. This substantial 37.2% gap underscores the immense challenge that nuanced, real-world physical reasoning poses to even the most advanced systems. Within this performance landscape, a clear divide also emerges between proprietary and open-weights models. As illustrated in Figure 5b, the top of the leaderboard is dominated by proprietary models, with Gemini-2.5-Pro (58.4%), Gemini-2.5-Flash (46.9%), and GPT-4.1 (46.4%) setting the benchmark. While the leading open-weights models such as GLM-4.5V (46.2%) deliver competitive results, the overall trend suggests that the architectural and data-scaling advantages of *proprietary VLMs currently provide a distinct edge in bridging the formidable semantic-to-physical gap in multi-level bidirectional action reasoning tasks*.

### 3.3 The Impact of Multiple Views on VLMs

**Richer Views Do Not Guarantee Better Performance.** To investigate how perceptual richness affects the semantic-to-physical gap, we conducted an ablation study varying the number of exocentric camera views from one to four using the BridgeData V2 dataset, which is well-suited for multi-exocentric-view analysis. Our experiment revealed that VLMs respond to additional visual data in highly distinct ways. As shown in Figure 6a, while a model like InternVL-14B demonstrates monotonic improvement with more views, more advanced reasoning models such as Gemini-2.5-Flash show a consistent decline in overall performance. This suggests a counterintuitive finding: providing *richer visual information does not consistently lead to better performance* and may, in some cases, introduce a reconciliation overhead that hinders the action reasoning capabilities of VLMs. To isolate models' reasoning capabilities from potential visual data

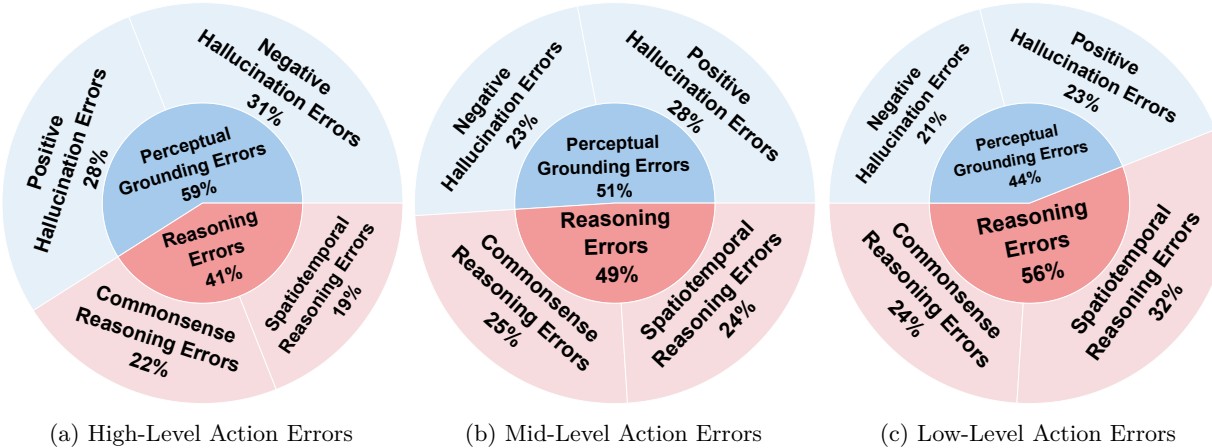

(a) High-Level Action Errors     (b) Mid-Level Action Errors     (c) Low-Level Action Errors

Figure 7: Error categorization of VLMs across high-level, mid-level, and low-level action reasoning tasks. As actions become more granular, failures shift from primarily Perceptual Grounding Errors to Reasoning Errors.

inconsistencies, we conducted an additional controlled ablation using synchronized and rectified DROID stereo pairs in Appendix C.5. This experiment provides further empirical evidence of degraded performance in advanced reasoning models like Gemini-2.5-Pro when exposed to multi-view inputs. A deeper analysis, detailed in Table 2, reveals this variation is strongly tied to the nature of the reasoning task. On one hand, *Action Inference tasks tend to benefit from richer perceptual input.* For instance, models like Qwen2.5-VL-72B and GLM-4.5V show steady gains on these tasks as more views are added. This suggests that for retrospective reasoning, more visual data provides more complete evidence, helping the model to disambiguate a past event. In stark contrast, *State Prediction tasks exhibit a much more fragile relationship with additional views*. This fragility largely explains the complex and sometimes negative performance trends observed in our experiment. For example, the performance of top models like GLM-4.5V and Gemini-2.5-Flash on high-level State Prediction tasks peaks with just a single camera view before declining. This divergence suggests that while more visual data helps clarify a completed action, it can complicate the process of forecasting a future state. We hypothesize that predicting an outcome requires precise geometric and physical reasoning, and that attempting to reconcile and project forward from multiple, slightly inconsistent perspectives imposes a significant cognitive overhead.

**Rotation Poses the Primary Challenge for Mid-Level Actions Understanding.** Building upon our discovery of a performance bottleneck at the mid-level of the action hierarchy, we conducted a more granular analysis to examine how error rates for specific mid-level action types—translation, rotation, and gripper actions—are affected by the number of exocentric views. As depicted in Figure 6b, the average normalized error rates indicate that *rotational actions represent the most formidable challenge for VLMs within the mid-level setting*. Across all tested exocentric viewpoint configurations, rotational actions consistently yielded the highest error percentages, ranging from 38.22% to 39.76%. We confirmed this finding via a paired $t$-test in Appendix C.13, showing that rotational errors are significantly more frequent than translational errors ($p = 0.013$, mean diff. $+3.27\%$) with a very large effect size Cohen's $d = 2.04$. In sharp contrast, gripper actions proved to be the least ambiguous, consistently showing the lowest error rates (decreasing from 25.89% with one view to 24.28% with four views), suggesting that additional perspectives are indeed beneficial for resolving simple binary states. However, both translation and rotation error rates showed no significant and consistent improvement with the addition of more views. Instead, their error rates fluctuated and even slightly increased overall as more views were added. This divergence suggests that while richer visual input helps clarify the discrete states of a gripper, it struggles to resolve the intricate geometric reasoning required for 3D translation and, most notably, rotation. This further emphasizes that *the core difficulty in grounding compositional language descriptions of continuous spatial transformations into robust physical understanding.*

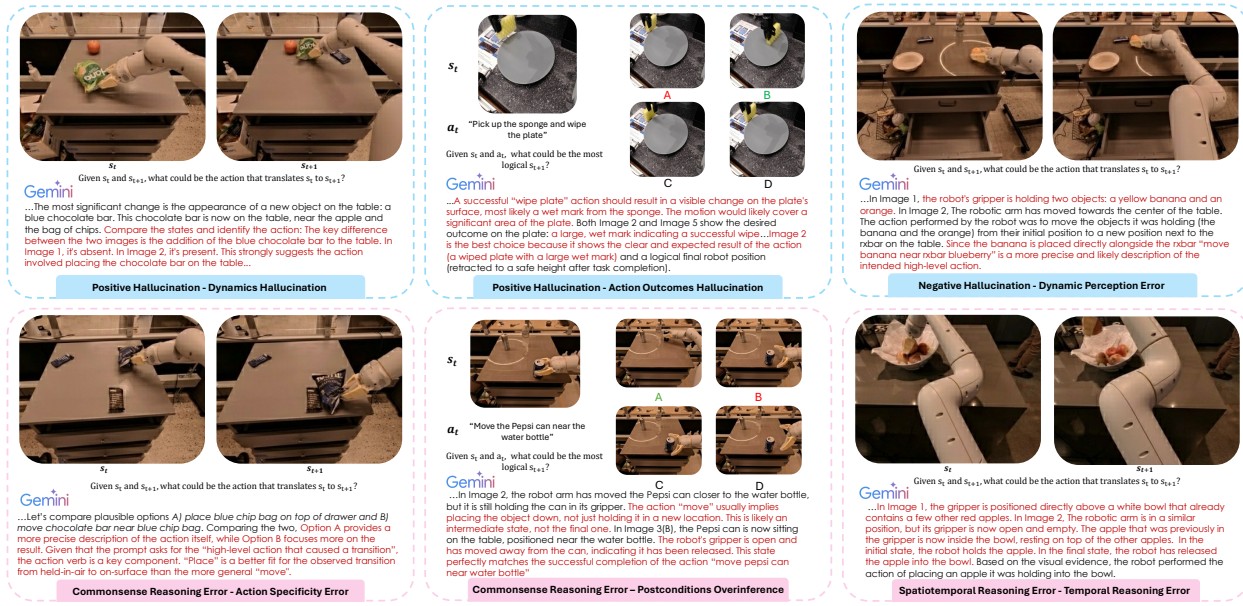

Figure 8: Six representative error examples illustrating Perceptual Grounding Errors and Reasoning Errors.

## 3.4 The Dominance of the Egocentric Perspective

**Egocentric Perspective Dominates in Single-View Scenarios.** To comprehensively understand the impact of camera perspective on action understanding, we conducted an ablation study investigating the efficacy of egocentric versus exocentric viewpoints, and their combination, in bidirectional reasoning tasks. This analysis was performed exclusively on the DROID dataset, as it is the only dataset in our benchmark that provides both camera perspectives. The average performance across representative VLMs under these various view settings is summarized in Table 3, with key insights highlighted in Table 4. Our findings reveal a clear advantage for the egocentric view in single-view scenarios. ***When only one viewpoint is available, the egocentric perspective unequivocally outperforms the exocentric view across all action levels for both State Prediction and Action Inference tasks***. We statistically validated this dominance using a two-proportion Z-test in Appendix C.14, confirming that the egocentric advantage is significant for both State Prediction ($p = 0.014$) and Action Inference ($p < 0.001$). For the State Prediction task, the egocentric view demonstrated a significant advantage, particularly on high-level actions with a **12.2%** margin. The dominance of the egocentric view is even more pronounced in the Action Inference task, where its lead on high-level actions is substantial at **23.8%**. This robust performance across all metrics underscores that the first-person, egocentric perspective provides a more informative and effective viewpoint for understanding cause-and-effect relationships in action reasoning tasks when visual input is limited to a single stream.

**Egocentric and Exocentric Fusion Leads to A Clash of Perspectives.** Contrary to the intuitive expectation that fused visual information would improve understanding, our analysis shows that current VLMs do not generally benefit from a combined egocentric and exocentric view. As highlighted in Table 4, providing ***ostensibly richer visual information through view fusion leads to a consistent degradation in performance in almost all scenarios when compared to using the superior egocentric view alone***. For the State Prediction task, the drop is most severe for high-level actions, with a **6.8%** margin compared to the egocentric-only baseline. A similar trend is observed in the Action Inference task, where performance with the fused view on high-level and mid-level inference declines by **3.3% and 3.5%**, respectively. This pattern suggests that current VLMs struggle to properly reconcile and integrate information from distinct first-person and third-person viewpoints. Rather than creating synergy, the combined input appears to introduce conflicting noise that prevents models from effectively bridging the semantic-to-physical gap. This reinforces the notion that effective embodied intelligence requires not just more data, but a principled approach to integrating diverse sensory inputs to avoid overwhelming the model's reasoning capacity.

# 4    Qualitative Error Analysis

Our quantitative findings indicate a reasoning bottleneck. To further validate this observation, we conducted a qualitative error analysis of the top-performing model, Gemini-2.5-Pro. Specifically, we randomly sampled 50 error cases for each action level within both the State Prediction and Action Inference tasks. Our qualitative analysis reveals an empirical shift in the nature of failures across the action hierarchy, from predominantly perceptual errors at the high level to geometric and physical reasoning errors at the low level. The distribution of these errors across the three action levels is illustrated in Figure 7, and a detailed error taxonomy with representative examples is provided in Table 33.

**General Error Trends.** A qualitative analysis of 150 sampled error cases reveals a clear and systematic shift in the challenges faced by VLMs across the three tiers of the action hierarchy. Across all levels, errors can be broadly categorized into two main types: Perceptual Grounding Errors and Reasoning Errors. At the high-level setting, perceptual errors are clearly dominant over reasoning errors (59.0% vs. 41.0%), meaning VLMs usually understand the logic of the goal but misidentify the objects involved. This balance gradually inverts as we move towards more granular action representations. At the low-level, reasoning errors become the majority of failures (56.0% vs. 44.0%), which indicates that the primary bottleneck at the low level is not a lack of visual information, but rather a deficiency in how models think about 3D spatial transformations and coordinate systems.

**Perceptual Grounding Errors.** This category encompasses failures where the model misinterprets the visual information present in the scene. These errors are not about faulty reasoning but about an incorrect "seeing" of the world, and they manifest in several ways. The first row of Figure 8 includes some of the most classical examples of Perceptual Grounding Errors we discovered through analysis of the model's internal reasoning process. For instance, the model might suffer from *Dynamics Hallucination*, where it fails to understand principles like object occlusion; when a previously hidden chocolate bar becomes visible, the model hallucinates a "placing" action rather than recognizing it was simply revealed. Similarly, *Action Outcomes Hallucination* occurs when the model invents a visual outcome that isn't there, such as misinterpreting a shadow as a "wet mark" to satisfy the command to "wipe the plate." Furthermore, *Dynamic Perception Errors* can corrupt the model's basic object identification during physical interactions. In a notable case, a robot holding an orange was perceived as holding both an orange and a banana, simply because the gripper's color resembled a banana, leading to a completely flawed understanding of the scene.

**Reasoning Errors.** This class of errors arises not from misperception, but from flawed logic, incorrect inferences, or a poor understanding of physical or temporal dynamics. As demonstrated in the second row of Figure 8, these failures can sometimes be subtle. For example, a model might commit an *Action Specificity Error*, a type of commonsense failure where it correctly identifies the objects involved but describes the action with a label that is factually true but unhelpfully imprecise. Another common commonsense failure is *Postconditions Overinference*, where the model makes rigid assumptions about the outcome of an action, such as insisting that a "move" command must conclude with the object being released, even if the visual evidence only shows it being held in a new location. Finally, *Temporal Reasoning Errors* reveal a fundamental confusion about the sequence of events. In these cases, the model may correctly identify the change in the scene but incorrectly infer the direction of time, leading it to mistake the initial state for the final one and consequently deduce the exact opposite of the action that occurred, such as believing an apple was placed into a bowl when it was actually picked up.

# 5    Related Work

**Benchmarks for Embodied Agents.** Prior benchmarks have been vital for advancing embodied AI. One line of work, Embodied Question Answering (EQA) (Das et al., 2018; Gordon et al., 2018; Anderson et al., 2018; Yu et al., 2019; Ku et al., 2020; Ma et al., 2022; Tan et al., 2023; Majumdar et al., 2024; Zhao et al., 2025b; Li et al., 2025; Jiang et al., 2025), focuses on navigation and visual perception. Another prominent category centers on language-guided manipulation (Zheng et al., 2022; Mees et al., 2022; Liu et al., 2023; Zhao et al., 2025a; Yang et al., 2025c; Yin et al., 2025) evaluating high-level reasoning and robotic skills. While essential, these benchmarks are primarily designed to measure task completion, often treating

| Benchmark | SP | AI | HL-QA | ML-QA | LL-QA | Multi-Views | Real Scenes |
|---|---|---|---|---|---|---|---|
| EQA (Das et al., 2018) | ✓ | ✗ | ✓ | ✗ | ✗ | ✗ | ✗ |
| MT-EQA (Yu et al., 2019) | ✓ | ✗ | ✓ | ✗ | ✗ | ✗ | ✗ |
| VLMbench (Zheng et al., 2022) | ✓ | ✗ | ✓ | ✗ | ✓ | ✓ | ✗ |
| ARNOLD (Gong et al., 2023) | ✓ | ✗ | ✓ | ✗ | ✗ | ✓ | ✗ |
| OpenEQA (Majumdar et al., 2024) | ✓ | ✗ | ✓ | ✗ | ✗ | ✗ | ✓ |
| VisualAgentBench (Liu et al., 2024) | ✓ | ✗ | ✓ | ✗ | ✗ | ✓ | ✗ |
| Embodied Agent Interface (Li et al., 2024a) | ✓ | ✗ | ✓ | ✗ | ✗ | ✗ | ✗ |
| EmbodiedBench (Yang et al., 2025c) | ✓ | ✗ | ✓ | ✗ | ✓ | ✓ | ✗ |
| ManipBench (Zhao et al., 2025a) | ✓ | ✓ | ✓ | ✗ | ✓ | ✗ | ✓ |
| WorldPredictionBench (Chen et al., 2025a) | ✓ | ✗ | ✓ | ✗ | ✗ | ✗ | ✓ |
| WM-ABench (Gao et al., 2025) | ✓ | ✗ | ✓ | ✓ | ✗ | ✓ | ✗ |
| **ActionEQA** | ✓ | ✓ | ✓ | ✓ | ✓ | ✓ | ✓ |

Table 5: Comparison of ACTIONEQA with prior work on evaluating embodied agents. SP denotes State Prediction; AI denotes Action Inference; HL, ML, and LL denote high-, mid-, and low-level actions.

the agent's reasoning process as a black box. They can tell us *if* an agent succeeded, but not provide a fine-grained diagnosis of *why* or *where* it failed in the critical translation from a semantic goal to a physical action. This lack of granularity makes it difficult to dissect an agent's understanding across the action hierarchy. ACTIONEQA is designed to fill this diagnostic void, providing a systematic framework to dissect the semantic-to-physical gap at every level of the action hierarchy.

**VLMs as Embodied Agents.** VLMs now serve as the central reasoning core for embodied agents (Jiang et al., 2022; Driess et al., 2023; Stone et al., 2023; O'Neill et al., 2024; Chen et al., 2025b), yet bridging the semantic-to-physical gap remains a primary challenge. The field has seen a rapid paradigm shift, moving from earlier approaches that decomposed goals into interpretable mid-level skills (Huang et al., 2022a; Ahn et al., 2022; Huang et al., 2022c) to modern end-to-end models (Brohan et al., 2022; 2023; Belkhale et al., 2024; Kim et al., 2024; Black et al., 2024; Schakkal et al., 2025) that directly generate low-level motor commands. While this end-to-end approach has increased agent capabilities, it has come at the cost of interpretability, rendering their internal reasoning increasingly opaque. This growing opacity creates a critical need for diagnostic tools.

# 6 Conclusion

We introduce ACTIONEQA, a new action-centric diagnostic benchmark grounded in real-world robotics data to evaluate the semantic-to-physical gap in VLMs. By leveraging a three-tiered action hierarchy and a bidirectional reasoning framework, ACTIONEQA comprehensively evaluates VLMs' ability to translate abstract goals into precise physical actions and pinpoints their failures in understanding action-based cause and effect. Our analysis of 34 VLMs reveals that the primary challenge is not a simple decline from abstract to concrete reasoning, but a distinct bottleneck at the mid-level of the action hierarchy to ground compositional language into 3D spatial transformations. We also find that VLMs are much better at inferring past actions but struggle to predict future outcomes, exposing a core weakness in their predictive physical reasoning. Furthermore, we show the bottleneck is often *reasoning*, not just perception. Richer visual input can impose a reasoning burden and degrade performance, with egocentric views proving dominant in single-view scenarios. These findings challenge the notion that perception is the main obstacle for embodied agents. Instead, they demonstrate a core deficit in spatial, physical, and predictive reasoning. We posit ACTIONEQA as a valuable tool to guide the development of next-generation VLMs that can bridge the gap from semantic understanding to physical execution in embodied agents.

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

# Appendix

## Table of Contents

## A    Extended Related Work

The pursuit of general-purpose robots has been significantly advanced by the integration of Vision-Language Models (VLMs) (Anthropic, 2025; OpenAI, 2025c; Comanici et al., 2025; Yang et al., 2025a; Zhu et al., 2025) as the reasoning core for embodied agents (Jiang et al., 2022; Huang et al., 2023; Li et al., 2024b; Huang et al., 2024). Research in this area has evolved from models that generate high-level textual plans for separate execution, such as PaLM-E (Driess et al., 2023), to end-to-end Vision-Language-Action (VLA) models (Zhen et al., 2024; Black et al., 2024; Team et al., 2024; Belkhale et al., 2024; Intelligence et al., 2025), which directly map multimodal inputs to low-level motor commands. More recent work has explored hierarchical policies like RT-H (Belkhale et al., 2024), which introduces language-based intermediate action representations, and generalist policies (Team et al., 2024; Black et al., 2024), which leverage massive datasets like Open X-Embodiment (O'Neill et al., 2024) to create adaptable, foundational models. While these models demonstrate remarkable capabilities in perception and planning, a critical challenge remains in their ability to precisely ground semantic instructions in physical actions. Our work, ACTIONEQA, directly addresses this by providing a framework to systematically analyze this "semantic-to-physical gap", moving beyond simple task success to diagnose *how* and *why* these powerful agents succeed or fail at the action interface.

Existing benchmarks have been instrumental in driving this progress, but they often evaluate agents from a "black box" perspective, focusing on high-level outcomes. Benchmarks like EQA (Das et al., 2018; Majumdar et al., 2024) focus on navigation and question-answering, while others like CALVIN (Mees et al., 2022), VLMBench (Zheng et al., 2022), and ARNOLD (Gong et al., 2023) evaluate an agent's ability to complete long-horizon, compositional, or language-guided manipulation tasks. While some, like ManipBench (Zhao et al., 2025a) and EmbodiedBench (Yang et al., 2025c), probe low-level physical reasoning and comprehensive capabilities, they do not offer a systematic way to pinpoint failures within the action-generation process. ACTIONEQA introduces a novel, diagnostic approach. By structuring evaluation across a **Three-Tiered Action Hierarchy** (high, mid, low) and employing **Bidirectional Reasoning Tasks**, our benchmark is uniquely designed to dissect the agent's understanding at different levels of abstraction, identifying the specific "cliff" where semantic understanding fails to translate into correct physical grounding.

The translation of intent to motion in embodied agents is inherently hierarchical. At the highest level, LLMs are increasingly used for **Task and Motion Planning (TAMP)**, decomposing complex commands into symbolic sub-goals (Kaelbling & Lozano-Pérez, 2011; Ahn et al., 2022; Huang et al., 2022a;b; Wang et al., 2024b). At the lowest level, imitation learning techniques like Behavior Cloning (Chen et al., 2021; Reed et al., 2022; Baker et al., 2022; Chi et al., 2023) and, more recently, Diffusion Policies (Chi et al., 2023; Ze et al., 2024) have proven highly effective at generating precise, continuous motor commands from observations. The crucial and most challenging link is the intermediate level, where abstract plans are grounded into semantic skills—a key bottleneck our work identifies. The performance degradation at the mid-level in ACTIONEQA highlights the importance of hierarchical architectures like RT-H that explicitly model this intermediate layer. By focusing on this critical interface, our benchmark provides a targeted tool for evaluating and fostering the development of models that can truly bridge the gap from high-level reasoning to precise, real-world action.

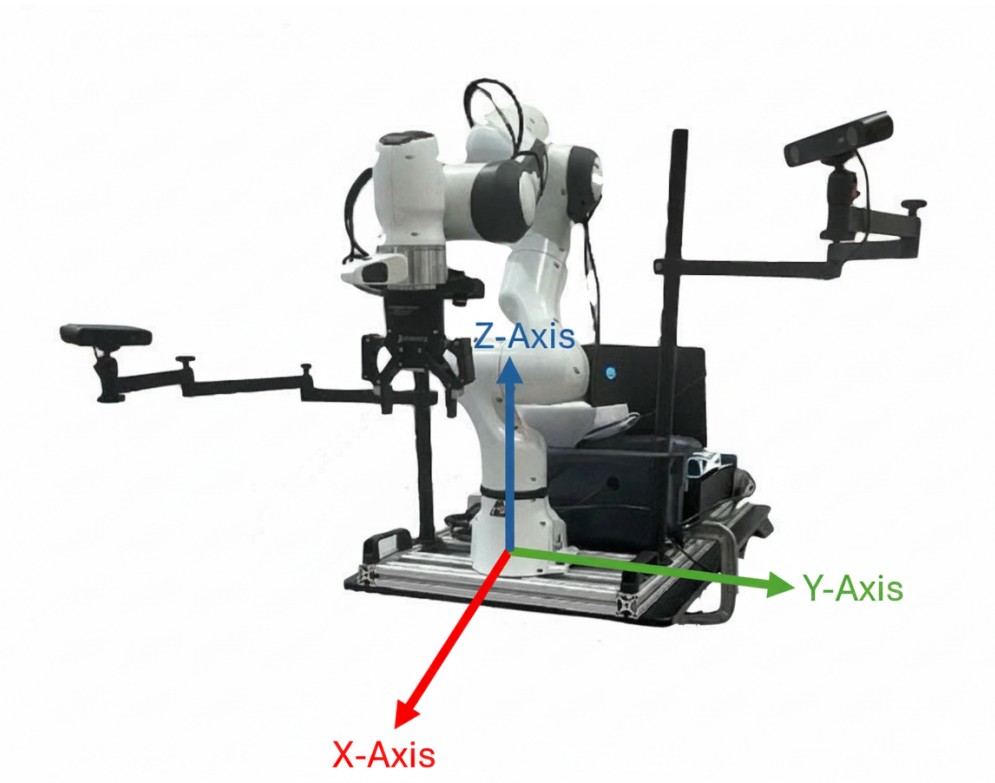

Figure 9: The ActionEQA benchmark adopts a standard coordinate system centered at the robot's base frame. The positive **X-axis** (red arrow) points State Prediction, the positive **Y-axis** (green arrow) points left, and the positive **Z-axis** (blue arrow) points upward.

# B   ActionEQA Benchmark Curation Details

This section provides a comprehensive breakdown of the 5-stage data curation pipeline for the ACTIONEQA benchmark, as illustrated in Figure 4 in the main paper. Our methodology is designed to systematically process large-scale, in-the-wild robotics data into high-quality, diagnostically valuable question-answering pairs. The final benchmark consists of 8,795 questions derived from three foundational datasets: DROID, BridgeData V2, and RT-1.

## B.1   Datasets Selection

Our benchmark, ACTIONEQA, is constructed entirely from large-scale, real-world robotics datasets to ensure that our analysis is grounded in the complexities of real-world physical interactions. We specifically selected three prominent datasets—DROID, BridgeData V2, and RT-1—for their scale, diversity, and detailed action-state logging, which are essential for our fine-grained, multi-level evaluation framework. Below, we detail the characteristics of each dataset.

### B.1.1   BridgeData V2

BridgeData V2 (Walke et al., 2023) is a large-scale dataset designed to facilitate research in generalizable robotic manipulation skills. It comprises approximately 60,000 trajectories collected across 24 distinct environments, featuring a wide array of tasks such as pick-and-place, pushing, folding, and sweeping. Each trajectory is accompanied by a natural language instruction, which we utilize for the high-level action definitions in our benchmark.

The demonstrations were collected on a 6-DoF WidowX 250 robot arm. The dataset provides detailed 7-dimensional state vectors for the end-effector at each timestep, which includes its 6D Cartesian pose (x, y, z, roll, pitch, yaw) and a value for the gripper state. For the purpose of ACTIONEQA, we derive the low-level action vector by calculating the difference between the robot's end-effector state at consecutive timesteps ($s_{t+1}$ and $s_t$). This method of calculating relative changes in pose and gripper status aligns seamlessly with the low-level action formulation used in ACTIONEQA. The dataset's richness in environmental and task diversity provides a solid foundation for testing a model's ability to generalize across different scenarios. Furthermore, BridgeData V2 offers multiple camera views, including a primary over-the-shoulder view and other randomized RGB camera perspectives, which supports our multi-view analysis experiments in Section 3.3.

```
The Raw Data Schema of the BridgeData V2 Dataset

BridgeData V2 = {
    "episode_metadata": {
        "episode_id": <tf.Tensor: shape=(), dtype=int32, numpy=38>, # Unique episode identifier
        "file_path": <tf.Tensor: shape=(), dtype=string>, # The file path where the original episode data is stored.
        "has_language": <tf.Tensor: shape=(), dtype=bool, numpy=True>, # Availability of language instruction
        "has_image_0": <tf.Tensor: shape=(), dtype=bool, numpy=True>, # Availability of the first camera view.
        "has_image_1": <tf.Tensor: shape=(), dtype=bool, numpy=True>, # Availability of the second camera view.
        "has_image_2": <tf.Tensor: shape=(), dtype=bool, numpy=True>, # Availability of the third camera view.
        "has_image_3": <tf.Tensor: shape=(), dtype=bool, numpy=True> # Availability of the fourth camera view.
    },
    "steps": <_VariantDataset element_spec={
        "is_first": TensorSpec(shape=(), dtype=tf.bool, name=None), # If this is the first step of the episode.
        "is_last": TensorSpec(shape=(), dtype=tf.bool, name=None), # If this is the last step of the episode.
        "is_terminal": TensorSpec(shape=(), dtype=tf.bool, name=None), # If the episode terminated at this step.
        "discount": TensorSpec(shape=(), dtype=tf.float32, name=None), # Discount factor for this step
        "reward": TensorSpec(shape=(), dtype=tf.float32, name=None), # Reward associated with this step
        "action": TensorSpec(shape=(7,), dtype=tf.float32, name=None), # [Δx, Δy, Δz, Δroll, Δpitch, Δyaw, Δgripper]
        "observation": {
            "state": TensorSpec(shape=(7,), dtype=tf.float32, name=None), # [x, y, z, roll, pitch, yaw, gripper]
            "image_0": TensorSpec(shape=(256, 256, 3), dtype=tf.uint8, name=None), # Image from the first camera view.
            "image_1": TensorSpec(shape=(256, 256, 3), dtype=tf.uint8, name=None), # Image from the second camera view.
            "image_2": TensorSpec(shape=(256, 256, 3), dtype=tf.uint8, name=None), # Image from the third camera view.
            "image_3": TensorSpec(shape=(256, 256, 3), dtype=tf.uint8, name=None)  # Image from the fourth camera view.
        },
        "language_instruction": TensorSpec(shape=(), dtype=tf.string, name=None), # Natural language instruction
        "language_embedding": TensorSpec(shape=(512,), dtype=tf.float32, name=None) # Natural language embedding.
    }>
}
```

Figure 10: The data structure of a raw episode in the BridgeData V2 dataset

### B.1.2 DROID

The **D**istributed **R**obot **I**nteraction **D**ataset (DROID) (Khazatsky et al., 2024) is an another large-scale, "in-the-wild" manipulation dataset known for its exceptional diversity in scenes, tasks, and lighting conditions. It contains over 76,000 successful demonstration trajectories, amounting to 350 hours of interaction data, collected across 564 unique scenes by 50 different data collectors. This extensive variation makes DROID an ideal source for evaluating the robustness and generalization capabilities of VLMs.

The data was collected using a standardized hardware setup consisting of a 7-DoF Franka Panda robotic arm. The dataset provides detailed 7-dimensional state vectors for the end-effector in the "observation" dictionary, recording its 6D Cartesian pose and gripper position at each timestep. To form the basis of our low-level action hierarchy, we calculate the $a_{\text{low}}$ vector by subtracting the end-effector state at timestep $t$ from the state at timestep $t + 1$. This resulting 7-dimensional vector represents the end-effector's motion and gripper state change:

$$a_{\text{low}} = [\Delta x, \Delta y, \Delta z, \Delta\text{roll}, \Delta\text{pitch}, \Delta\text{yaw}, \Delta g]$$

where $(\Delta x, \Delta y, \Delta z)$ denotes the translational displacement, $(\Delta\text{roll}, \Delta\text{pitch}, \Delta\text{yaw})$ represents the rotational change, and $\Delta g$ specifies the change in gripper state.

A key feature of the DROID dataset for ACTIONEQA is its camera configuration. Each data collection setup includes two exocentric (third-person) stereo cameras and one egocentric (first-person) wrist-mounted camera. This provides the multi-perspective visual data necessary for our analysis of how different camera viewpoints affect a VLM's action understanding, as detailed in Section 3.4 of the main paper.

```
The Raw Data Schema of the DROID Dataset

DROID = {
"episode_metadata": {
            "recording_folderpath": tf.Text, # path to the folder of recordings
            "file_path": tf.Text, # path to the original data file
            },
     "steps": {
          ...
          "language_instruction": tf.Text, # language instruction
          "language_instruction_2": tf.Text, # alternative language instruction
          "language_instruction_3": tf.Text, # alternative language instruction
          "observation": {
                    "gripper_position": tf.Tensor(1, dtype=float64), # gripper position state
                    "cartesian_position": tf.Tensor(6, dtype=float64), # robot Cartesian state
                    "joint_position": tf.Tensor(7, dtype=float64), # joint position state
                    "wrist_image_left": tf.Image(180, 320, 3, dtype=uint8), # wrist camera RGB left viewpoint
                    "exterior_image_1_left": tf.Image(180, 320, 3, dtype=uint8), # exterior camera 1 left viewpoint
                    "exterior_image_2_left": tf.Image(180, 320, 3, dtype=uint8), # exterior camera 2 left viewpoint
          },
          "action_dict": {
                    "gripper_position": tf.Tensor(1, dtype=float64), # commanded gripper position
                    "gripper_velocity": tf.Tensor(1, dtype=float64), # commanded gripper velocity
                    "cartesian_position": tf.Tensor(6, dtype=float64), # commanded Cartesian position
                    "cartesian_velocity": tf.Tensor(6, dtype=float64), # commanded Cartesian velocity
                    "joint_position": tf.Tensor(7, dtype=float64),  # commanded joint position
                "joint_velocity": tf.Tensor(7, dtype=float64), # commanded joint velocity
          },
          ...
    },
    }
```

Figure 11: The data structure of a raw episode in the DROID dataset

### B.1.3 RT-1

The **R**obotics **T**ransformer **1** (RT-1) dataset (Brohan et al., 2022) was collected to train a large-scale, multi-task transformer model for robotics. It contains over 130,000 episodes spanning more than 700 distinct task instructions, collected by a fleet of 13 Everyday Robots mobile manipulators over 17 months. The sheer scale and task diversity of the RT-1 dataset make it an invaluable resource for assessing the limits of current VLMs.

The robots used for collection are mobile manipulators, each equipped with a 7-DoF arm and a mobile base. Consequently, the action space in RT-1 is a combination of 7 dimensions for arm movement and 3 dimensions

for base movement (x, y, yaw). ActionEQA's core objective is to dissect the semantic-to-physical gap in *arm manipulation*. The concurrent movement of the mobile base introduces confounding variables that make it difficult to isolate and analyze the arm's action at a granular level. For instance, a change in the visual scene could be due to the arm, the base, or both, creating ambiguity for our mid-level and low-level reasoning tasks. To prevent this confusion and maximize the quality and focus of the ActionEQA benchmark, we exclusively utilize the RT-1 data to construct our high-level State Prediction and Action Inference tasks. These high-level tasks, defined by natural language goals, are less sensitive to the specific motor commands, allowing us to leverage the dataset's vast task diversity without compromising the integrity of our fine-grained analysis. The data was collected in real-world office kitchen environments, adding another layer of realistic complexity to the high-level evaluation.

---

**The Raw Data Schema of the RT-1 Dataset**

```
RT-1 = {
    "aspects": {
        "has_aspects": tf.Tensor(False, dtype=bool),
        "success": tf.Tensor(False, dtype=bool),
        "already_success": tf.Tensor(False, dtype=bool),
        "feasible": tf.Tensor(False, dtype=bool),
        "undesirable": tf.Tensor(False, dtype=bool),
    },

    "attributes": {
        "collection_mode": tf.Tensor(0, dtype=int64),
        "collection_mode_name": tf.Tensor(b"UNSPECIFIED_COLLECTION_MODE", dtype=string),
        "data_type": tf.Tensor(0, dtype=int64),
        "data_type_name": tf.Tensor(b"UNSPECIFIED_DATA_TYPE", dtype=string),
        "env": tf.Tensor(0, dtype=int64),
        "env_name": tf.Tensor(b"UNSPECIFIED_ENV_TYPE", dtype=string),
        "location": tf.Tensor(0, dtype=int64),
        "location_name": tf.Tensor(b"UNSPECIFIED_LOCATION", dtype=string),
        "objects_family": tf.Tensor(0, dtype=int64),
        "objects_family_name": tf.Tensor(b"UNSPECIFIED_OBJECTS_FAMILY", dtype=string),
        "task_family": tf.Tensor(0, dtype=int64),
        "task_family_name": tf.Tensor(b"UNSPECIFIED_TASK_FAMILY", dtype=string),
    },

    "steps": tf.data.DatasetSpec({
        "action": {
            "base_displacement_vector":  tf.TensorSpec((2,), tf.float32),
            "base_displacement_vertical_rotation": tf.TensorSpec((1,), tf.float32),
            "rotation_delta":            tf.TensorSpec((3,), tf.float32),
            "world_vector":              tf.TensorSpec((3,), tf.float32),
            "gripper_closedness_action": tf.TensorSpec((1,), tf.float32),
            "terminate_episode":         tf.TensorSpec((3,), tf.int32),
        },

        "observation": {
            "image":                           tf.TensorSpec((256, 320, 3), tf.uint8),
            "natural_language_instruction":    tf.TensorSpec((), tf.string),
            "natural_language_embedding":      tf.TensorSpec((512,), tf.float32),
            "base_pose_tool_reached":          tf.TensorSpec((7,), tf.float32),
            "vector_to_go":                    tf.TensorSpec((3,), tf.float32),
            "rotation_delta_to_go":            tf.TensorSpec((3,), tf.float32),
            "src_rotation":                    tf.TensorSpec((4,), tf.float32),
            "orientation_start":               tf.TensorSpec((4,), tf.float32),
            "orientation_box":                 tf.TensorSpec((2, 3), tf.float32),
            "robot_orientation_positions_box": tf.TensorSpec((3, 3), tf.float32),
            "workspace_bounds":                tf.TensorSpec((3, 3), tf.float32),
            "height_to_bottom":                tf.TensorSpec((1,), tf.float32),
            "gripper_closed":                  tf.TensorSpec((1,), tf.float32),
            "gripper_closedness_commanded":    tf.TensorSpec((1,), tf.float32),
        },
        ...
    })
}
```

Figure 12: The data structure of a raw episode in the RT-1 dataset

### B.2 Stage 1 & 2: Raw Episode Extraction and Key Frame Pair Selection

The curation process begins by extracting raw trajectory episodes from the source datasets. Each episode contains sequences of state images and corresponding 7-Degree-of-Freedom (7DoF) end-effector state vectors. From these raw sequences, we perform a rigorous key frame pair selection to ensure that every data point used in ACTIONEQA is of high quality and represents a meaningful physical interaction.
This selection stage filters out low-quality or uninformative data using several key heuristics:

- **Visual Quality Filtering:** We automatically discard any image pairs $(s_t, s_{t+1})$ that exhibit significant motion blur, poor or inconsistent lighting, or visual obstruction of the robot's end-effector. This step ensures that the benchmark fairly assesses a model's reasoning capabilities without being confounded by corrupted visual input.

- **Trivial Action Filtering:** To focus the benchmark on substantial and clearly discernible robot actions, we filter out "trivial motions" by enforcing thresholds on the 7DoF action vector. A frame pair is selected only if the action is significant enough to cause a visually meaningful change. Due to variations in camera proximity to the action in the source datasets, we established dataset-specific thresholds. For a frame pair to be selected, at least two dimensions of its 7DoF action vector must exceed their corresponding thresholds. This prevents the selection of frames with minor, uni-dimensional adjustments and favors more complex, multi-dimensional movements.

  The thresholds for translational movements (in meters), rotational movements (in degrees), and gripper action (normalized) are defined as follows:

  - **BridgeData V2:** $[0.1m, 0.1m, 0.1m, 18°, 18°, 18°, 0.2]$ — Used for this dataset as the cameras are generally positioned closer to the manipulation scene.
  - **DROID:** $[0.18m, 0.18m, 0.18m, 18°, 18°, 18°, 0.2]$ — A higher translational threshold is used for this dataset because the cameras are typically positioned further from the scene, requiring a larger movement to be visually significant.

  This dual-threshold, multi-dimensional filtering approach guarantees that every selected $(s_t, a_t, s_{t+1})$ triplet represents an unambiguous and interpretable change in the physical environment, forming a solid basis for generating meaningful questions.

### B.3 Stage 3: Ground-Truth Action Extraction

After curating a high-quality set of key frame pairs, the next stage is to establish the ground-truth action representations across all three levels of our action hierarchy. This creates the multi-level ground truth necessary for ACTIONEQA's fine-grained diagnostic evaluation.

- **High-Level Action ($a_t^{\textbf{high}}$):** The ground-truth high-level action is extracted directly from the natural language task description provided in the source datasets (e.g., "Put the lemon on the plate"). This captures the "what," or the overall semantic intent of the robot's action. A list of representative high-level actions can be found in Table 7.

- **Low-Level Action ($a_t^{\textbf{low}}$):** This is the raw 7-DoF command vector, which defines the "physical how." It is generated by subtracting the end-effector state at $s_t$ from the state at $s_{t+1}$, resulting in a vector: $[\Delta x, \Delta y, \Delta z, \Delta\text{roll}, \Delta\text{pitch}, \Delta\text{yaw}, \Delta\text{gripper}]$.

- **Mid-Level Action ($a_t^{\textbf{mid}}$):** To represent the "semantic how," we translate the numerical low-level action ($a_t^{\text{low}}$) into a structured, language-based description. This is achieved by mapping the low-level vector to a predefined set of **14 customized mid-level action labels**. As detailed in Table 6, these labels describe fundamental motion primitives, such as "Move along positive X direction" or "Tilt clockwise around Y-axis," providing a compositional description of the end-effector's movement.

### B.4 Stage 4: Distractor Action Generation

A critical step in creating a challenging benchmark is the generation of plausible but incorrect "distractor" options for our multiple-choice questions. To ensure the distractors are relevant and require nuanced reasoning to differentiate from the correct answer, we employ distinct strategies for each level of the action hierarchy.

| Motion Type | Action (+) | Action (-) |
|---|---|---|
| *Translational Motions* | | |
| X-axis | Move along positive X direction | Move along negative X direction |
| Y-axis | Move along positive Y direction | Move along negative Y direction |
| Z-axis | Move along positive Z direction | Move along negative Z direction |
| *Rotational Motions* | | |
| Roll | Tilt clockwise around X-axis | Tilt counterclockwise around X-axis |
| Pitch | Tilt downward around Y-axis | Tilt upward around Y-axis |
| Yaw | Rotate counterclockwise around Z-axis | Rotate clockwise around Z-axis |
| *Gripper Actuation* | | |
| Gripper | Open gripper | Close gripper |

Table 6: The 14 mid-level action labels, organized into 7 opposing pairs. The (+) and (-) columns correspond to actions treated as positive and negative directions in this work, respectively.

| Data Source | Unique High-Level Actions |
|---|---|
| DROID | `pick, place, put, remove, take` |
| Bridge V2 | `flip, fold, move, pick, put` |
| RT-1 | `move, pick, place` |

Table 7: Unique high-level actions for each data source in ActionEQA.

**High-Level Distractors.** For high-level actions, we employ a sophisticated prompt-based strategy to generate distractors that are both plausible and subtly incorrect. We use a powerful vision-language model, Qwen-QVQ-72B, which is provided with the initial image ($s_t$), the final image ($s_{t+1}$), and the ground-truth action description ($a_t^{\text{high}}$). As detailed in the prompt illustrated in Figure 26, the model is instructed to generate three alternative actions. These distractors are designed to be stylistically consistent with the ground truth while introducing a variety of specific, challenging errors. These errors include applying the correct action to the wrong object, using the correct object but performing the wrong action, or describing a plausible alternative outcome that did not actually occur. The resulting options are then reviewed by humans to confirm they are valid, challenging, and unambiguously incorrect.

**Mid-Level Distractors.** To generate challenging mid-level distractors, we first identify the set of ground-truth mid-level action labels that describe the robot's movement. We then generate all possible combinations of these identified action labels. For example, if the ground-truth action is "Move along positive X direction, Move along positive Y direction, and Rotate counterclockwise around Z-axis," this involves three distinct motion components: translation in X, translation in Y, and yaw (rotation around Z). This results in a total of $2^3 = 8$ possible combinations of these three movements. From this pool of combinations, we randomly select three, ensuring they are not identical to the ground-truth action. This method creates distractors that are highly relevant to the correct action but require a precise understanding of the combined movements to correctly identify as incorrect.

**Low-Level Distractors.** Numerical distractors for the low-level are generated using heuristic techniques designed to test a model's grasp of physical transformations. These techniques include inverting the signs of the dominant motion axis (opposite direction), removing the dominant axis, or permuting the axis values (wrong axis).

### B.5 Stage 5: QA Generation and Human Verification

In the final stage, the ground-truth action and the three generated distractors are assembled into multiple-choice questions for our two bidirectional reasoning tasks: **State Prediction** and **Action Inference**. This automated generation is followed by a comprehensive final review process:

- **Format Consistency**: Ensuring all questions and answers adhere to a standardized format.

- **QA Tagging**: Categorizing each question based on the action level and reasoning type.

- **Human Verification:** The most critical step is a meticulous human review of every single question-answer pair. Our annotators check each question for correctness, clarity, ambiguity, and overall fairness. This crucial step catches subtle errors that automated methods might miss and guarantees that the final 8,795 questions in the ActionEQA benchmark are of the highest possible quality, ready for robust model evaluation.

## B.6 Dataset Statistics

| Statistic | DROID | BridgeData V2 | RT-1 | Total |
|---|---|---|---|---|
| Questions | 2953 | 4732 | 1110 | 8795 |
| Unique Images | 8026 | 14,146 | 4144 | 26,213 |
| Unique Episodes | 367 | 1707 | 555 | 2629 |

Table 8: A breakdown of curation statistics for the benchmark across the three data sources.

| QA Type | DROID | | BridgeData V2 | | RT-1 | |
|---|---|---|---|---|---|---|
| | SP | AI | SP | AI | SP | AI |
| High-Level | 353 | 353 | 983 | 983 | 555 | 555 |
| Mid-Level | 509 | 489 | 438 | 890 | N/A | N/A |
| Low-Level | 509 | 740 | 438 | 1000 | N/A | N/A |
| **Total** | **2953** | | **4732** | | **1110** | |

Table 9: A detailed breakdown of the question types for each data source in the ActionEQA.

As illustrated in Figure 3 and Table 8, the final ActionEQA benchmark comprises a total of 8,795 high-quality, multiple-choice questions curated from 2,629 unique episodes and 26,213 images. These questions are systematically distributed across three real-world robotics datasets, two bidirectional reasoning settings, and three levels of the action hierarchy.

The questions are sourced from three distinct datasets to ensure diversity in tasks, environments, and hardware. As detailed in Table 8, BridgeData V2 is the largest contributor with 4,732 questions, followed by DROID with 2,953 questions, and RT-1 with 1,110 questions.

Structurally, the benchmark is designed to evaluate two complementary reasoning abilities. **Action Inference**, which requires backward-looking reasoning, is the more prevalent setting with 5,010 questions. **State Prediction**, which assesses State Prediction-looking predictive capabilities, accounts for the remaining 3,785 questions. These settings are applied across the action hierarchy to form our six core evaluation categories: High-Level State Prediction (H-SP: 1,891), Mid-Level State Prediction (M-SP: 947), Low-Level State Prediction (L-SP: 947), High-Level Action Inference (H-AI: 1,891), Mid-Level Action Inference (M-AI: 1,379), and Low-Level Action Inference (L-AI: 1,740). Table 9 offers a more detailed breakdown of these question types within each data source.

### B.7 Action Distributions

### B.7.1 High-Level Actions

To better understand the semantic diversity of high-level QAs in ACTIONEQA, we analyze the distribution of high-level goal-oriented actions across our curated dataset. Table 10 presents the aggregate frequency of high-level action verbs. The distribution reflects the nature of general-purpose robotic manipulation, which is heavily centered around object rearrangement. Consequently, the fundamental primitives *put* (44.98%), *pick* (21.09%), and *place* (16.62%) constitute the vast majority (82.69%) of the dataset. More specialized interactions such as *fold* and *flip* appear less frequently, representing long-horizon or non-prehensile manipulation tasks.

We further break down these distributions by data source in Table 11 to highlight dataset-specific characteristics. This comparison reveals significant lexical bias inherent to each source dataset. For instance, while BridgeData V2 heavily favors the verb *put* (73.9%), RT-1 exclusively utilizes *place* (44.9%) and *pick* (32.2%) for similar rearrangement tasks, with zero instances of *put*. DROID demonstrates the most balanced vocabulary, uniquely contributing actions like *remove* (14.3%) and *take* (8.5%). Additionally, task-specific actions are localized; *fold* and *flip* are unique to BridgeData V2, reflecting its inclusion of cloth manipulation episodes. This variance underscores the importance of evaluating VLMs across multiple data sources to ensure robustness to synonymous phrasing and diverse task semantics.

| Action | Count | Percentage (%) |
|--------|-------|----------------|
| put    | 1954  | 44.98          |
| pick   | 916   | 21.09          |
| place  | 722   | 16.62          |
| move   | 400   | 9.21           |
| remove | 144   | 3.31           |
| fold   | 92    | 2.12           |
| take   | 86    | 1.98           |
| flip   | 30    | 0.69           |
| **Total** | **4344** | **100.00** |

Table 10: Combined distribution of high-level actions across all datasets.

| Action | DROID | BridgeData V2 | RT-1 |
|--------|-------|---------------|------|
| put    | 49.7% | 73.9%         | 0.0% |
| pick   | 16.2% | 15.9%         | 32.2% |
| place  | 10.7% | 0.0%          | 44.9% |
| move   | 0.6%  | 4.1%          | 23.0% |
| remove | 14.3% | 0.0%          | 0.0% |
| fold   | 0.0%  | 4.7%          | 0.0% |
| take   | 8.5%  | 0.0%          | 0.0% |
| flip   | 0.0%  | 1.5%          | 0.0% |

Table 11: High-level action distribution comparison across datasets.

### B.7.2 Mid-Level Actions

While high-level actions describe the goal, mid-level actions in ACTIONEQA capture the language-based kinematic changes required to achieve them. As shown in Table 12, our benchmark maintains a remarkably balanced distribution across the three fundamental components of manipulation: Translation (35.87%), Rotation (34.00%), and Gripper actuation (30.14%). This balance is crucial for a diagnostic benchmark, ensuring that a model's performance score reflects a comprehensive understanding of all physical articulation types rather than being dominated by simple translational movements.

Table 13 offers a comparative analysis between mid-level QAs derived from the DROID and BridgeData V2 datasets. Interestingly, while the complexity of samples is consistent across datasets, their kinematic signatures differ significantly. DROID is notably rotation-heavy (46.02%), reflecting the complex 7-DoF manipulation tasks often found in that dataset. In contrast, BridgeData V2 is translation-heavy (45.03%), likely due to the wider spatial range of its collected trajectories. This divergence ensures that ACTIONEQA tests a model's ability to ground language into different physical embodiments and workspace constraints.

A granular breakdown of the specific motion primitives is provided in Table 14. Vertical movement (Z-axis) accounts for 15.82% of all actions, and Z-axis rotation (Yaw) accounts for 18.2% of all actions. This prevalence aligns with the standard top-down grasping approach common in robotic manipulation, where the agent must orient the gripper (Yaw) and lower it (Negative Z) to interact with objects. Furthermore, we observe a significant skew in gripper actuation, with *Close Gripper* (21.78%) appearing more frequently than *Open Gripper* (8.35%). This asymmetry is characteristic of manipulation datasets, where the critical, reasoned action is often the active grasping of an object, while release actions are less frequent or less visually distinct in collected trajectories.

| Action Type | Count | Percentage (%) |
|---|---|---|
| Translation | 1821 | 35.87 |
| Rotation | 1726 | 34.00 |
| Gripper | 1530 | 30.14 |
| **Total** | **5077** | **100.00** |

Table 12: Distribution of mid-level action types in ActionEQA.

| | DROID | | | BridgeData V2 | | |
|---|---|---|---|---|---|---|
| Action Type | Count | % | Avg/Sample | Count | % | Avg/Sample |
| Translation | 526 | 23.90 | 0.53 | 1295 | 45.03 | 0.98 |
| Rotation | 1013 | 46.02 | 1.02 | 713 | 24.79 | 0.54 |
| Gripper | 662 | 30.08 | 0.66 | 868 | 30.18 | 0.65 |
| **Total** | **2201** | **100.00** | **2.21** | **2876** | **100.00** | **2.17** |

Table 13: Comparison of mid-level action distributions between DROID and BridgeData V2 datasets.

| Type | Action | Count | Percentage (%) |
|------|--------|-------|----------------|
| *Translation* | Z+ (Positive Z) | 464 | 9.14 |
| | Y+ (Positive Y) | 425 | 8.37 |
| | Z- (Negative Z) | 339 | 6.68 |
| | Y- (Negative Y) | 315 | 6.20 |
| | X+ (Positive X) | 147 | 2.90 |
| | X- (Negative X) | 131 | 2.58 |
| *Rotation* | Yaw- (CW around Z) | 466 | 9.18 |
| | Yaw+ (CCW around Z) | 458 | 9.02 |
| | Roll+ (CW around X) | 265 | 5.22 |
| | Pitch+ (Down around Y) | 236 | 4.65 |
| | Pitch- (Up around Y) | 172 | 3.39 |
| | Roll- (CCW around X) | 129 | 2.54 |
| *Gripper* | Gripper- (Close) | 1106 | 21.78 |
| | Gripper+ (Open) | 424 | 8.35 |
| **Total** | | **5077** | **100.00** |

Table 14: Distribution of mid-level actions across all datasets in ActionEQA.

### B.7.3   Low-Level Actions

At the lowest level of the hierarchy, ACTIONEQA evaluates the precise physical execution of commands. We analyze the distribution of these actions using 7-DoF vectors $[\Delta x, \Delta y, \Delta z, \Delta \text{roll}, \Delta \text{pitch}, \Delta \text{yaw}, \Delta \text{gripper}]$. To quantify the magnitude of these actions, we compute the Euclidean norm for translation in meters and rotation in degrees, and the absolute change for gripper actuation measured as a continuous value from 0 to 1.

Table 15 and Table 16 provide a detailed statistical summary of these magnitudes across low-level QAs derived from the DROID and BridgeData V2 datasets. A key observation is that the actions in our benchmark represent substantial physical movements rather than trivial micro-adjustments. The average translational movement is 16.0 cm for DROID and 12.2 cm for BridgeData V2, confirming that the benchmark evaluates meaningful state transitions.

The comparison between datasets reveals distinct physical characteristics. DROID consistently exhibits larger action magnitudes across both translation ($\mu = 0.1603$ m) and rotation ($\mu = 28.07°$) compared to BridgeData V2 ($\mu = 0.1218$ m and $\mu = 20.99°$, respectively). This difference is likely attributable to the hardware disparities. DROID utilizes a Franka Emika Panda arm with a larger workspace, whereas BridgeData V2 employs a smaller WidowX 250 robot. The high standard deviations observed in both datasets indicate a high degree of variance in the evaluation data, which ensures that models are tested on a diverse range of spatial orientations. The distribution of these magnitudes is further visualized in Figure 13.

| Dataset | Magnitude Type | Mean | Std | Median | Q25 | Q75 | Min | Max |
|---|---|---|---|---|---|---|---|---|
| DROID | Translation (m) | 0.1603 | 0.0757 | 0.1554 | 0.0983 | 0.2103 | 0.0014 | 0.4160 |
| | Rotation (°) | 28.0662 | 18.9103 | 22.1797 | 19.2778 | 27.8144 | 3.8514 | 126.5623 |
| | Gripper | 0.3715 | 0.3406 | 0.2600 | 0.0000 | 0.7004 | 0.0000 | 1.0000 |
| BridgeData V2 | Translation (m) | 0.1218 | 0.0502 | 0.1189 | 0.0901 | 0.1459 | 0.0074 | 0.3173 |
| | Rotation (°) | 20.9946 | 12.8826 | 19.3699 | 11.7905 | 25.5753 | 1.6132 | 96.7925 |
| | Gripper | 0.3484 | 0.2908 | 0.3600 | 0.0010 | 0.5065 | 0.0000 | 0.9574 |

Table 15: Low-Level Action Magnitude Statistics across Datasets

| Dataset | Translation (m) | Rotation (°) | Gripper |
|---|---|---|---|
| DROID | $0.1603 \pm 0.0757$ | $28.07 \pm 18.91$ | $0.3715 \pm 0.3406$ |
| BridgeData V2 | $0.1218 \pm 0.0502$ | $20.99 \pm 12.88$ | $0.3484 \pm 0.2908$ |

Table 16: Low-Level Action Magnitude Summary (Mean ± Std)

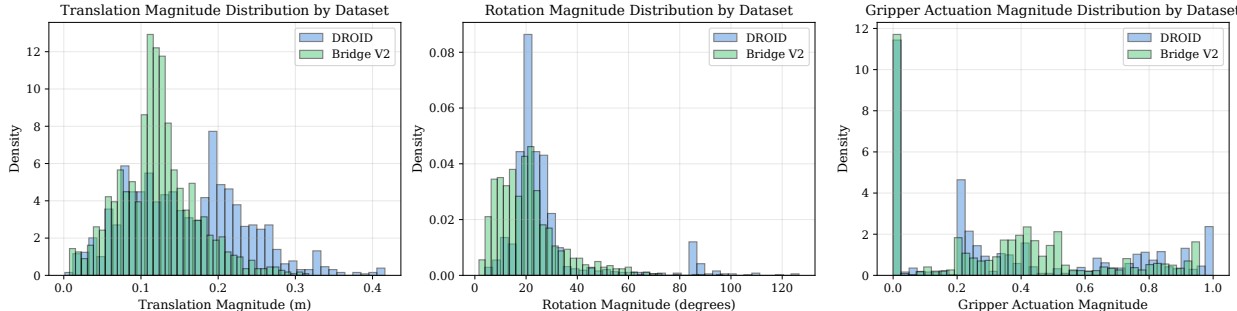

Figure 13: Distribution of low-level action magnitudes across DROID and BridgeData V2 datasets.

### B.8 Answer Key Distribution

To validate the fairness of ACTIONEQA, we conducted a rigorous statistical analysis of the ground-truth answer key distribution across the entire benchmark.

As demonstrated in Table 17, the distribution of the four answer options across the full dataset is highly uniform. Each option appears as the correct answer approximately 25% of the time, with deviations ranging from only -0.22% to +0.36%. A Chi-square goodness-of-fit test yields a statistic of $\chi^2 = 0.66$ ($p = 0.88$), indicating no statistically significant deviation from a perfectly uniform distribution. This confirms that there is no global position bias that a model could exploit.

To ensure that biases do not exist at the source level, we analyzed the three constituent datasets independently. As shown in Table 18, BridgeData V2 ($p = 0.66$), DROID ($p = 0.74$), and RT-1 ($p = 0.57$) all exhibit uniform answer distributions. This consistency indicates that our distractor generation and shuffling protocols remained robust across different data collection procedures and environments.

Finally, Table 19 provides a detailed breakdown across all 14 evaluation settings, splitting the data by source, bidirectional task type (State Prediction vs. Action Inference), and hierarchy level (High/Mid/Low). While minor fluctuations exist due to smaller sample sizes in specific sub-categories, there are no systematic patterns favoring any specific answer position.

Collectively, these results confirm that ACTIONEQA is free from answer position artifacts. Consequently, model performance on this benchmark can be attributed to genuine visual-action reasoning rather than the exploitation of dataset imbalances.

| Answer Key | Count | Percentage (%) | Expected | Deviation |
|:---:|:---:|:---:|:---:|:---:|
| A | 2190 | 24.90 | 2198.8 | -8.8 |
| B | 2179 | 24.78 | 2198.8 | -19.8 |
| C | 2230 | 25.36 | 2198.8 | +31.2 |
| D | 2196 | 24.97 | 2198.8 | -2.8 |
| **Total** | **8795** | **100.00** | | |

Table 17: Overall answer key distribution across all datasets. Chi-square test: $\chi^2 = 0.66$, $p = 0.8826$.

| Dataset | A (%) | B (%) | C (%) | D (%) | $\chi^2$ (p-value) |
|:---:|:---:|:---:|:---:|:---:|:---:|
| BridgeData V2 | 24.9 | 24.5 | 25.7 | 24.9 | 1.61 (0.6565) |
| DROID | 24.6 | 25.8 | 24.6 | 25.0 | 1.24 (0.7443) |
| RT-1 | 25.9 | 23.2 | 25.8 | 25.0 | 2.03 (0.5664) |

Table 18: Answer key distribution by dataset (percentages and chi-square test results).

| Dataset | Task Type | A (%) | B (%) | C (%) | D (%) | n |
|---|---|---|---|---|---|---|
| BridgeData V2 | High-SP | 24.0 | 25.2 | 27.0 | 23.8 | 983 |
| BridgeData V2 | Mid-SP | 24.4 | 26.5 | 24.7 | 24.4 | 438 |
| BridgeData V2 | Low-SP | 26.9 | 24.7 | 22.4 | 26.0 | 438 |
| BridgeData V2 | High-AI | 25.7 | 23.7 | 26.8 | 23.8 | 983 |
| BridgeData V2 | Mid-AI | 23.1 | 25.5 | 27.2 | 24.2 | 890 |
| BridgeData V2 | Low-AI | 25.6 | 22.6 | 24.2 | 27.6 | 1000 |
| DROID | High-SP | 24.4 | 27.8 | 25.2 | 22.7 | 353 |
| DROID | Mid-SP | 23.6 | 24.6 | 27.3 | 24.6 | 509 |
| DROID | Low-SP | 24.4 | 27.3 | 25.7 | 22.6 | 509 |
| DROID | High-AI | 25.5 | 24.1 | 23.8 | 26.6 | 353 |
| DROID | Mid-AI | 28.0 | 27.0 | 21.7 | 23.3 | 489 |
| DROID | Low-AI | 22.8 | 24.9 | 23.9 | 28.4 | 740 |
| RT-1 | High-SP | 26.8 | 24.1 | 23.8 | 25.2 | 555 |
| RT-1 | High-AI | 25.0 | 22.3 | 27.7 | 24.9 | 555 |

Table 19: Detailed answer key distribution for all dataset variants.

# C  Additional Experimental Details

| Company | Model Name | Release Date | Official Name | Evaluation Pipeline |
|---|---|---|---|---|
| *Open-Weights Models* | | | | |
| Google | Gemma-3-27b-it | 2025-03 | `google/gemma-3-27b-it` | Gemini API |
| | Gemma-3-12b-it | 2025-03 | `google/gemma-3-12b-it` | Gemini API |
| | Gemma-3-4b-it | 2025-03 | `google/gemma-3-4b-it` | Gemini API |
| Zhipu AI | GLM-4.5V | 2025-08 | `GLM-4.5V` | Zhipu Foundation Model Open Platform API |
| | GLM-4.1V-Thinking | 2025-07 | `GLM-4.1V-Thinking-FlashX` | Zhipu Foundation Model Open Platform API |
| | GLM-4V | 2024-08 | `GLM-4V-Plus-0111` | Zhipu Foundation Model Open Platform API |
| Baidu | ERNIE-4.5-VL-28B-A3B-PT | 2025-06 | `baidu/ERNIE-4.5-VL-28B-A3B-PT` | ModelScope API |
| Meta | Llama-4-Scout-17B-16E-Ins | 2025-04 | `meta-llama/Llama-4-Scout-17B-16E-Instruct` | ModelScope API |
| | Llama-4-Mav-17B-128E-Ins | 2025-04 | `meta-llama/Llama-4-Mav-17B-128E-Instruct` | ModelScope API |
| Shanghai AI Lab | InternVL3-14B | 2025-04 | `OpenGVLab/InternVL3-14B` | Hugging Face Transformers |
| | InternVL3-8B | 2025-04 | `OpenGVLab/InternVL3-8B` | Hugging Face Transformers |
| | InternVL3-2B | 2025-04 | `OpenGVLab/InternVL3-2B` | Hugging Face Transformers |
| | InternVL2.5-8B-MPO | 2025-04 | `OpenGVLab/InternVL2.5-8B-MPO` | Hugging Face Transformers |
| | InternVL2.5-2B-MPO | 2025-04 | `OpenGVLab/InternVL2.5-2B-MPO` | Hugging Face Transformers |
| | InternVL2.5-8B | 2024-12 | `OpenGVLab/InternVL2.5-8B` | Hugging Face Transformers |
| | InternVL2.5-2B | 2024-12 | `OpenGVLab/InternVL2.5-2B` | Hugging Face Transformers |
| Alibaba | Qwen3-VL-8B-Ins | 2025-10 | `Qwen/Qwen3-VL-8B-Instruct` | Hugging Face Transformers |
| | Qwen2.5-VL-72B-Ins | 2025-01 | `Qwen/Qwen2.5-VL-72B-Instruct` | ModelScope API |
| | Qwen2.5-VL-32B-Ins | 2025-01 | `Qwen/Qwen2.5-VL-32B-Instruct` | ModelScope API |
| | Qwen2.5-VL-7B-Ins | 2025-01 | `Qwen/Qwen2.5-VL-7B-Instruct` | Hugging Face Transformers |
| | Qwen2.5-VL-3B-Ins | 2025-01 | `Qwen/Qwen2.5-VL-3B-Instruct` | Hugging Face Transformers |
| | Qwen2-VL-7B-Ins | 2024-08 | `Qwen/Qwen2-VL-7B-Instruct` | Hugging Face Transformers |
| | Qwen2-VL-2B-Ins | 2024-08 | `Qwen/Qwen2-VL-2B-Instruct` | Hugging Face Transformers |
| AIDC | Ovis2.5-9B | 2025-08 | `AIDC-AI/Ovis2.5-9B` | Hugging Face Transformers |
| | Ovis2.5-2B | 2025-08 | `AIDC-AI/Ovis2.5-2B` | Hugging Face Transformers |
| OpenBMB | MiniCPM-V-4.5 | 2025-08 | `openbmb/MiniCPM-V-4.5` | Hugging Face Transformers |
| *Proprietary Models* | | | | |
| OpenAI | o4-mini | 2025-04 | `o4-mini-2025-04-16` | OpenAI API |
| | GPT-4.1 | 2025-04 | `gpt-4.1-2025-04-14` | OpenAI API |
| Google | Gemini-2.5-Pro | 2025-06 | `gemini-2.5-pro` | Gemini API |
| | Gemini-2.5-Flash | 2025-06 | `gemini-2.5-flash` | Gemini API |
| | Gemini-2.5-Flash-Lite | 2025-06 | `gemini-2.5-flash-lite` | Gemini API |
| | Gemini-2.0-Flash | 2025-02 | `gemini-2.0-flash` | Gemini API |
| Anthropic | Claude-Opus-4 | 2025-05 | `claude-opus-4-20250514` | Anthropic API |
| | Claude-Sonnet-4 | 2025-05 | `claude-sonnet-4-20250514` | Anthropic API |

Table 20: Details of Vision Language Models (VLMs) assessed on ActionEQA

## C.1  Evaluated Vision-Language Models

To comprehensively assess the capabilities of contemporary vision-language models (VLMs), we evaluated a diverse set of 34 leading open-source and proprietary models on our new ACTIONEQA benchmark. This evaluation provides a thorough analysis of the challenges posed by ACTIONEQA and establishes a baseline for future research aimed at improving different levels of action understanding in embodied agents.

**Open-Source Models.**  We evaluated 14 series of prominent open-source VLMs, covering a range of architectures and parameter scales. This set includes: Gemma 3 (27B, 12B, 4B) (Team et al., 2025), GLM-4.5V (Hong et al., 2025), GLM-4.1V-Thinking (Hong et al., 2025), GLM-4V-9B (Wang et al., 2024c; GLM et al., 2024), ERNIE-4.5-VL-28B-A3B-PT (Baidu ERNIE Team, 2025), Llama-4Scout-17B-16f-Ins & Llama-4-Maverick-17B-128e-Ins (Meta, 2025), InternVL3 (14B, 8B, 2B) (Zhu et al., 2025), InternVL2.5-MPO (8B, 2B) (Wang et al., 2024d), InternVL2.5 (8B, 2B) (Chen et al., 2024), Qwen3-VL (8B) (Yang et al., 2025a), Qwen2.5-VL (72B, 32B, 7B, 3B) (Bai et al., 2025a), Qwen2-VL (7B, 2B) (Wang et al., 2024a), , MiniCPM-V-4.5 (Yu et al., 2025; Yao et al., 2025) ,and Ovis2.5 (9B, 2B) (Lu et al., 2025).

**Proprietary Models.**  For a comparative perspective against closed-source counterparts, we benchmarked eight state-of-the-art proprietary models: Google's Gemini-2.5-Pro, Gemini-2.5-Flash, Gemini-2.5-Flash-Lite, and Gemini-2.0-Flash (Comanici et al., 2025); OpenAI's GPT-4.1 (OpenAI, 2025a) and o4-mini (OpenAI, 2025c;b); and Anthropic's Claude-Sonnet-4 and Claude-Opus-4 (Anthropic, 2025). These models, all accessible via API, represent the forefront of commercially available multimodal foundation models.

**Evaluated Model Configurations.**  The specific model versions and the evaluation inference pipeline for each model are detailed in Table 20. Vision input was processed using the default settings provided by each

model's official implementation. To guarantee reproducibility, all models were configured with a temperature of 0 and evaluated under the zero-shot setting, with a maximum output length of 2048 tokens. Open-weights models smaller than 14B were evaluated via Hugging Face Transformers on two Linux workstations, with each machine configured with an NVIDIA 4090 (24 GB) GPU.

## C.2 Evaluation Protocol

The primary metric for evaluation is **accuracy**. Each question is considered correct only if the model's response exactly matches the ground-truth answer. These accuracy scores form the basis for our overall performance calculation. To ensure a balanced assessment, we compute the overall performance using a hierarchical averaging method. We define $M$ as the set of all unique metric types (e.g., $M = \{$H-SP, M-SP, ...$\}$). For each metric type $m \in M$, we denote $D_m$ as the set of all its reported accuracy scores, where $S_{m,i}$ is the $i$-th score within that set.

Our calculation begins by finding the average score for each metric type, $\bar{S}_m$, by summing its reported accuracy scores and dividing by the count of those scores, $|D_m|$.

$$\bar{S}_m = \frac{1}{|D_m|} \sum_{i=1}^{|D_m|} S_{m,i} \tag{1}$$

Next, the final overall performance, $P_{\text{overall}}$, is calculated as the arithmetic mean of the average scores for each of the $|M|$ unique metric types.

$$P_{\text{overall}} = \frac{1}{|M|} \sum_{m \in M} \bar{S}_m \tag{2}$$

This approach guarantees that each fundamental skill category contributes equally to the final score.

## C.3 Human Performance Evaluation

To establish a clear upper bound for performance and to contextualize our model evaluations, we conducted a thorough human performance evaluation on the ACTIONEQA benchmark. We recruited two professional annotators and assigned them disjoint subsets of the question-answering items, such that each annotator completed half of the items with no overlap. The evaluation was administered through our custom-built, intuitive user interface developed with Gradio, as demonstrated in Figures 14, 15, 16, 17, 18, 19. Critically, to ensure a direct and equitable comparison, the human annotators received the exact same visual inputs, instructions, and multiple-choice options as the VLMs.

To verify the reliability and consistency of human judgments on our benchmark, we calculated the Inter-Annotator Agreement (IAA). For this analysis, we randomly sampled 20 items from each of the six distinct QA categories, creating a representative subset of 120 items. Both annotators completed this subset independently to allow for a direct comparison of their responses.

Our primary metric was **Raw Percentage Agreement**, which calculates the proportion of items where both annotators chose the identical answer (Lombard et al., 2002). On the 120-item subset, the annotators achieved a high raw agreement rate of 95.8%. To assess the statistical stability of this finding, we computed a 95% confidence interval (CI) using the bootstrap percentile method. This procedure involved generating 1,000 bootstrap resamples from the 120 items and calculating the agreement for each, yielding a narrow 95% CI of [92.5%, 98.3%], confirming the robustness of the agreement score.

Furthermore, to account for the possibility of agreement occurring by chance in a multiple-choice format, we also computed **Cohen's Kappa ($\kappa$)** (Cohen, 1960). Given the four-option question structure, the probability of chance agreement $P_e$ is 0.25. With an observed agreement $P_o$ of 0.958, the resulting Cohen's Kappa was $\kappa = 0.944$. According to established interpretive scales, this value signifies "almost perfect" agreement (Landis & Koch, 1977). To further ensure that no specific level of the action hierarchy introduced undue ambiguity, we decomposed the agreement analysis by action tier, as detailed in Table 21. Taken together, these strong agreement metrics underscore the unambiguous nature of the tasks within the ACTIONEQA benchmark. They also validate that the aggregated human performance reported in the main paper serves as a reliable and meaningful ceiling for evaluating model capabilities.

| Action Level | Raw Agreement | 95% CI | Cohen's Kappa ($\kappa$) |
|:---:|:---:|:---:|:---:|
| High-Level | 97.5% | $[92.5\%, 100.0\%]$ | 0.967 |
| Mid-Level | 97.5% | $[92.5\%, 100.0\%]$ | 0.967 |
| Low-Level | 92.5% | $[85.0\%, 97.5\%]$ | 0.900 |
| **Overall** | **95.8%** | $[\mathbf{92.5\%, 98.3\%}]$ | **0.944** |

Table 21: Breakdown of Inter-Annotator Agreement (IAA) across the three tiers of the action hierarchy.

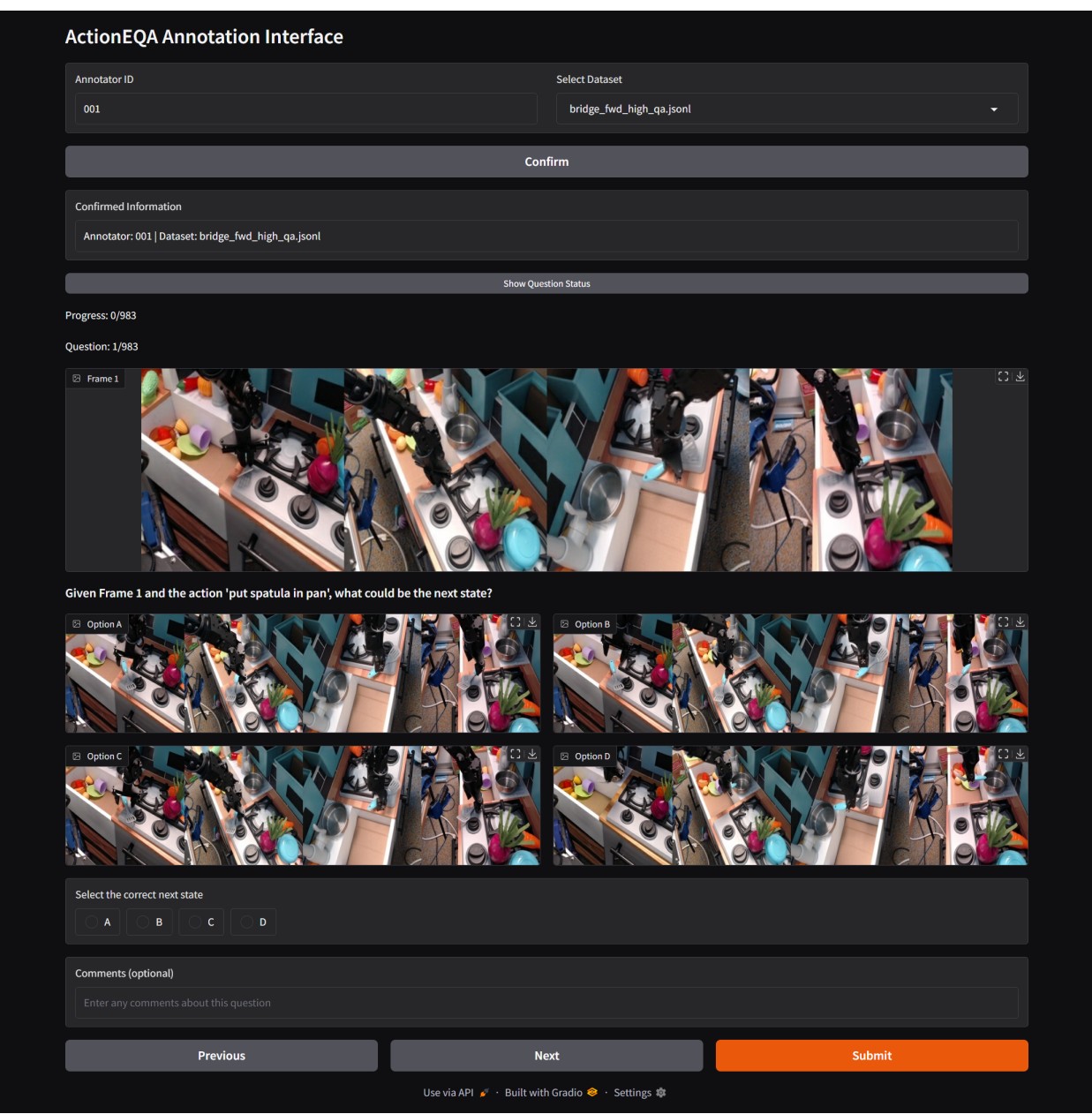

Figure 14: Example of a **High-Level State Prediction** task on our customized Gradio-based annotation interface for evaluating human performance. In this interface, annotators are presented with a current state image, a high-level goal-oriented action, and four candidate images representing possible next states. Their task is to select the image that best matches the most likely next state.

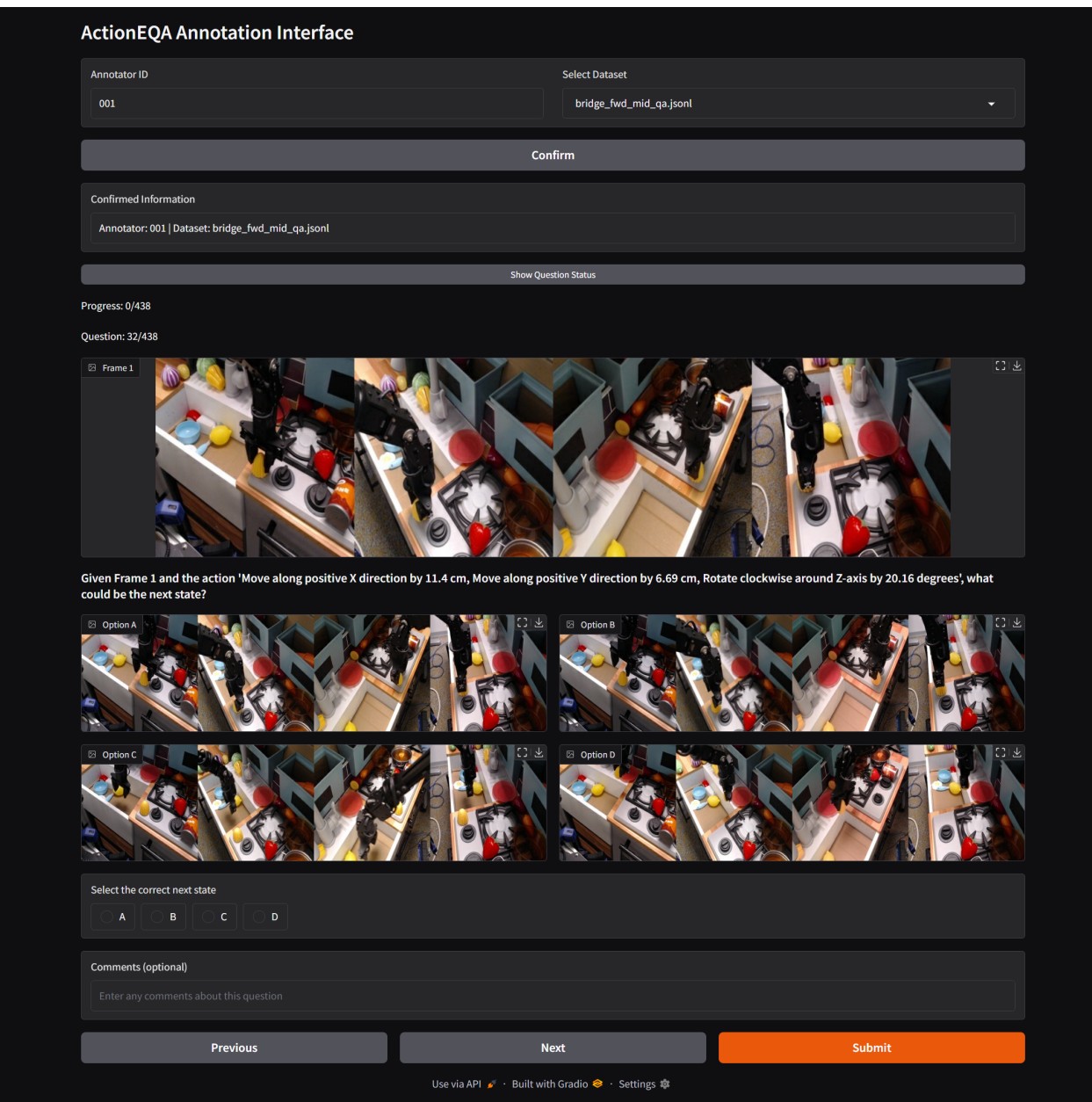

Figure 15: Example of a **Mid-Level State Prediction** task on our customized Gradio-based annotation interface for evaluating human performance. In this interface, annotators are presented with a current state image, a mid-level natural language motion description, and four candidate images representing possible next states. Their task is to select the image that best matches the most likely next state.

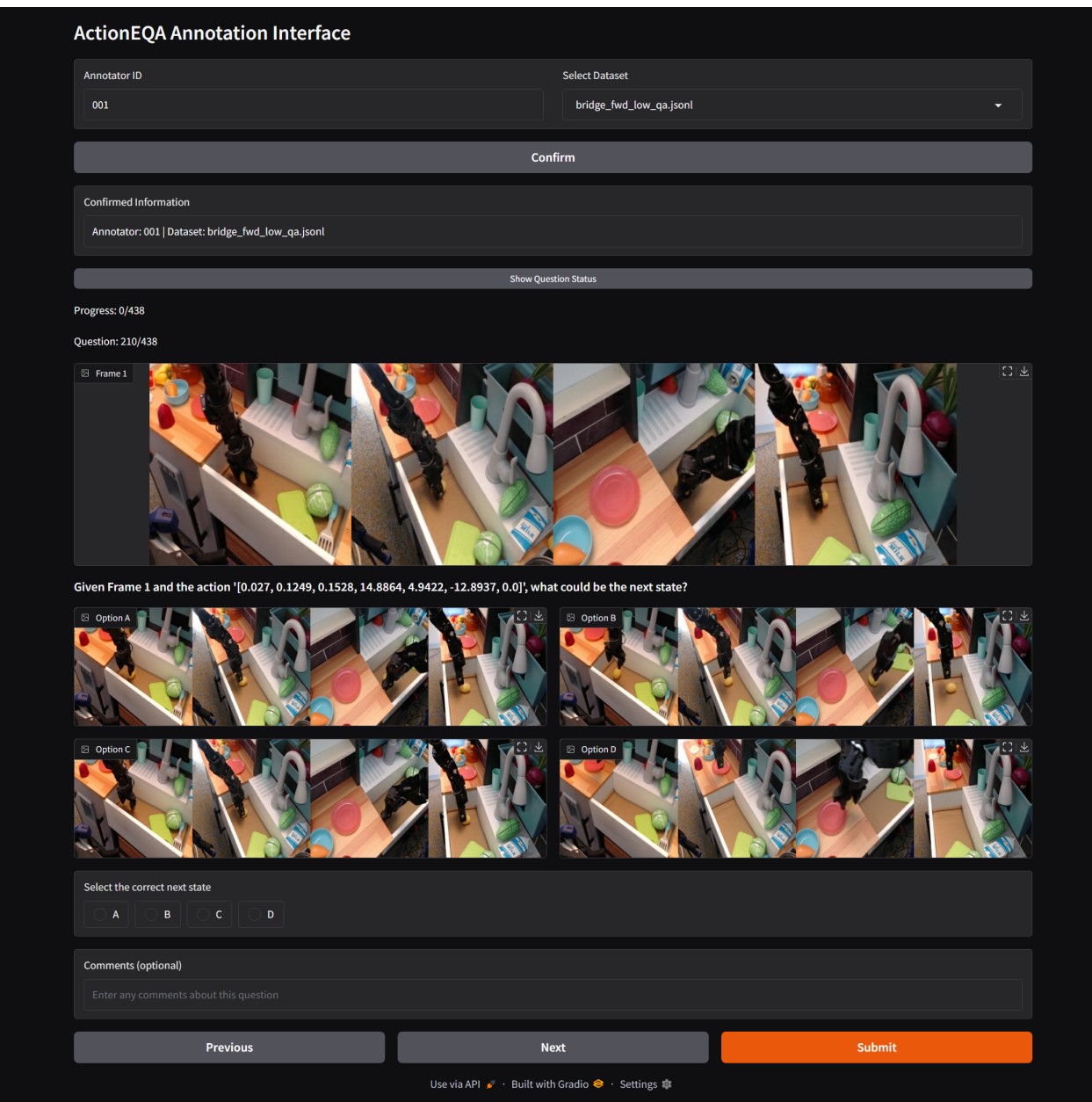

Figure 16: Example of a **Low-Level State Prediction** task on our customized Gradio-based annotation interface for evaluating human performance. Annotators are shown a current state image, a low-level 7-DoF motor command specifying relative changes in translational motion (x, y, z), rotational motion (roll, pitch, yaw), and gripper width, and four candidate images representing possible next states. Their task is to select the image that best corresponds to the most likely next state.

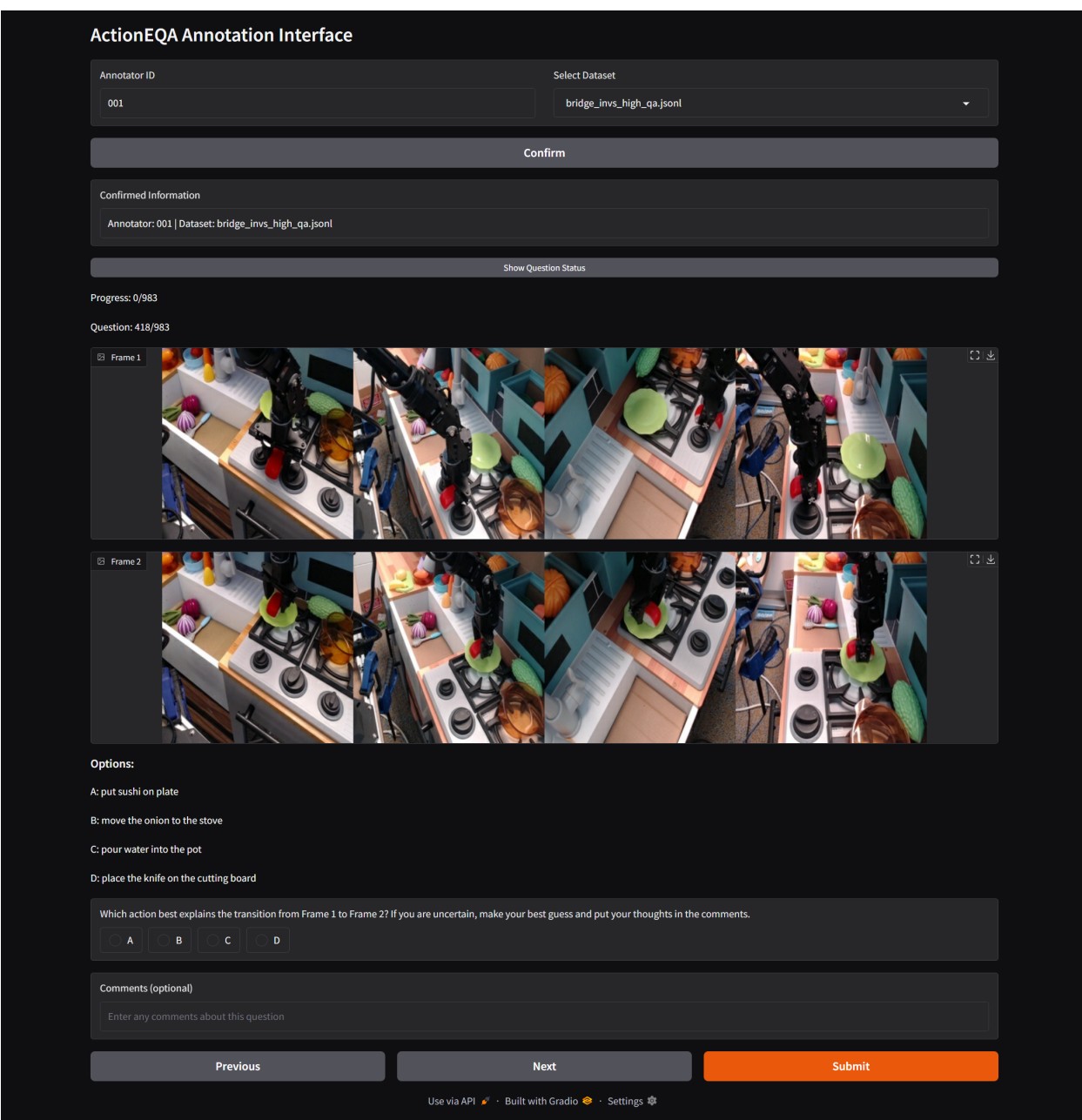

Figure 17: Example of a **High-Level Action Inference** task on our customized Gradio-based annotation interface for evaluating human performance. Annotators are shown a current state image and its resulting next state image, along with four candidate goal-oriented action descriptions. Their task is to identify the action most likely responsible for the observed transition.

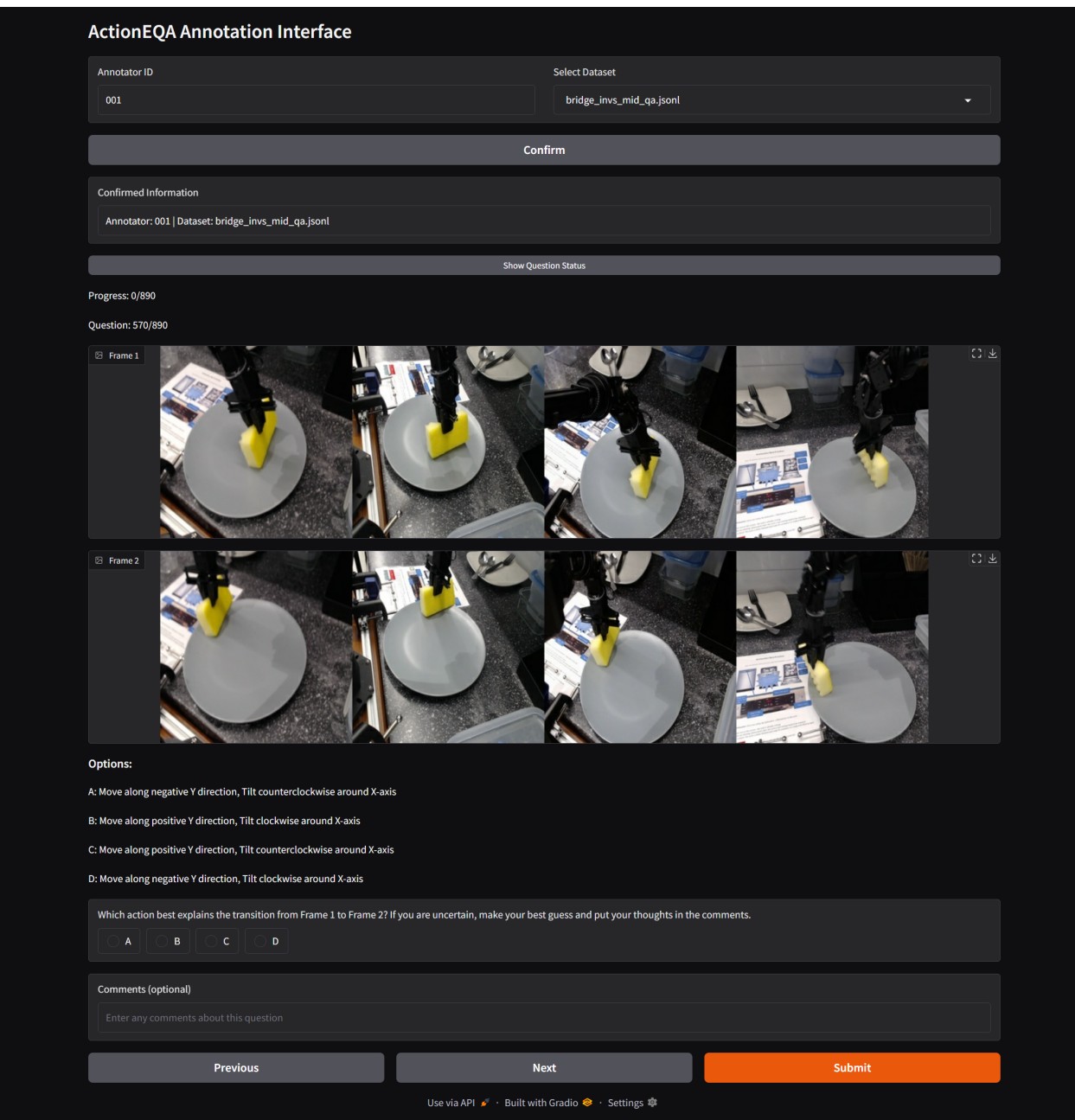

Figure 18: Example of a **Mid-Level Action Inference** task on our customized Gradio-based annotation interface for evaluating human performance. Annotators are shown a current state image and its resulting next state image, along with four candidate natural language motion descriptions. Their task is to identify the action most likely responsible for the observed transition.

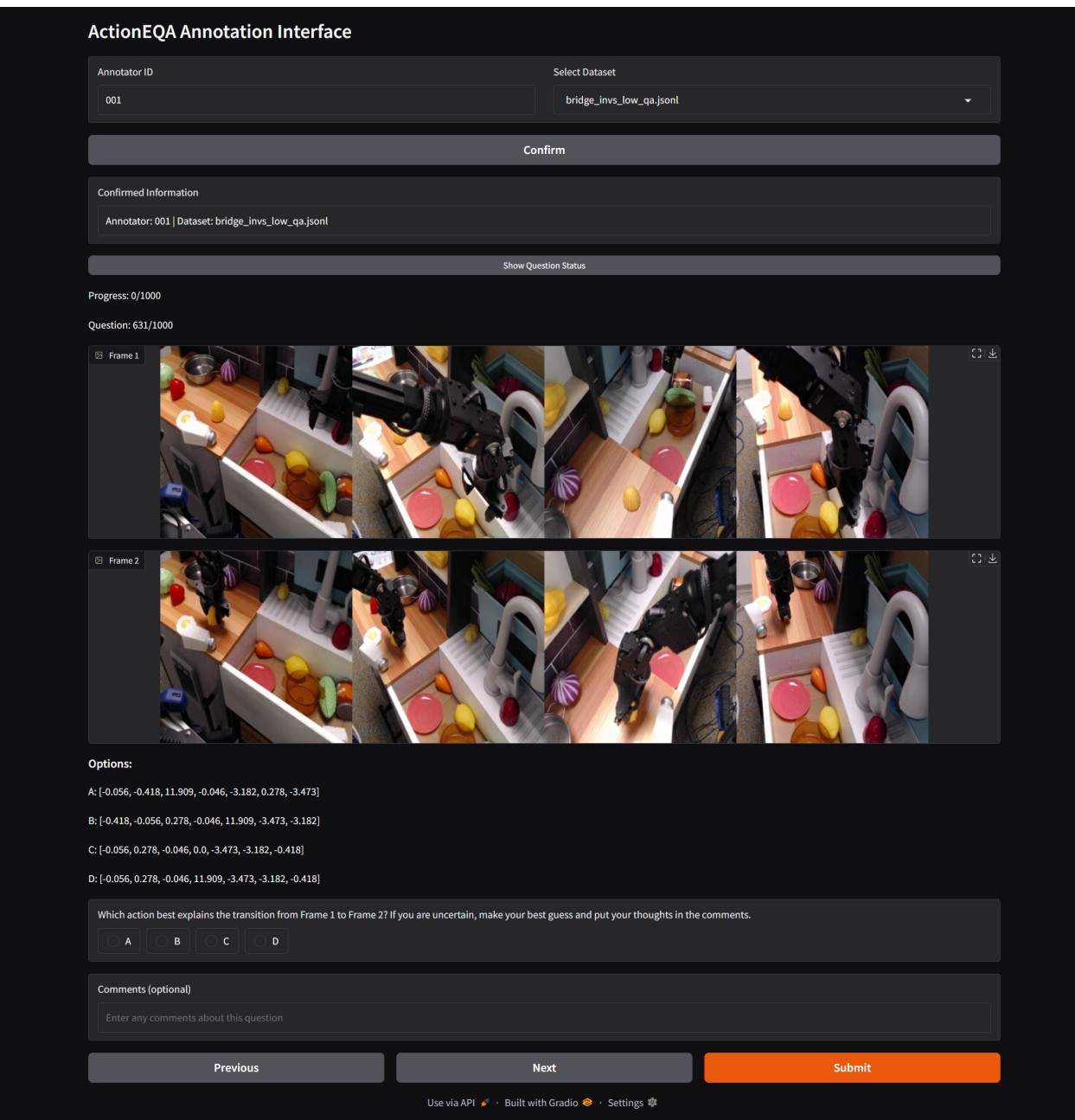

Figure 19: Example of a **Low-Level Action Inference** task on our customized Gradio-based annotation interface for evaluating human performance. Annotators are presented with a current state image and its resulting next state image, along with four candidate low-level 7-DoF motor commands. Each command specifies relative changes in translational motion (x, y, z), rotational motion (roll, pitch, yaw), and gripper width. The task is to identify the action most likely responsible for the observed transition.

### C.4 Distractor Calibration and Quality Analysis

To substantiate the validity of the ACTIONEQA benchmark and ensure that the observed performance bottlenecks are not artifacts of label noise or ambiguous multiple-choice options, we conducted an additional rigorous distractor calibration analysis. We constructed a stratified random sample of 150 QA pairs from the benchmark. To ensure the sample was statistically representative, items were distributed equally across the three action tiers (50 questions per tier) and balanced between the two bidirectional reasoning tasks (25 State Prediction and 25 Action Inference per tier). Additionally, samples were drawn proportionally from DROID, BridgeData V2, and RT-1.

We recruited 3 professional annotators to conduct this evaluation. For each sampled question, human annotators were presented with the visual context and the set of three distractors (blinded to the ground truth) through our custom Gradio-based annotation interface, as shown in Figure 20. They evaluated the quality of distractors using two 5-point Likert scales:

1. **Ambiguity Score (1–5):** Measures whether any distractor could be considered correct based on the visual evidence. A score of 1 indicates that all distractors are strictly incorrect and the ground truth is unique. A score of 5 indicates that one or more distractors are factually compatible with the scene.

2. **Plausibility Score (1–5):** Measures the realism of the distractors. A score of 1 indicates trivial or nonsense distractors, while 5 indicates highly realistic, physically valid actions that fit the scene context but are incorrect.

As shown in Table 22, the benchmark demonstrates exceptional quality. The average ambiguity score of 1.15/5.00 confirms that ambiguous items are extremely rare. Simultaneously, the average plausibility score of 4.55/5.00 confirms that the distractors are not trivial, representing plausible actions that require robust reasoning to rule out.

| Action Level | Avg. Ambiguity Score ($\downarrow$) (1–5 Scale, 1 is Best) | Avg. Plausibility Score ($\uparrow$) (1–5 Scale, 5 is Best) |
|---|---|---|
| High-Level | 1.08 | 4.42 |
| Mid-Level | 1.21 | 4.51 |
| Low-Level | 1.15 | 4.73 |
| **Average** | **1.15** | **4.55** |

Table 22: Distractor calibration analysis based on 150 questions evaluated independently by 3 professional annotators. Ambiguity measures label noise and Plausibility measures distractor realism. The results show the benchmark is clean yet challenging.

Figure 20: Annotation interface used for distractors calibration analysis

## C.5 Controlled Multi-View Analysis using Synchronized Stereo Pairs

To address concerns that the observed performance decline in multi-view settings might be an artifact of data quality such as temporal misalignment, calibration errors, or timestamp mismatch, we conduct a controlled ablation study using the stereo camera configuration of the DROID dataset. DROID utilizes hardware-synchronized ZED stereo cameras which provide the closest approximation to a perfectly calibrated multi-view setup. These stereo pairs are captured via a simultaneous hardware trigger, ensuring zero temporal drift between frames, and are geometrically rectified to maintain precise spatial alignment.

As shown in Table 23, the results of this controlled study provide additional empirical evidence to support the trend observed in Section 3.3. If the performance drops observed in our primary experiments were merely the result of noisy or unaligned data, the use of perfectly synchronized DROID stereo pairs should have allowed the models to maintain or improve their single-view performance. Instead, we observe a consistent decline in accuracy among top-tier reasoning models. For instance, in High-Level State Prediction, Gemini-2.5-Pro decreases from 52.8 to 48.4, while its Low-Level State Prediction accuracy falls significantly from 48.9 to 40.3. The fact that this decline occurs despite the absence of temporal or geometric noise strongly supports the hypothesis that the challenge lies in the model's inability to effectively fuse and reconcile distinct 3D perspectives.

Furthermore, this experiment reinforces our findings regarding the mid-level bottleneck. Even with perfectly aligned stereo data, Gemini-2.5-Pro's performance on mid-level Action Inference drops from 42.3 to 33.3. This indicates that the difficulty of grounding compositional language into 3D rotational transformations is an inherent reasoning failure that cannot be resolved by providing more visual data, even when that data is perfectly calibrated. The average performance across all tested models confirms this trend, showing a decline in five out of six QA categories in the dual-view configuration. By utilizing hardware-synchronized data to rule out external artifacts, this analysis confirms that the primary bottleneck in multi-view embodied reasoning is the cognitive load placed on the VLM, necessitating more principled approaches to multi-view fusion in future research.

| | Left Camera | | | | | | Left + Right Cameras | | | | | |
| | State Prediction | | | Action Inference | | | State Prediction | | | Action Inference | | |
| Model | H | M | L | H | M | L | H | M | L | H | M | L |
|---|---|---|---|---|---|---|---|---|---|---|---|---|
| Gemini-2.5-Pro | $52.8_{\pm5.2}$ | $44.5_{\pm4.3}$ | $48.9_{\pm4.3}$ | $70.2_{\pm4.8}$ | $42.3_{\pm4.4}$ | $73.8_{\pm3.2}$ | $48.4_{\pm5.2}$ | $38.9_{\pm4.2}$ | $40.3_{\pm4.3}$ | $65.4_{\pm5.0}$ | $33.3_{\pm4.2}$ | $70.7_{\pm3.3}$ |
| Gemini-2.5-Flash | $43.4_{\pm5.2}$ | $33.3_{\pm4.1}$ | $40.2_{\pm4.3}$ | $69.8_{\pm4.8}$ | $33.2_{\pm4.2}$ | $50.6_{\pm3.6}$ | $43.9_{\pm5.2}$ | $33.2_{\pm4.1}$ | $40.9_{\pm4.3}$ | $60.1_{\pm5.1}$ | $31.5_{\pm4.1}$ | $50.0_{\pm3.6}$ |
| GLM-4.5V | $35.3_{\pm5.0}$ | $39.5_{\pm4.2}$ | $37.2_{\pm4.2}$ | $53.7_{\pm5.2}$ | $30.1_{\pm4.1}$ | $43.0_{\pm3.6}$ | $39.9_{\pm5.1}$ | $37.3_{\pm4.2}$ | $39.3_{\pm4.2}$ | $57.5_{\pm5.2}$ | $26.8_{\pm3.9}$ | $47.8_{\pm3.6}$ |
| Gemma-3-27B | $45.8_{\pm5.2}$ | $34.6_{\pm4.1}$ | $29.1_{\pm4.0}$ | $51.6_{\pm5.2}$ | $26.0_{\pm3.9}$ | $32.5_{\pm3.4}$ | $43.1_{\pm5.2}$ | $30.5_{\pm4.0}$ | $27.7_{\pm3.9}$ | $49.3_{\pm5.2}$ | $27.2_{\pm4.0}$ | $35.0_{\pm3.4}$ |
| InternVL3-14B | $46.3_{\pm5.2}$ | $26.7_{\pm3.8}$ | $28.5_{\pm3.9}$ | $59.7_{\pm5.1}$ | $27.4_{\pm4.0}$ | $44.9_{\pm3.6}$ | $44.5_{\pm5.2}$ | $31.8_{\pm4.0}$ | $31.3_{\pm4.0}$ | $61.5_{\pm5.1}$ | $32.5_{\pm4.2}$ | $49.6_{\pm3.6}$ |
| Ovis2.5-9B | $42.7_{\pm5.2}$ | $29.1_{\pm4.0}$ | $29.5_{\pm4.0}$ | $61.5_{\pm5.1}$ | $28.3_{\pm4.0}$ | $55.2_{\pm3.6}$ | $33.4_{\pm4.9}$ | $24.2_{\pm3.7}$ | $30.8_{\pm4.0}$ | $58.9_{\pm5.1}$ | $29.4_{\pm4.0}$ | $59.3_{\pm3.5}$ |
| Average | $44.4_{\pm5.2}$ | $34.6_{\pm4.1}$ | $35.6_{\pm4.2}$ | $61.1_{\pm5.1}$ | $31.2_{\pm4.1}$ | $50.0_{\pm3.6}$ | $42.2_{\pm5.2}$ | $32.6_{\pm4.1}$ | $35.1_{\pm4.1}$ | $58.8_{\pm5.1}$ | $30.1_{\pm4.1}$ | $52.1_{\pm3.6}$ |

Table 23: Ablation study comparing model performance across Left exocentric camera and Left + Right stereo pair on DROID with 95% Wald confidence intervals. H/M/L denote High/Mid/Low-Level action reasoning tasks. Light gray highlights the optimal view configuration for each model and task.

### C.6 Fine-Grained Error Analysis of State Prediction

In the main text, our evaluation shows that State Prediction is generally more challenging than Action Inference. To better understand the sources of model failure, we provide a more fine-grained analysis of the top-performing model Gemini-2.5-Pro across the three levels of State Prediction tasks.

### C.6.1 High-Level State Prediction

For High-Level State Prediction tasks, we categorize performance by action type and introduce an *Error Contribution Ratio* defined as the ratio of an action's error contribution to its frequency in the dataset. A ratio greater than 1.0 indicates that an action is disproportionately difficult relative to its prevalence.

As detailed in Table 24, standard pick-and-place primitives (`put`, `pick`, `place`) exhibit ratios near 1.0 and accuracies hovering around 70%. This indicates that VLMs have established a robust baseline understanding of simple object transportation, where the primary change is translational. However, significant outliers emerge when the action implies complex geometric transformation or state ambiguity. The action `flip` yields the highest error ratio (1.68) and the second-highest error rate (53.33%). This aligns strongly with our findings in the mid-level analysis in Section 3.2, providing additional empirical evidence that the struggle to correctly recognize rotational movement is not merely an artifact of coordinate vectors but a fundamental semantic deficit. Even when predicting a high-level outcome, VLMs struggles to simulate how an object's orientation changes during a flip.

A distinct asymmetry also appears in removal tasks compared to placement tasks. Actions such as `take` (Ratio 1.61) and `remove` (Ratio 1.40) represent a pronounced challenge compared to their inverse, `put` (Ratio 0.96). We hypothesize this stems from the visual ambiguity of the resulting state. While `put` usually results in an object being clearly visible on a surface, `remove` or `take` often results in the object being occluded or held in the gripper, requiring the model to predict "absence" or "containment" rather than simple visibility. Interestingly, `fold` achieves the highest accuracy (80.43%) and lowest error ratio (0.62). Despite the physical complexity of cloth manipulation, the visual delta between an unfolded cloth and a folded one is drastic and visually distinct. This suggests that for State Prediction, high visual contrast in the outcome can compensate for complex underlying physics, whereas actions with subtler geometric changes like `flip` remain prone to reasoning errors.

| Action | Performance Metrics | | | | | Normalized Error Distribution | | |
| --- | --- | --- | --- | --- | --- | --- | --- | --- |
| | Total | Correct | Errors | Acc. (%) | Err. Rate (%) | Freq. (%) | Err. Contrib. (%) | Ratio |
| put | 977 | 681 | 296 | 69.70 | 30.30 | 44.98 | 42.96 | 0.96 |
| pick | 458 | 314 | 144 | 68.56 | 31.44 | 21.09 | 20.90 | 0.99 |
| place | 361 | 256 | 105 | 70.91 | 29.09 | 16.62 | 15.24 | 0.92 |
| move | 200 | 127 | 73 | 63.50 | 36.50 | 9.21 | 10.60 | 1.15 |
| remove | 72 | 40 | 32 | 55.56 | 44.44 | 3.31 | 4.64 | 1.40 |
| fold | 46 | 37 | 9 | 80.43 | 19.57 | 2.12 | 1.31 | 0.62 |
| take | 43 | 21 | 22 | 48.84 | 51.16 | 1.98 | 3.19 | 1.61 |
| flip | 15 | 7 | 8 | 46.67 | 53.33 | 0.69 | 1.16 | 1.68 |
| **Overall** | **2172** | **1483** | **689** | **68.28** | **31.72** | **100.00** | **100.00** | **-** |

Table 24: Model performance on high-level state prediction tasks broken down by action type. **Left:** Raw accuracy metrics. **Right:** Normalized error analysis, where a Ratio > 1 indicates the action contributes disproportionately to the total error count.

### C.6.2 Mid-Level State Prediction

To better understand the reasoning failures underlying the high-level errors discussed above, we further examine the model's performance on Mid-Level State Prediction. Table 25 summarizes the error rates across the three primary axes of manipulation: Translation, Rotation, and Gripper actuation.

At the aggregate level, the data corroborates the existence of a rotational bottleneck in predictive reasoning. The Rotation category exhibits the highest overall error rate at 58.01%, compared to 54.97% for Translation. This empirically supports that VLMs struggle more to simulate changes in orientation, which require an understanding of 3D object geometry, than changes in position. Notably, Roll+ (clockwise rotation around the X-axis) presents the highest difficulty among rotational primitives (64.34%). We attribute this to the fact that roll is often the least visually salient rotation in 2D images, frequently manifesting as subtle shape distortions rather than obvious displacements.

Within the Translation category, a significant anomaly appears in the vertical axis. While upward movement (Z+) has an error rate of 54.01%, downward movement (Z-) spikes to 68.29%. This asymmetry suggests a deficiency in depth perception and contact physics. Specifically, due to the nature of robotic manipulation tasks, moving down often involves interactions with the table or objects causing occlusion and collision, whereas moving up into free space is visually simpler. The model appears to struggle specifically with predicting the visual consequences of the gripper approaching the background.

Finally, Gripper errors are surprisingly high (56.22%), surpassing the Translation average. Specifically, the model struggles more to predict the result of `Open` (59.52%) than `Close` (54.07%). This may stem from the inherent visual ambiguity of opening a gripper. If the gripper is empty, the visual change is minimal, whereas closing often results in a distinct grasp state. This indicates that the model may lack the fine-grained spatial resolution required to reliably predict the articulation of small robot components.

| Category | Specific Action | Occurences | Errors | Error Rate (%) |
|---|---|---|---|---|
| **Translation** | Move X+ | 55 | 27 | 49.09 |
| | Move X- | 80 | 45 | 56.25 |
| | Move Y+ | 157 | 86 | 54.78 |
| | Move Y- | 125 | 69 | 55.20 |
| | Move Z+ | 337 | 182 | 54.01 |
| | Move Z- | 41 | 28 | **68.29** |
| | Category Avg. | 795 | 437 | 54.97 |
| **Rotation** | Roll+ | 143 | 92 | **64.34** |
| | Roll- | 42 | 23 | 54.76 |
| | Pitch+ | 115 | 68 | 59.13 |
| | Pitch- | 79 | 33 | 41.77 |
| | Yaw+ | 225 | 128 | 56.89 |
| | Yaw- | 189 | 116 | 61.38 |
| | Category Avg. | 793 | 460 | 58.01 |
| **Gripper** | Open | 289 | 172 | 59.52 |
| | Close | 442 | 239 | 54.07 |
| | Category Avg. | 731 | 411 | 56.22 |

Table 25: Detailed error analysis of Mid-Level State Prediction tasks. The data reveals a performance hierarchy where rotational dynamics and depth-related translations (Z-axis) pose the greatest challenges.

### C.6.3 Low-Level State Prediction

To investigate the Low-Level State Prediction task to determine whether errors stem from the complexity of the motion (large changes are harder to simulate) or the subtlety of the visual change (small changes are harder to perceive), we binned the samples into magnitude quartiles (Q1–Q4) for Translation, Rotation, and Gripper actuation to observe performance trends across the full spectrum of physical action.

As shown in Table 26, the relationship between action magnitude and accuracy varies fundamentally by action type. Translation accuracy appears to scale linearly with motion magnitude, rising steadily from 38.6% in Q1 to 56.5% in Q4. This monotonic improvement suggests that VLMs are more adept at predicting the outcome when the translation is large and visually distinct.

In stark contrast, Rotation exhibits a non-linear trend that seems to be consistent with the Rotational Bottleneck we identified in our earlier analysis. Specifically, performance peaks at moderate angles at Q2 where the motion is visible but not geometrically complex. However, for large rotations within the range of Q4, performance collapses to 37.1%, a sharp drop of 15.4% relative to the baseline. We hypothesize that large rotations introduce pronounced perspective shifts and self-occlusions, which in turn impair the model's ability to preserve a coherent internal representation of the environment's 3D geometry.

Finally, Gripper actuation shows negligible variance across quartiles, implying that VLMs treat gripper actions as binary semantic events (Open/Close) rather than continuous kinematic changes, rendering the specific magnitude of the aperture change largely irrelevant to the prediction process.

| Magnitude Type | Quartile | Range | Acc. | Δ |
|---|---|---|---|---|
| Translation (m) | Q1 | 0.001–0.093 | 38.6% | – |
| | Q2 | 0.093–0.123 | 50.0% | +11.4 % |
| | Q3 | 0.123–0.168 | 55.9% | +17.4 % |
| | Q4 | 0.168–0.416 | **56.5%** | **+18.0 %** |
| Rotation (°) | Q1 | 2.2–16.0 | 52.5% | – |
| | Q2 | 16.0–20.5 | **57.6%** | +5.1 % |
| | Q3 | 20.5–25.6 | 53.8% | +1.3 % |
| | Q4 | 25.6–123.6 | 37.1% | **-15.4 %** |
| Gripper | Q1 | 0.000–0.201 | 51.3% | – |
| | Q2 | 0.201–0.374 | 48.7% | -2.5 % |
| | Q3 | 0.374–0.597 | 50.4% | -0.8 % |
| | Q4 | 0.597–0.991 | 50.6% | -0.6 % |

Table 26: Model performance on Low-Level State Prediction broken down by action magnitude quartiles across the combined dataset (BridgeData V2 + DROID). Q1 represents the smallest movements, Q4 the largest. Δ indicates the performance change relative to the Q1 baseline.

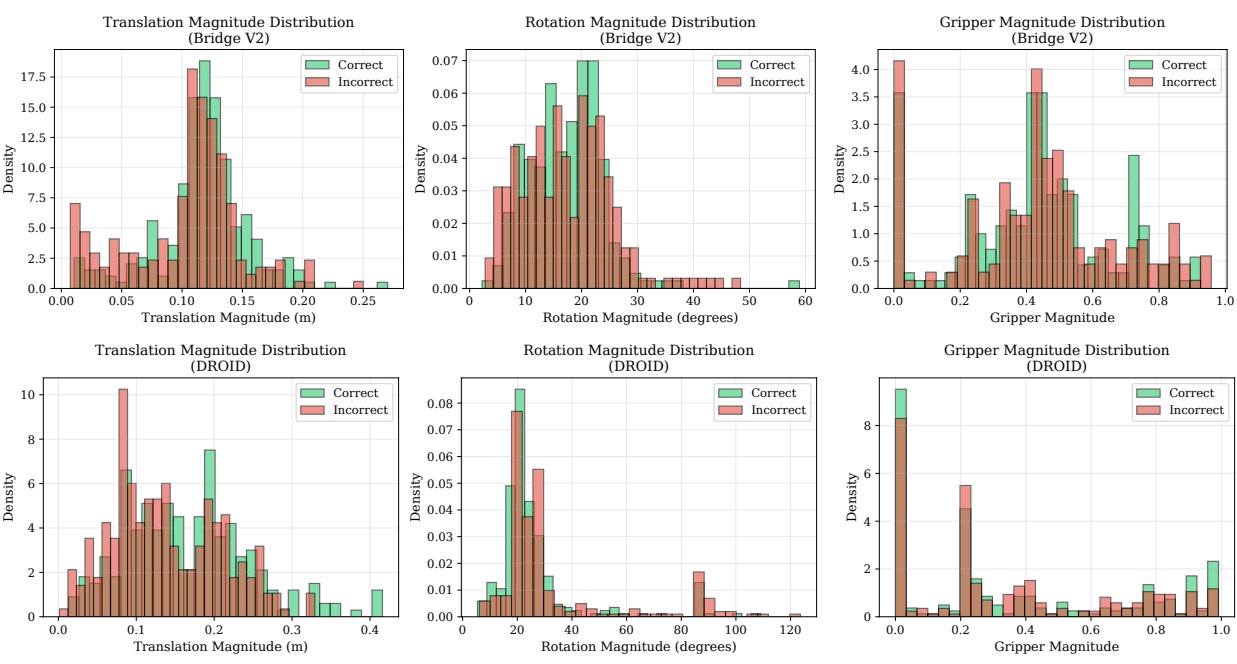

Figure 21: Distribution of correct vs. incorrect predictions on BridgeData V2 (top) and DROID (bottom).

## C.7 Analysis of Data Contamination and Systemic Failure Modes

To address the possibility that our findings are circumstantial or primarily driven by data contamination, we provide a detailed analysis of the relationship between the raw source data and our diagnostic framework. ACTIONEQA introduces a novel task layer that is fundamentally distinct from the original data distributions. The source datasets primarily consist of high-level natural language goal descriptions paired with raw trajectory data. However, ACTIONEQA requires models to map these images to a customized 14-label mid-level taxonomy and perform bidirectional reasoning tasks that were never part of the original training objectives or ground-truth labels.

As evidenced in Table 1 and Appendix C.11, the statistical consistency of the mid-level bottleneck across 34 diverse architectures provides strong evidence that the observed failures are systemic rather than a result of data familiarity. If the models were simply leveraging memorized associations from their training data, we would expect performance to remain relatively high across all tiers derived from that same data. Instead, we observe a consistent performance drop at the mid-level across all model families, including proprietary models such as Gemini-2.5-Pro and GPT-4.1. For instance, Gemini-2.5-Pro achieves 70.4% accuracy on high-level tasks but drops to 43.4% at the mid-level. This sharp decline signifies a collapse in the model's ability to ground compositional language into 3D transformations, a reasoning skill that remains deficient regardless of the model's prior exposure to the raw images.

These failure modes are likely rooted in current VLM architectures, which typically utilize 2D Vision Transformer backbones (Dosovitskiy, 2020). These backbones process images as flattened patches and lack explicit inductive biases for 3D coordinate transformations or perspective (Yang et al., 2025b). This architectural limitation explains why models struggle significantly more with mid-level rotational commands, which require an understanding of an axis of rotation and a reference frame, than with high-level semantic identification. Furthermore, as shown in Table 3 and Table 4, the clash of perspectives observed during multi-view fusion suggests that models lack a principled mechanism for spatial cross-view reconciliation. Rather than building a unified 3D representation, the models appear to struggle with the increased cognitive load of reconciling multiple 2D perspectives, a problem that is particularly pronounced in the State Prediction task. Consequently, ACTIONEQA serves as a fine-grained diagnostic tool that exposes a core deficit in spatial and physical reasoning that cannot be solved by simply increasing exposure to raw robotic data.

### C.8 Additional Analysis

**Analysis of Individual Model Strengths.** A closer examination of Table 1 reveals that different models exhibit distinct strengths and weaknesses across the six reasoning tasks. Among proprietary models, Gemini-2.5-Pro demonstrates the most consistent state-of-the-art performance, achieving the highest scores across the majority of categories. GPT-4.1 also shows robust capabilities, particularly on the BridgeData V2 dataset where it scores highest overall (51.2%), excelling in high-level state prediction (H-SP: 80.4%) and mid-level state prediction (M-SP: 48.4%). Interestingly, Claude-Opus-4 displays a spiky performance profile, showing exceptional strength in Low-Level Action Inference (L-AI) with scores of 71.9% on DROID and 74.2% on BridgeData V2, significantly outperforming most other models on this specific task, while being less competitive in other areas. Among open-weight models, GLM-4.5V stands out as the top performer, showing strong and balanced capabilities, especially on the BridgeData V2 (49.6% avg.) and RT-1 (68.0% avg.) datasets. The second-best model, Ovis2.5-9B, showcases a particular aptitude for high-level reasoning, securing the top score among open models in High-Level Action Inference (H-AI) on the DROID dataset (61.5%). In contrast, Llama-4-Maverick demonstrates a unique specialization in low-level reasoning, achieving the highest scores for Low-Level Action Inference (L-AI) on both the DROID (60.3%) and BridgeData V2 (61.8%) datasets. This specialization suggests that its architecture or training data may be particularly well-suited for translating visual changes into precise, low-level motor command reconstructions.

**Model Scalability Insights.** Figure 22 and Table 1 offer clear insights into the impact of model scaling. For most model families, larger parameter counts generally correlate with improved performance, though with diminishing returns. The Gemma-3 series shows a consistent upward trend in overall performance from the 4B (32.5%) to the 12B (38.2%) and 27B (40.5%) versions. Similarly, the Qwen2.5-VL family demonstrates performance gains with scale, moving from 30.7% (3B) to 38.9% (72B). Notably, the largest model in this series, Qwen2.5-VL-72B-Ins, achieves a remarkable score of 76.0% in High-Level Action Inference on the RT-1 dataset, the highest among all open-weight models, suggesting that scale can unlock specific, complex reasoning capabilities. The trend is even more pronounced among the proprietary models. The progression from Gemini-2.5-Flash-Lite (43.2%) to Gemini-2.5-Flash (46.9%) and finally to Gemini-2.5-Pro (58.4%) illustrates a significant performance leap tied to model capability. This suggests that while scaling is not a panacea, it remains a critical factor in advancing VLMs' ability to bridge the semantic-to-physical gap, particularly for the most challenging reasoning tasks evaluated in ACTIONEQA.

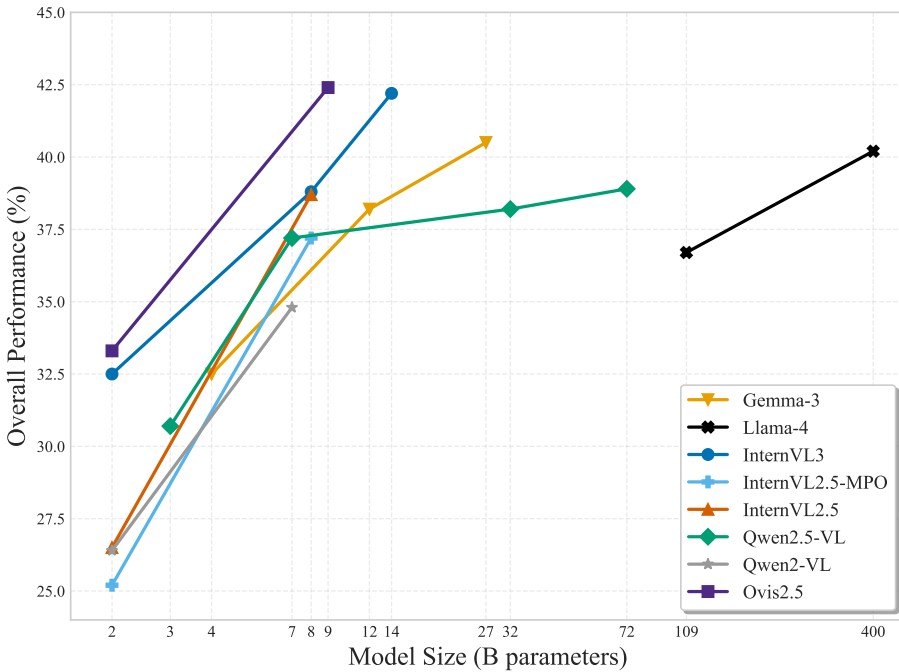

Figure 22: Overall performance scaling with model size for selected open-weight models

**Performance Consistency Across Datasets.** Models exhibit varied levels of generalization across the three distinct real-world datasets. Some models, like GLM-4.5V, demonstrate high consistency, achieving top-tier open-weights performance on the DROID (39.8% avg.), BridgeData V2 (49.6% avg.), and RT-1 (68.0% avg.) datasets. This indicates a strong ability to generalize its action understanding to different environments, camera setups, and manipulation tasks. In contrast, other models show dataset-specific strengths. For instance, InternVL3-14B performs well on BridgeData V2 (44.6% avg.) and RT-1 (61.1% avg.) but is comparatively weaker on DROID (38.9% avg.). Conversely, Ovis2.5-9B excels on DROID (41.2% avg.) but is less competitive on BridgeData V2 (42.0% avg.). These variations underscore the diversity of challenges presented by each dataset and highlight that strong performance on one domain does not automatically translate to others.

**The Asymmetry of Prediction vs. Inference.** As noted in the main paper, our benchmark reveals a fundamental asymmetry where models consistently find Action Inference (backward-looking) easier than State Prediction (forward-looking), which is also evidenced by the trend seen in Figure 23. The detailed results in Table 1 provide granular evidence for this phenomenon. For the top-performing model, Gemini-2.5-Pro, the average score across all Action Inference tasks is 63.7%, whereas its average for State Prediction tasks is only 53.2%—a significant gap of 10.4 percentage points. This pattern holds for the leading open-weight model, GLM-4.5V, which scores 47.1% on inference tasks versus 44.2% on prediction tasks. This persistent gap across diverse models suggests that while current VLMs are adept at rationalizing an observed outcome by identifying a plausible cause, they are significantly weaker at predictive physical reasoning—that is, simulating the future outcome of a given action. This points to a core limitation in their internal models of physics and causality.

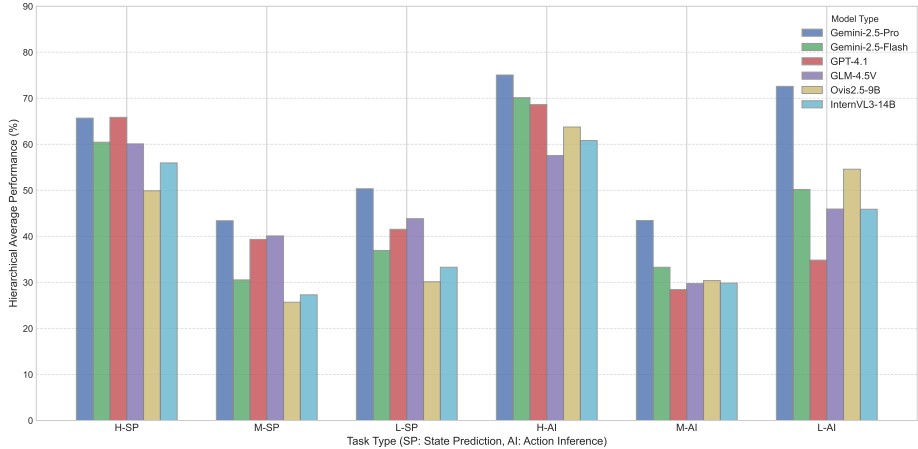

Figure 23: Performance Comparison of Top VLMs Across Six Tasks (Averaged Over All Datasets)

### C.9 Ablation Study on the Impacts of Multiple Exocentric Views

**Rationale.** The primary goal of this ablation study is to understand how Vision-Language Models (VLMs) leverage increased visual information from multiple viewpoints. By systematically increasing the number of exocentric views, we can assess whether richer perceptual data leads to improved reasoning and task performance, or if it introduces a "reasoning burden" that may hinder the model's capabilities. This analysis is crucial for understanding the trade-offs between perceptual richness and computational and cognitive load in embodied AI systems. The findings from this study provide insights into the optimal sensory configurations for different tasks and model architectures.

**Detailed Experimental Setup.** To investigate the influence of perceptual richness on model performance, we conducted an ablation study varying the number of exocentric camera views provided as input. This study was performed using the BridgeData V2 dataset, which is uniquely suited for this analysis due to the availability of multiple concurrent, non-egocentric viewpoints. We systematically evaluated model performance across four distinct configurations: single-view, two-view, three-view, and four-view inputs. For each configuration, the input images were standardized to maintain a consistent aspect ratio while increasing the total pixel information available to the model. The image resolutions were set to 256×256 pixels for the 1-view configuration, 512×256 pixels for the 2-view configuration, 768×256 pixels for the 3-view configuration, and 1024×256 pixels for the 4-view configuration, as shown in Figure 24.

Figure 24: A sample frame of BridgeData V2 dataset presented in 1 to 4 exocentric views used in the ablation study to examine the effects of richer visual information

**Methodology for Normalized Error Rate Proportions.** To accurately visualize the relative sources of model failure for Figure 6b and avoid misleading conclusions drawn from imbalanced action distributions in the dataset, we developed a robust methodology for calculating **Normalized Error Rate Proportions**. The process begins by categorizing all actions into three distinct types: Translation, Rotation, and Gripper. For each action type $\tau$, we first compute its **Normalized Error Rate**, calculated by dividing the total count of errors involving that action type ($E_\tau$) by the total number of questions designed to test it ($Q_\tau$). This normalization step is essential as it provides a per-opportunity failure rate, revealing the inherent difficulty the model has with each category, independent of how frequently that action appears. Subsequently, these individual normalized error rates are summed to establish a **Total Error Rate Magnitude**, a composite score representing the overall error landscape. The final **Normalized Error Rate Proportion** for each action is then determined by dividing its individual normalized rate by this total magnitude. This yields a set of proportions that sum to 100%, directly corresponding to the segments in our stacked bar charts. This approach ensures our analysis pinpoints the primary conceptual bottlenecks—for instance, whether flawed geometric reasoning in rotations contributes more to the overall error profile than failures in recognizing gripper state changes—rather than merely reflecting the underlying statistical distribution of actions in the evaluation data.

| View | Gemini-2.5-flash | | | GLM-4.5V | | |
|---|---|---|---|---|---|---|
| | Trans% | Rot% | Grip% | Trans% | Rot% | Grip% |
| 1 View | 32.01 | 42.10 | 25.89 | 36.70 | 37.42 | 25.89 |
| 2 Views | 34.34 | 41.33 | 24.33 | 37.26 | 36.69 | 26.05 |
| 3 Views | 35.70 | 36.63 | 27.67 | 37.72 | 39.81 | 22.47 |
| 4 Views | 33.41 | 38.69 | 27.90 | 39.35 | 40.00 | 20.65 |
| **AVERAGE** | **33.86** | **39.69** | **26.45** | **37.76** | **38.48** | **23.76** |

Table 27: Normalized Error Rate Proportions for gemini-2.5-flash and GLM-4.5V on the BridgeData V2 dataset

| View | Trans% | Rot% | Grip% |
|---|---|---|---|
| 1 View | 34.36 | 39.76 | 25.89 |
| 2 Views | 35.80 | 39.01 | 25.19 |
| 3 Views | 36.71 | 38.22 | 25.07 |
| 4 Views | 36.38 | 39.35 | 24.28 |
| **AVERAGE** | **35.81%** | **39.09%** | **25.11%** |

Table 28: Average Normalized Error Rate Proportions for Combined Models on the BridgeData V2 dataset

### C.10   Ablation Study on the Impacts of Different Camera Perspectives

**Rationale.**   To comprehensively understand the role of camera perspective in action understanding, we conducted an ablation study to investigate the efficacy of egocentric versus exocentric viewpoints. The egocentric (first-person) perspective offers a direct, action-oriented view from the agent's perspective, while the exocentric (third-person) perspective provides a broader situational context. A common intuition is that fusing these two viewpoints would provide a more complete and robust visual input, thereby enhancing a model's reasoning capabilities. However, as explored in our main findings, providing richer visual information does not always lead to better performance and can introduce a "reasoning burden." This study aims to directly test that hypothesis by evaluating whether current Vision-Language Models (VLMs) can effectively synergize information from distinct perspectives or if the fusion leads to a "clash of perspectives" that degrades performance.

**Detailed Experimental Setup.**   This analysis was performed exclusively on the DROID dataset, whose suitability for this study is unparalleled. It is the only dataset in our benchmark that provides a rich, multi-perspective view of each action, comprising one egocentric (first-person) camera and two distinct exocentric (third-person) cameras. This unique data composition allows for a rigorous and controlled comparison between a pure first-person perspective, a pure third-person perspective, and their direct fusion, enabling a precise evaluation of how VLMs handle different viewpoints. We evaluated VLMs across three distinct view configurations: the **Exocentric View Only** (the third-person, static scene view), the **Egocentric View Only** (the first-person view from the robot's gripper camera), and the **Combined View** (a concatenated input of both). To ensure a fair comparison and control for variations in input data size, all images were standardized. Single-view inputs (both egocentric and exocentric) were resized to $320 \times 180$ pixels. For the combined dual-view configuration, the two individual views were horizontally concatenated, forming a single $640 \times 180$ pixel input.

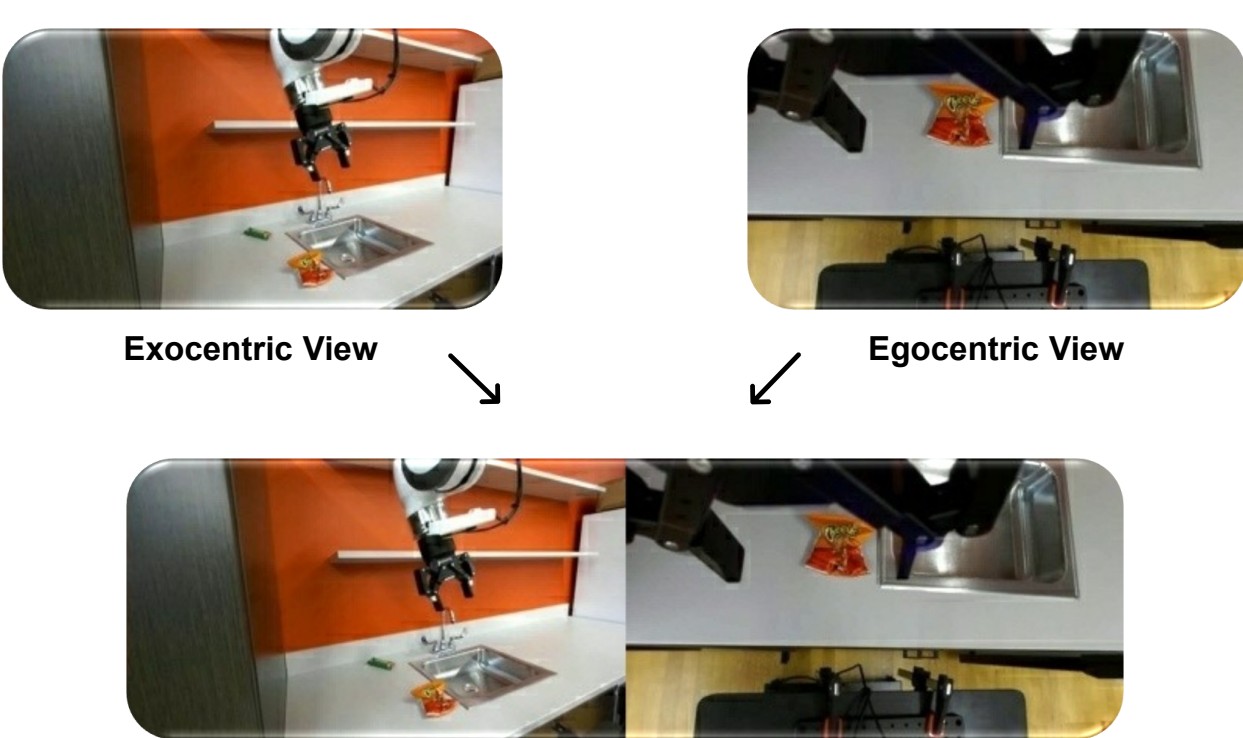

**Concatenated Exocentric & Egocentric Views**

Figure 25: A sample frame of the DROID dataset presented in exocentric, egocentric, and combined views used in the ablation study to examine the effects of different camera perspectives.

## C.11 Statistical Significance of the Mid-Level Performance Bottleneck

We investigated the hypothesis of a Mid-Level Performance Bottleneck, which posits that models perform significantly worse on mid-level action reasoning tasks compared to both high-level and low-level tasks. To ensure a controlled comparison across the hierarchy, we restricted our analysis to the DROID and BridgeData V2 datasets, as these provide tasks at all three levels. We excluded RT-1 to prevent dataset-specific difficulty distributions from confounding the hierarchical analysis.

For each of the $n = 34$ models, we calculated aggregate performance scores for High-Level ($X_{\mathrm{H},i}$), Mid-Level ($X_{\mathrm{M},i}$), and Low-Level ($X_{\mathrm{L},i}$) tasks by averaging the respective State Prediction and Action Inference scores. To statistically verify the V-shaped performance curve, we defined two difference variables for each model $i$: $D_{\mathrm{HM},i} = X_{\mathrm{H},i} - X_{\mathrm{M},i}$ and $D_{\mathrm{LM},i} = X_{\mathrm{L},i} - X_{\mathrm{M},i}$. We conducted two one-sided paired $t$-tests to determine if performance at the extremes (High and Low) is strictly superior to the Mid-Level, testing the null hypotheses $H_0 : \mu_D \leq 0$ against the alternatives $H_1 : \mu_D > 0$ at $\alpha = 0.05$.

The analysis, summarized in Table 29, strongly supports the bottleneck hypothesis. First, comparing High-Level to Mid-Level performance reveals a massive disparity. The mean difference was $\bar{D}_{\mathrm{HM}} = +18.93\%$, with a standard deviation of differences $S_{D_{\mathrm{HM}}} = 7.70$. The resulting $t$-statistic is:

$$t = \frac{\bar{D}_{\mathrm{HM}}}{S_{D_{\mathrm{HM}}}/\sqrt{n}} = \frac{18.93}{1.32} \approx 14.34 \tag{3}$$

With $df = 33$, this yields a $p$-value $< 0.001$. We further quantified this gap using Cohen's $d$, resulting in $d \approx 2.46$, an extremely large effect size indicating that models are drastically more proficient at high-level abstraction than mid-level reasoning.

Second, we examined the recovery in performance at the Low-Level. The mean difference between Low-Level and Mid-Level tasks was $\bar{D}_{\mathrm{LM}} = +8.66\%$, with a standard deviation of differences $S_{D_{\mathrm{LM}}} = 6.82$. The $t$-statistic was calculated as:

$$t = \frac{\bar{D}_{\mathrm{LM}}}{S_{D_{\mathrm{LM}}}/\sqrt{n}} = \frac{8.66}{1.17} \approx 7.40 \tag{4}$$

This also yields a $p$-value $< 0.001$. The corresponding effect size is $d \approx 1.27$, confirming that models handle primitive execution significantly better than intermediate translation.

Collectively, these tests allow us to reject the null hypotheses for both comparisons. The statistical evidence confirms the existence of a sharp, V-shaped performance bottleneck: current Vision-Language Models struggle most severely not with high-level planning or low-level actuation, but with the intermediate reasoning required to bridge the two.

| Statistic | High vs. Mid | Low vs. Mid |
|---|---|---|
| Mean Score (Comparator) | 47.82 (High) | 37.55 (Low) |
| Mean Score (Mid-Level) | 28.89 | 28.89 |
| Mean Difference ($\bar{D}$) | **+18.93** | **+8.66** |
| Std. Deviation of Diff. ($S_D$) | 7.70 | 6.82 |
| Standard Error ($SE_{\bar{D}}$) | 1.32 | 1.17 |
| $t$-**statistic** | **14.34** | **7.40** |
| $p$-**value** (one-tailed) | **< 0.001** | **< 0.001** |
| **Effect Size** (Cohen's $d$) | **2.46** | **1.27** |

Table 29: Detailed Statistical Analysis of the Mid-Level Bottleneck. Positive mean differences indicate superior performance over Mid-Level.

### C.12 Statistical Significance of Action Inference vs. State Prediction

To rigorously validate the claim that models perform significantly better on **Action Inference (AI)** tasks compared to **State Prediction (SP)** tasks, we conducted a one-sided paired sample $t$-test. This approach controls for inter-model variability by comparing the performance differences within each model across the two task types.

The sample consists of the $n = 34$ models evaluated in our main experiments. For each model $i$, we computed aggregate scores $X_{\mathrm{SP},i}$ and $X_{\mathrm{AI},i}$ by averaging accuracy across the seven applicable sub-tasks (High, Mid, and Low-Level for DROID and BridgeData V2; High-Level for RT-1). We define the performance difference for the $i$-th model as $D_i = X_{\mathrm{AI},i} - X_{\mathrm{SP},i}$. We tested the null hypothesis $H_0 : \mu_D \leq 0$ (models perform equal to or worse on AI) against the alternative hypothesis $H_1 : \mu_D > 0$ (models perform strictly better on AI) at a significance level of $\alpha = 0.05$.

The results of the analysis are summarized in Table 30. The observed mean difference between Action Inference and State Prediction was $\bar{D} = +4.31\%$, with a standard deviation of differences $S_D = 4.56$. The $t$-statistic was calculated as:

$$t = \frac{\bar{D}}{S_D/\sqrt{n}} = \frac{4.31}{0.782} \approx 5.51 \tag{5}$$

With $df = 33$ degrees of freedom, this yields a one-tailed $p$-value of $2.84 \times 10^{-6}$, which is well below the significance threshold ($p < 0.05$). Consequently, we reject the null hypothesis.

To quantify the magnitude of this result, we calculated Cohen's $d$ effect size:

$$d = \frac{\bar{D}}{S_D} = \frac{4.31}{4.56} \approx 0.95 \tag{6}$$

A Cohen's $d$ of 0.95 indicates a large effect size. This confirms that the performance gap is not only statistically significant but also substantial in magnitude, supporting the conclusion that VLMs demonstrate a robust capability bias toward inferring actions over predicting states.

| Statistic | Symbol | Value |
|---|---|---|
| Mean Score (State Prediction) | $\bar{X}_{\mathrm{SP}}$ | 40.51 |
| Mean Score (Action Inference) | $\bar{X}_{\mathrm{AI}}$ | 44.82 |
| Mean Difference | $\bar{D}$ | +4.31 |
| Standard Error of Mean | $SE_{\bar{D}}$ | 0.782 |
| $t$-**statistic** | $t$ | **5.51** |
| $p$-**value** (one-tailed) | $p$ | $< \mathbf{0.001}$ |
| **Effect Size** (Cohen's $d$) | $d$ | **0.95** |

Table 30: Paired Sample T-Test results comparing Action Inference (AI) and State Prediction (SP) performance.

### C.13 Statistical Significance of the Mid-Level Rotational Bottleneck

In Section 3, we identified a critical bottleneck at the mid-level of the action hierarchy, observing that rotational actions consistently yield higher error rates than translational or gripper actions. To ensure this finding is statistically robust and not an artifact of dataset distribution or variance in visual inputs, we conducted a formal significance test using the **Normalized Error Rate Proportions** presented in Figure 6b, which is derived from Table 28.

We focused our analysis on the Mid-Level Action Inference tasks within the BridgeData V2 dataset ($N = 890$, see Table 9). We define the null hypothesis $H_0$ as the scenario where rotational actions pose no greater challenge than translational actions (the second-highest error category). Consequently, the alternative hypothesis $H_1$ is that rotational actions are strictly more difficult.

We treat the model performance across the four distinct exocentric view configurations ($i \in \{1, 2, 3, 4\}$) as paired observations to control for perceptual variance. Let $R_i$ and $T_i$ represent the Normalized Error Rate Proportions for Rotation and Translation, respectively, for view configuration $i$. We define the difference $d_i$ as:

$$d_i = R_i - T_i \tag{7}$$

The hypotheses are formally stated as:

$$H_0 : \mu_d \leq 0$$
$$H_1 : \mu_d > 0$$

where $\mu_d$ is the true population mean of the differences. We omit a formal test against Gripper errors, as the descriptive gap ($R_{avg} - G_{avg} \approx 14\%$) is substantially larger than the Rotation-Translation gap, making the Rotation-Translation comparison the critical threshold for establishing primacy.

Using the data from Table 28, the observed differences for the four view configurations are:

$$D = \{5.40, 3.21, 1.51, 2.97\}$$

We calculate the sample mean difference ($\bar{d}$) and the standard deviation of the differences ($s_d$):

$$\bar{d} = \frac{1}{n} \sum_{i=1}^{n} d_i = 3.2725 \tag{8}$$

$$s_d = \sqrt{\frac{\sum_{i=1}^{n} (d_i - \bar{d})^2}{n - 1}} \approx 1.604 \tag{9}$$

The standard error ($SE$) and the resulting $t$-statistic are calculated as follows:

$$t = \frac{\bar{d}}{s_d / \sqrt{n}} = \frac{3.2725}{1.604 / \sqrt{4}} \approx 4.08 \tag{10}$$

With $n - 1 = 3$ degrees of freedom, a $t$-statistic of 4.08 corresponds to a one-tailed $p$-value of **0.013**. Since $p < 0.05$, we reject the null hypothesis.

To quantify the magnitude of this discrepancy, we calculated **Cohen's** $d$ for paired samples, yielding a value of **2.04**. This indicates a very large effect size, suggesting that the difficulty gap between rotational and translational reasoning is substantial and not merely a marginal statistical finding. Furthermore, the 95% confidence interval for the mean difference is $[0.72, 5.82]$, which does not span zero. This reinforces that the increased error rate for rotational actions is a systematic phenomenon across varying levels of perceptual information. The results are summarized in Table 31.

| Comparison Pair | Mean Diff. ($\bar{d}$) | $t$-stat | $p$-value | Effect Size ($d$) | 95% CI |
|---|---|---|---|---|---|
| Rotation vs. Translation | +3.27% | 4.08 | 0.013 | 2.04 | $[0.72, 5.82]$ |

Table 31: Summary of the Paired $t$-Test, Effect Size, and Confidence Interval validating that Rotational errors are significantly higher than Translational errors across all view configurations in BridgeData V2.

## C.14   Statistical Significance of Egocentric View Dominance

To provide a rigorous validation of the observations made in Section 3.4 regarding the dominance of the Egocentric view over the Exocentric view in single-view settings, we performed a statistical significance test on the aggregate performance of the evaluated models. While the main text breaks down performance by action hierarchy levels (High, Mid, Low), this analysis combines these levels to evaluate the overall impact of camera perspective on the two fundamental reasoning tasks: **State Prediction** and **Action Inference**.

We employ a one-tailed Two-Proportion Z-Test to determine if the accuracy of the Egocentric view $P_{ego}$ is statistically significantly greater than that of the Exocentric view $P_{exo}$. Our null hypothesis $H_0$ posits that $P_{ego} \leq P_{exo}$, while the alternative hypothesis $H_1$ asserts that $P_{ego} > P_{exo}$. To account for the varying number of questions across the hierarchy levels in the DROID dataset, we first compute the weighted average accuracy $\bar{P}$ for a specific task across all levels $l \in \{High, Mid, Low\}$. This is defined as:

$$\bar{P} = \frac{\sum_l (Acc_l \times N_l)}{\sum_l N_l} \tag{11}$$

where $Acc_l$ corresponds to the average model accuracy for level $l$ and $N_l$ represents the sample size for that level. The statistical significance is then assessed using the Z-statistic calculated from the pooled sample proportion $\hat{p}$:

$$Z = \frac{\bar{P}_{ego} - \bar{P}_{exo}}{\sqrt{\hat{p}(1 - \hat{p}) \left( \frac{1}{n_{ego}} + \frac{1}{n_{exo}} \right)}} \tag{12}$$

For the State Prediction task, aggregating the High, Mid, and Low levels results in a total sample size of $N = 1,371$ per viewpoint. The weighted average accuracy for the Egocentric view is calculated at $40.77\%$, compared to $36.67\%$ for the Exocentric view, resulting in a performance gap of $+4.10\%$. The Z-test yields a score of $Z = 2.20$, corresponding to a $p$-value of $0.0139$. Since $p < 0.05$, we reject the null hypothesis. This confirms that despite the noise introduced by lower-level reasoning tasks where the performance gap narrows, the Egocentric view provides a statistically significant advantage for predicting future states overall.

For the Action Inference task, aggregating all hierarchy levels results in a total sample size of $N = 1,582$ per viewpoint. Here, the Egocentric view achieves a weighted accuracy of $53.55\%$, while the Exocentric view achieves $45.11\%$, yielding a substantial gap of $+8.44\%$. The test produces a Z-score of $4.76$ and a $p$-value of $< 0.00001$. This result is highly significant, providing strong statistical evidence that the Egocentric perspective is dominant for inferring the causes of observed transitions.

As summarized in Table 32, these findings reinforce the conclusion that in single-view settings, the first-person perspective offers critical visual cues that are significantly more beneficial for both predictive and inferential action reasoning than a static third-person view.

| Task Setting | Total Sample ($N$) | Ego. Accuracy | Exo. Accuracy | Gap ($\Delta$) | Pooled $\hat{p}$ | Z-Score | $p$-value | Result |
|---|---|---|---|---|---|---|---|---|
| State Prediction | 1,371 | 40.77% | 36.67% | +4.10% | 0.387 | 2.20 | **0.0139** | Significant |
| Action Inference | 1,582 | 53.55% | 45.11% | +8.44% | 0.493 | 4.76 | **< 1e-5** | Highly Significant |

Table 32: Statistical Significance Test Results: Egocentric vs. Exocentric View Performance on DROID. The analysis aggregates High, Mid, and Low hierarchy levels to assess overall task dominance.

### C.15 Direct-Answer and Chain-Of-Thought Prompts

Transparency in prompting is crucial for the reproducibility of VLM evaluations. In ACTIONEQA, we employed two distinct prompting strategies tailored to our quantitative and qualitative analysis goals: Direct-Answer prompts for standardized performance measurement, and Chain-of-Thought prompts for in-depth error diagnosis.

**Direct-Answer (DA) Prompts.** For the primary quantitative evaluation presented in Section 3, we utilized Direct-Answer prompts. These zero-shot prompts provide the model with the necessary visual context (single or multiple images), define the specific task (State Prediction or Action Inference), and present the multiple-choice options. Crucially, these prompts include strict output formatting instructions, directing the model to generate *only* the single character corresponding to its chosen answer (e.g., "A"). This approach minimizes parsing ambiguity and ensures a standardized evaluation across all models. The exact DA prompts used for High-, Mid-, and Low-level tasks are illustrated in Figures 27, 28, and 29 for State Prediction, and Figures 33, 34, and 35 for Action Inference.

**Chain-of-Thought (CoT) Prompts.** To conduct the detailed qualitative error analysis and case studies discussed in Section 4, we employed Chain-of-Thought prompts. Unlike the DA prompts, these instruct the model to explicitly articulate its step-by-step reasoning process before providing a final answer. By forcing the model to "show its work," we can look inside the black box to determine whether an incorrect answer stemmed from a failure to perceive the scene correctly (Perceptual Grounding Error) or a failure in logic and physics (Reasoning Error). The CoT prompts used to elicit these internal reasoning traces are shown in Figures 30, 31, and 32 for State Prediction, and Figures 36, 37, and 38 for Action Inference.

Additionally, Figure 26 illustrates the prompt used with Qwen-QVQ-72B to generate plausible, context-aware high-level distractor actions during the benchmark curation process (Appendix B.4).

---

### ● ● ●    Prompt for Generating Distractor Options for High-Level QA

```
You are an AI assistant specializing in generating assessment materials for visual understanding.

You will be given:
1. STARTING_IMAGE: A visual representation of the initial scene.
2. ENDING_IMAGE: A visual representation of the scene after an action has occurred.
3. GROUND_TRUTH_OPTION: A concise description of the action that transformed the STARTING_IMAGE into the ENDING_IMAGE.

Your task:
Generate 3 plausible but incorrect multiple-choice distractor options describing the action performed.
These distractors should be similar in phrasing and detail to the GROUND_TRUTH_OPTION, but not accurately describe the
actual transformation between the images.

Guidelines for Distractors:
- Plausibility: Each distractor should describe an action that could reasonably happen with the given objects or scene.
- Relevance: Refer only to objects and elements actually present in the images.
- Subtlety: Distractors should be close enough to the ground truth to be confusing, but clearly incorrect when comparing
the two images.
- Variety of Error: Try to cover different types of mistakes, such as:
    - Incorrect Action: Correct object(s), but wrong action.
    - Incorrect Object(s): Correct action, but with the wrong object(s).
    - Incorrect Location or State Change: Plausible action, but does not match the specific change shown.
    - Plausible Alternative: An action that could have happened but didn't.
- Consistent Style: Match the complexity, verb tense, and detail level of the GROUND_TRUTH_OPTION.
- Focus on Change: Distractors should describe plausible actions, but should not describe the specific transformation from
STARTING_IMAGE to ENDING_IMAGE.

Avoid:
- Distractors that are obviously unrelated or nonsensical.
- Trivial negations (e.g., "do not put X in Y" if the ground truth is "put X in Y").
- Descriptions of actions that leave everything unchanged (unless as a plausible misdirection).

Output Instructions:
List the 3 distractor options, each on a new line. Do not include explanations or additional text.

Current Task Information:
- STARTING_IMAGE (initial state): See Image 1
- ENDING_IMAGE (goal state): See Image 2
- GROUND_TRUTH_OPTION: {gt_option}

Number of Distractors to Generate: 3

Generate the 3 distractor options:
```

Figure 26: Prompt for generating distractor options in high-level State Prediction and Action Inference tasks.

**Direct-Answer Prompt for High-Level State Prediction Questions**

You are analyzing a robotic manipulation task involving a robot arm interacting with objects.
The primary goal of your analysis is to determine the correct state of the robot's gripper and related objects after a given high-level (goal-oriented) action.
For each task, you will be provided with the following visual information:

- **Image 1 (Initial State)**: An image containing {num_views} camera viewpoint(s) that show the robot's state (e.g., arm pose, gripper status) and objects' positions *before* performing an action.
- **Images 2, 3, 4, and 5 (Candidate Resulting States)**: Four distinct candidate images. Each shows a possible state of the scene after the action. Each candidate image also contains {num_views} camera viewpoint(s) with the identical arrangement and properties as Image 1.

### Your Task:
You are provided with:
- **Image 1 (Initial State)**.
- A high-level action: {action}.
- **Images 2, 3, 4, and 5 (Candidate Resulting States)**.

Based on the the **Initial State (Image 1)**, your objective is to select the candidate image that most accurately depicts the scene resulting from the given goal-oriented high-level action.

### Options:
A) Image 2
B) Image 3
C) Image 4
D) Image 5

### CRITICAL OUTPUT FORMAT INSTRUCTIONS:
**Your response MUST be ONLY the single capital letter (A, B, C, or D) corresponding to your chosen answer.**
**ABSOLUTELY NO other text, reasoning, explanation, punctuation, or formatting should be included.**
**Generating anything other than the single correct letter (A, B, C, or D) will result in an incorrect evaluation.**
You are acting as a precision evaluation component.

### Final Answer:
(Output only the single letter A, B, C, or D below)

Figure 27: Direct-Answer prompt for answering high-level State Prediction tasks

---

● ● ●          **Direct-Answer Prompt for Mid-Level State Prediction Questions**

You are analyzing a robotic manipulation task involving a robot arm interacting with objects.
The primary goal of your analysis is to determine the correct final pose of the robot's gripper after a given action.
For each task, you will be provided with the following visual information:

- **Image 1 (Initial State)**: An image containing {num_views} camera viewpoint(s) that show the robot's state (e.g., arm pose, gripper status) and objects' positions *before* performing an action.
- **Images 2, 3, 4, and 5 (Candidate Resulting States)**: Four distinct candidate images. Each shows a possible state of the scene after the action. Each candidate image also contains {num_views} camera viewpoint(s) with the identical arrangement and properties as Image 1.

### Reference Frame:
The coordinate system is **always defined relative to the base of the robot arm**.
This frame applies consistently, even if the robot's base itself is not fully visible in Images 1-5.

To help visualize these axes, imagine yourself standing at the robot's base, oriented to look in its primary "forward" direction:
- **X-axis**: Represents this "forward" direction from the base (positive: forward, negative: backward).
- **Y-axis**: Extends to your **left** from this perspective (positive: left, negative: right).
- **Z-axis**: Points **upward** from the base where you are positioned (positive: up, negative: down).

### Interpreting Gripper Actions:
The state of the gripper is defined by its "proportional aperture," a value from 0.0 (fully closed) to 1.0 (fully open).
Action Type ("Open" vs. "Close"): The terms describe the direction of change between the initial state and the resulting state.
Action Value (Extent of Change): The value represents the magnitude of the change. It is calculated as the absolute difference between the starting and ending proportional apertures.
Let P1 be the proportional aperture in the initial state and P2 be the proportional aperture in the resulting state. The value is calculated with the formula:
value = absolute_value(P2 - P1)
- **Close gripper** means the gripper jaws are **more closed** in the resulting state than in the initial state.
- **Open gripper** means the gripper jaws are **more open** in the resulting state than in the initial state.

### Your Task:
You are provided with:
- **Image 1 (Initial State)**.
- A mid-level action: '{action}'.
- **Images 2, 3, 4, and 5 (Candidate Resulting States)**.

First try to infer the position of the base of the robot arm based on position of robot arm in Image 1, and understand the reference frame.
Your objective is to select the candidate image that most accurately depicts the scene resulting *strictly* from the given action, adhering to the considerations below.

**Important Considerations for Selection:**

1.  **Accurate Execution of Action & Gripper's Geometric Pose:**
    * The resulting scene must strictly and exclusively result from the successful execution of the specified action: '{action}'.
    * The robot gripper must achieve the appropriate final 3D geometric pose (position and orientation) as dictated by the action.
    * The change in the robot's pose must be the minimal and most direct transformation required by the command.
2.  **Overall Scene Consistency and Handling Imperfections:**
    * Choose the image that presents the most coherent and logically consistent result of performing the action.
    * If no candidate is flawless, prefer the one that best minimizes both errors in execution and any unjustified modifications to the environment.

Based on the **Initial State (Image 1)**, the given **mid-level action** '{action}', and the **selection considerations above**, choose the candidate image (from Images 2-5) that best represents the resulting state.

### Options:
A) Image 2, B) Image 3, C) Image 4, D) Image 5

### CRITICAL OUTPUT FORMAT INSTRUCTIONS:
**Your response MUST be ONLY the single capital letter (A, B, C, or D) corresponding to your chosen answer.**
**ABSOLUTELY NO other text, reasoning, explanation, punctuation, or formatting should be included.**
**Generating anything other than the single correct letter (A, B, C, or D) will result in an incorrect evaluation.**
You are acting as a precision evaluation component.

### Final Answer:
(Output only the single letter A, B, C, or D below)

Figure 28: Direct-Answer prompt for answering mid-level State Prediction questions

---

**● ● ●**        **Direct-Answer Prompt for Low-Level State Prediction Questions**

You are analyzing a robotic manipulation task involving a robot arm interacting with objects.
The primary goal of your analysis is to determine the correct final pose of the robot's gripper after a given action.
For each task, you will be provided with the following visual information:
- **Image 1 (Initial State)**: An image containing {num_views} camera viewpoint(s) that show the robot's state (e.g., arm pose, gripper status) and objects' positions *before* performing an action.
- **Images 2, 3, 4, and 5 (Candidate Resulting States)**: Four distinct candidate images. Each shows a possible state of the scene after the action. Each candidate image also contains {num_views} camera viewpoint(s) with the identical arrangement and properties as Image 1.

### Reference Frame:
The coordinate system is **always defined relative to the base of the robot arm**.
This frame applies consistently, even if the robot's base itself is not fully visible in Images 1-5.
## Linear Axes
To help visualize these axes, imagine yourself standing at the robot's base, oriented to look in its primary "forward" direction:
* **X-axis**: Represents the "forward" direction from the base (positive: forward, negative: backward).
* **Y-axis**: Extends to your left from this perspective (positive: left, negative: right).
* **Z-axis**: Points upward from the base (positive: up, negative: down).
## Rotations
The direction of rotation is defined from the perspective of looking **from the origin outwards** along the positive axis. All positive rotations follow the **right-hand rule**.
* **Roll (around +X axis):** Positive roll is a **clockwise** rotation, causing the top of the gripper to move to the right.
* **Pitch (around +Y axis):** Positive pitch is a **clockwise** rotation, causing the front of the gripper to move down.
* **Yaw (around +Z axis):** Positive yaw is a **counter-clockwise** rotation, causing the front of the gripper to move to the left.

### Your Task:
You are provided with:
- **Image 1 (Initial State)**.
- A low-level action: {action}.
- **Images 2, 3, 4, and 5 (Candidate Resulting States)**.

### Low-Level Action Format:
The low-level action you need to identify is a 7-element numerical vector. The options provided (A, B, C, D) will be such vectors. The format is: [delta_X, delta_Y, delta_Z, delta_roll, delta_pitch, delta_yaw, delta_gripper_state]
Where:
- `delta_X, delta_Y, delta_Z`: Represent the translational change of the robot's gripper endpoint in **meters** along the X, Y, and Z axes of the **base frame**, respectively.
- `delta_roll, delta_pitch, delta_yaw`: Represent the rotational change of the robot's gripper endpoint in **degrees** around the **base frame's** X-axis (roll), then Y-axis (pitch), then Z-axis (yaw) respectively.
- `delta_gripper_state`: This 7th element of the action vector specifies the **commanded change** to be applied to the gripper's current openness state. Gripper openness is represented by a normalized value, where **1.0 means fully open** and **0.0 means fully closed**.
    - A positive `delta_gripper_state` value from the action vector commands the gripper to become more **open** (i.e., its openness value moves towards 1.0).
    - A negative `delta_gripper_state` value commands the gripper to become more **closed** (i.e., its openness value moves towards 0.0).
    - A `delta_gripper_state` value of zero commands no change to the current gripper openness.

Your objective is to select the candidate image that most accurately depicts the scene resulting *strictly* from the given action, adhering to the considerations below.
**Important Considerations for Selection:**
1.  **Accurate Execution of Action & Gripper's Geometric Pose:**
    * The resulting scene must strictly and exclusively result from the successful execution of the specified action: {action}.
    * The robot gripper must achieve the appropriate final 3D geometric pose (position and orientation) as dictated by the action.
    * The change in the robot's pose must be the minimal and most direct transformation required by the command.
2.  **Overall Scene Consistency and Handling Imperfections:**
    * Choose the image that presents the most coherent and logically consistent result of performing the action.
    * If no candidate is flawless, prefer the one that best minimizes both errors in execution and any unjustified modifications to the environment.

Based on the **Initial State (Image 1)**, the given **low-level action** {action}, and the **selection considerations above**, choose the candidate image (from Images 2-5) that best represents the resulting state.
### Options:
A) Image 2, B) Image 3, C) Image 4, D) Image 5

### CRITICAL OUTPUT FORMAT INSTRUCTIONS:
**Your response MUST be ONLY the single capital letter (A, B, C, or D) corresponding to your chosen answer.**
**ABSOLUTELY NO other text, reasoning, explanation, punctuation, or formatting should be included.**
**Generating anything other than the single correct letter (A, B, C, or D) will result in an incorrect evaluation.**
You are acting as a precision evaluation component.

### Final Answer:
(Output only the single letter A, B, C, or D below)

Figure 29: Direct-Answer prompt for answering low-level State Prediction questions

● ● ●        **Chain-of-Thought Prompt for High-Level State Prediction Questions**

```
You are analyzing a robotic manipulation task involving a robot arm interacting with objects.
The primary goal of your analysis is to determine the correct state of the robot's gripper and related objects after a
given high-level (goal-oriented) action.
For each task, you will be provided with the following visual information:

- **Image 1 (Initial State)**: An image containing {num_views} camera viewpoint(s) that show the robot's state (e.g., arm
pose, gripper status) and objects' positions *before* performing an action.
- **Images 2, 3, 4, and 5 (Candidate Resulting States)**: Four distinct candidate images. Each shows a possible state of
the scene after the action. Each candidate image also contains {num_views} camera viewpoint(s) with the identical
arrangement and properties as Image 1.

### Your Task:
You are provided with:
- **Image 1 (Initial State)**.
- A high-level action: {action}.
- **Images 2, 3, 4, and 5 (Candidate Resulting States)**.

Based on the the **Initial State (Image 1)**, your objective is to select the candidate image that most accurately depicts
the scene resulting from the given goal-oriented high-level action.

### Options:
A) Image 2
B) Image 3
C) Image 4
D) Image 5

### CRITICAL OUTPUT FORMAT INSTRUCTIONS:
Please do step by step reasoning first, then give your final answer. For example, if you think the correct answer is 'A',
your response should be this format: '<think>(replace with your reasoning here)</think><answer>A</answer>'.

**You must choose the best possible option from A, B, C, or D. If none are perfectly correct, select the most likely or
closest answer.**

### Final Answer:
```

Figure 30: Chain-of-Thought prompt for answering high-level State Prediction questions

---

**Chain-of-Thought Prompt for Mid-Level State Prediction Questions**

```
You are analyzing a robotic manipulation task involving a robot arm interacting with objects.
The primary goal of your analysis is to determine the correct final pose of the robot's gripper after a given action.
For each task, you will be provided with the following visual information:

- **Image 1 (Initial State)**: An image containing {num_views} camera viewpoint(s) that show the robot's state (e.g., arm
pose, gripper status) and objects' positions *before* performing an action.
- **Images 2, 3, 4, and 5 (Candidate Resulting States)**: Four distinct candidate images. Each shows a possible state of
the scene after the action. Each candidate image also contains {num_views} camera viewpoint(s) with the identical
arrangement and properties as Image 1.

### Reference Frame:
The coordinate system is **always defined relative to the base of the robot arm**.
This frame applies consistently, even if the robot's base itself is not fully visible in Images 1-5.

To help visualize these axes, imagine yourself standing at the robot's base, oriented to look in its primary "forward"
direction:
- **X-axis**: Represents this "forward" direction from the base (positive: forward, negative: backward).
- **Y-axis**: Extends to your **left** from this perspective (positive: left, negative: right).
- **Z-axis**: Points **upward** from the base where you are positioned (positive: up, negative: down).

### Interpreting Gripper Actions:
The state of the gripper is defined by its "proportional aperture," a value from 0.0 (fully closed) to 1.0 (fully open).
Action Type ("Open" vs. "Close"): The terms describe the direction of change between the initial state and the resulting
state.
Action Value (Extent of Change): The value represents the magnitude of the change. It is calculated as the absolute
difference between the starting and ending proportional apertures.
Let P1 be the proportional aperture in the initial state and P2 be the proportional aperture in the resulting state. The
value is calculated with the formula:
value = absolute_value(P2 - P1)
- **Close gripper** means the gripper jaws are **more closed** in the resulting state than in the initial state.
- **Open gripper** means the gripper jaws are **more open** in the resulting state than in the initial state.

### Your Task:
You are provided with:
- **Image 1 (Initial State)**.
- A mid-level action: '{action}'.
- **Images 2, 3, 4, and 5 (Candidate Resulting States)**.

First try to infer the position of the base of the robot arm based on position of robot arm in Image 1, and understand the
reference frame.
Your objective is to select the candidate image that most accurately depicts the scene resulting *strictly* from the given
action, adhering to the considerations below.

**Important Considerations for Selection:**

1.  **Accurate Execution of Action & Gripper's Geometric Pose:**
    * The resulting scene must strictly and exclusively result from the successful execution of the specified action:
'{action}'.
    * The robot gripper must achieve the appropriate final 3D geometric pose (position and orientation) as dictated by the
action.
    * The change in the robot's pose must be the minimal and most direct transformation required by the command.
2.  **Overall Scene Consistency and Handling Imperfections:**
    * Choose the image that presents the most coherent and logically consistent result of performing the action.
    * If no candidate is flawless, prefer the one that best minimizes both errors in execution and any unjustified
modifications to the environment.

Based on the **Initial State (Image 1)**, the given **mid-level action** '{action}', and the **selection considerations
above**, choose the candidate image (from Images 2-5) that best represents the resulting state.

### Options:
A) Image 2, B) Image 3, C) Image 4, D) Image 5

### CRITICAL OUTPUT FORMAT INSTRUCTIONS:
Please do step by step reasoning first, then give your final answer. For example, if you think the correct answer is 'A',
your response should be this format: '<think>(replace with your reasoning here)</think><answer>A</answer>'.

**You must choose the best possible option from A, B, C, or D. If none are perfectly correct, select the most likely or
closest answer.**

### Final Answer:
```

Figure 31: Chain-of-Thought prompt for answering mid-level State Prediction questions

**Chain-of-Thought Prompt for Low-Level State Prediction Questions**

You are analyzing a robotic manipulation task involving a robot arm interacting with objects.
The primary goal of your analysis is to determine the correct final pose of the robot's gripper after a given action.
For each task, you will be provided with the following visual information:
- **Image 1 (Initial State)**: An image containing {num_views} camera viewpoint(s) that show the robot's state (e.g., arm pose, gripper status) and objects' positions *before* performing an action.
- **Images 2, 3, 4, and 5 (Candidate Resulting States)**: Four distinct candidate images. Each shows a possible state of the scene after the action. Each candidate image also contains {num_views} camera viewpoint(s) with the identical arrangement and properties as Image 1.

### Reference Frame:
The coordinate system is **always defined relative to the base of the robot arm**.
This frame applies consistently, even if the robot's base itself is not fully visible in Images 1-5.
## Linear Axes
To help visualize these axes, imagine yourself standing at the robot's base, oriented to look in its primary "forward" direction:
* **X-axis**: Represents the "forward" direction from the base (positive: forward, negative: backward).
* **Y-axis**: Extends to your left from this perspective (positive: left, negative: right).
* **Z-axis**: Points upward from the base (positive: up, negative: down).
## Rotations
The direction of rotation is defined from the perspective of looking **from the origin outwards** along the positive axis. All positive rotations follow the **right-hand rule**.
* **Roll (around +X axis):** Positive roll is a **clockwise** rotation, causing the top of the gripper to move to the right.
* **Pitch (around +Y axis):** Positive pitch is a **clockwise** rotation, causing the front of the gripper to move down.
* **Yaw (around +Z axis):** Positive yaw is a **counter-clockwise** rotation, causing the front of the gripper to move to the left.

### Your Task:
You are provided with:
- **Image 1 (Initial State)**.
- A low-level action: {action}.
- **Images 2, 3, 4, and 5 (Candidate Resulting States)**.
### Low-Level Action Format:
The low-level action you need to identify is a 7-element numerical vector. The options provided (A, B, C, D) will be such vectors. The format is: [delta_X, delta_Y, delta_Z, delta_roll, delta_pitch, delta_yaw, delta_gripper_state]
Where:
- `delta_X, delta_Y, delta_Z`: Represent the translational change of the robot's gripper endpoint in **meters** along the X, Y, and Z axes of the **base frame**, respectively.
- `delta_roll, delta_pitch, delta_yaw`: Represent the rotational change of the robot's gripper endpoint in **degrees** around the **base frame's** X-axis (roll), then Y-axis (pitch), then Z-axis (yaw) respectively.
- `delta_gripper_state`: This 7th element of the action vector specifies the **commanded change** to be applied to the gripper's current openness state. Gripper openness is represented by a normalized value, where **1.0 means fully open** and **0.0 means fully closed**.
    - A positive `delta_gripper_state` value from the action vector commands the gripper to become more **open** (i.e., its openness value moves towards 1.0).
    - A negative `delta_gripper_state` value commands the gripper to become more **closed** (i.e., its openness value moves towards 0.0).
    - A `delta_gripper_state` value of zero commands no change to the current gripper openness.

Your objective is to select the candidate image that most accurately depicts the scene resulting *strictly* from the given action, adhering to the considerations below.
**Important Considerations for Selection:**
1.  **Accurate Execution of Action & Gripper's Geometric Pose:**
    * The resulting scene must strictly and exclusively result from the successful execution of the specified action: {action}.
    * The robot gripper must achieve the appropriate final 3D geometric pose (position and orientation) as dictated by the action.
    * The change in the robot's pose must be the minimal and most direct transformation required by the command.
2.  **Overall Scene Consistency and Handling Imperfections:**
    * Choose the image that presents the most coherent and logically consistent result of performing the action.
    * If no candidate is flawless, prefer the one that best minimizes both errors in execution and any unjustified modifications to the environment.
Based on the **Initial State (Image 1)**, the given **low-level action** {action}, and the **selection considerations above**, choose the candidate image (from Images 2-5) that best represents the resulting state.
### Options:
A) Image 2, B) Image 3, C) Image 4, D) Image 5

### CRITICAL OUTPUT FORMAT INSTRUCTIONS:
Please do step by step reasoning first, then give your final answer. For example, if you think the correct answer is 'A', your response should be this format: '<think>(replace with your reasoning here)</think><answer>A</answer>'.
**You must choose the best possible option from A, B, C, or D. If none are perfectly correct, select the most likely or closest answer.**

### Final Answer:

Figure 32: Chain-of-Thought prompt for answering low-level State Prediction questions

**Direct-Answer Prompt for High-Level Action Inference Questions**

```
You are analyzing a robotic manipulation sequence to determine the specific high-level action that caused a transition
between two observed states.
For each task, you will be provided with the following visual information:

- **Image 1 (Initial State)**: An image containing {num_views} camera viewpoint(s) that show the robot's state (e.g., arm
pose, gripper status) and objects' positions *before* performing an action.
- **Image 2 (Resulting State)**: An image containing {num_views} camera viewpoint(s) that show the robot's state (e.g.,
arm pose, gripper status) and objects' positions *after* performing the action.

### Your Task:
Carefully analyze the Initial State (Image 1) and the Resulting State (Image 2).
Determine which high-level (goal-oriented) action most plausibly caused the transition from the initial state to the
resulting state.
Focus on changes in object positions, object interactions, and overall scene context.

If none of the options are a perfect match, select the one that most closely describes the transformation.

### Options:
A) {A}
B) {B}
C) {C}
D) {D}

### CRITICAL OUTPUT FORMAT INSTRUCTIONS:
**Your response MUST be ONLY the single capital letter (A, B, C, or D) corresponding to your chosen answer.**
**ABSOLUTELY NO other text, reasoning, explanation, punctuation, or formatting should be included.**
**Generating anything other than the single correct letter (A, B, C, or D) will result in an incorrect evaluation.**
You are acting as a precision evaluation component.

### Final Answer:
(Output only the single letter A, B, C, or D below)
```

Figure 33: Direct-Answer prompt for answering high-level Action Inference questions

---

### Direct-Answer Prompt for Mid-Level Action Inference Questions

You are analyzing a robotic manipulation sequence to determine the specific mid-level action that caused a transition between two observed states.
For each task, you will be provided with the following visual information:

- **Image 1 (Initial State)**: An image containing {num_views} camera viewpoint(s) that show the robot's state (e.g., arm pose, gripper status) and objects' positions *before* performing an action.
- **Image 2 (Resulting State)**: An image containing {num_views} camera viewpoint(s) that show the robot's state (e.g., arm pose, gripper status) and objects' positions *after* performing the action.

### Reference Frame:
The coordinate system is **always defined relative to the base of the robot arm**.
This frame applies consistently, even if the robot's base itself is not fully visible in Image 1 (Initial State) or Image 2 (Resulting State).

To help visualize these axes, imagine yourself standing at the robot's base, oriented to look in its primary "forward" direction:
- **X-axis**: Represents this "forward" direction from the base (positive: forward, negative: backward).
- **Y-axis**: Extends to your **left** from this perspective (positive: left, negative: right).
- **Z-axis**: Points **upward** from the base where you are positioned (positive: up, negative: down).

### Interpreting Gripper Actions:
The state of the gripper is defined by its "proportional aperture," a value from 0.0 (fully closed) to 1.0 (fully open).
Action Type ("Open" vs. "Close"): The terms describe the direction of change between the initial state and the resulting state.
Action Value (Extent of Change): The value represents the magnitude of the change. It is calculated as the absolute difference between the starting and ending proportional apertures.
Let P1 be the proportional aperture in the initial state and P2 be the proportional aperture in the resulting state. The value is calculated with the formula:
value = absolute_value(P2 - P1)
- **Close gripper** means the gripper jaws are **more closed** in the resulting state than in the initial state.
- **Open gripper** means the gripper jaws are **more open** in the resulting state than in the initial state.

### Your Task:
Carefully analyze the Initial State (Image 1) and the Resulting State (Image 2).
Determine which of the provided mid-level action (Options A, B, C, or D) best describes the transformation that occurred from Image 1 to Image 2.

If you find that no option is a perfect match, select the one that represents the closest and most plausible transformation.

### Options:
A) {A}
B) {B}
C) {C}
D) {D}

### CRITICAL OUTPUT FORMAT INSTRUCTIONS:
**Your response MUST be ONLY the single capital letter (A, B, C, or D) corresponding to your chosen answer.**
**ABSOLUTELY NO other text, reasoning, explanation, punctuation, or formatting should be included.**
**Generating anything other than the single correct letter (A, B, C, or D) will result in an incorrect evaluation.**
You are acting as a precision evaluation component.

### Final Answer:
(Output only the single letter A, B, C, or D below)

Figure 34: Direct-Answer prompt for answering mid-level Action Inference questions

---

● ● ●    **Direct-Answer Prompt for Low-Level Action Inference Questions**

You are analyzing a robotic manipulation task involving a robot arm interacting with objects.
The primary goal of your analysis is to determine the correct final pose of the robot's gripper after a given action.
For each task, you will be provided with the following visual information:
- **Image 1 (Initial State)**: An image containing {num_views} camera viewpoint(s) that show the robot's state (e.g., arm pose,
gripper status) and objects' positions *before* performing an action.
- **Images 2, 3, 4, and 5 (Candidate Resulting States)**: Four distinct candidate images. Each shows a possible state of the
scene after the action. Each candidate image also contains {num_views} camera viewpoint(s) with the identical arrangement and
properties as Image 1.

### Reference Frame:
The coordinate system is **always defined relative to the base of the robot arm**.
This frame applies consistently, even if the robot's base itself is not fully visible in Images 1-5.
## Linear Axes
To help visualize these axes, imagine yourself standing at the robot's base, oriented to look in its primary "forward"
direction:
* **X-axis**: Represents the "forward" direction from the base (positive: forward, negative: backward).
* **Y-axis**: Extends to your left from this perspective (positive: left, negative: right).
* **Z-axis**: Points upward from the base (positive: up, negative: down).
## Rotations
The direction of rotation is defined from the perspective of looking **from the origin outwards** along the positive axis. All
positive rotations follow the **right-hand rule**.
* **Roll (around +X axis):** Positive roll is a **clockwise** rotation, causing the top of the gripper to move to the right.
* **Pitch (around +Y axis):** Positive pitch is a **clockwise** rotation, causing the front of the gripper to move down.
* **Yaw (around +Z axis):** Positive yaw is a **counter-clockwise** rotation, causing the front of the gripper to move to the
left.

### Your Task:
You are provided with:
- **Image 1 (Initial State)**.
- A low-level action: {action}.
- **Images 2, 3, 4, and 5 (Candidate Resulting States)**.

### Low-Level Action Format:
The low-level action you need to identify is a 7-element numerical vector. The options provided (A, B, C, D) will be such
vectors. The format is: [delta_X, delta_Y, delta_Z, delta_roll, delta_pitch, delta_yaw, delta_gripper_state]
Where:
- `delta_X, delta_Y, delta_Z`: Represent the translational change of the robot's gripper endpoint in **meters** along the X, Y,
and Z axes of the **base frame**, respectively.
- `delta_roll, delta_pitch, delta_yaw`: Represent the rotational change of the robot's gripper endpoint in **degrees** around
the **base frame's** X-axis (roll), then Y-axis (pitch), then Z-axis (yaw) respectively.
- `delta_gripper_state`: This 7th element of the action vector specifies the **commanded change** to be applied to the gripper's
current openness state. Gripper openness is represented by a normalized value, where **1.0 means fully open** and **0.0 means
fully closed**.
    - A positive `delta_gripper_state` value from the action vector commands the gripper to become more **open** (i.e., its
openness value moves towards 1.0).
    - A negative `delta_gripper_state` value commands the gripper to become more **closed** (i.e., its openness value moves
towards 0.0).
    - A `delta_gripper_state` value of zero commands no change to the current gripper openness.

Your objective is to select the candidate image that most accurately depicts the scene resulting *strictly* from the given
action, adhering to the considerations below.
**Important Considerations for Selection:**
1.  **Accurate Execution of Action & Gripper's Geometric Pose:**
    * The resulting scene must strictly and exclusively result from the successful execution of the specified action: {action}.
    * The robot gripper must achieve the appropriate final 3D geometric pose (position and orientation) as dictated by the
action.
    * The change in the robot's pose must be the minimal and most direct transformation required by the command.
2.  **Overall Scene Consistency and Handling Imperfections:**
    * Choose the image that presents the most coherent and logically consistent result of performing the action.
    * If no candidate is flawless, prefer the one that best minimizes both errors in execution and any unjustified modifications
to the environment.

Based on the **Initial State (Image 1)**, the given **low-level action** {action}, and the **selection considerations above**,
choose the candidate image (from Images 2-5) that best represents the resulting state.
### Options:
A) Image 2, B) Image 3, C) Image 4, D) Image 5

### CRITICAL OUTPUT FORMAT INSTRUCTIONS:
**Your response MUST be ONLY the single capital letter (A, B, C, or D) corresponding to your chosen answer.**
**ABSOLUTELY NO other text, reasoning, explanation, punctuation, or formatting should be included.**
**Generating anything other than the single correct letter (A, B, C, or D) will result in an incorrect evaluation.**
You are acting as a precision evaluation component.

### Final Answer:
(Output only the single letter A, B, C, or D below)

Figure 35: Direct-Answer prompt for answering low-level Action Inference questions

● ● ●          **Chain-of-Thought Prompt for High-Level Action Inference Questions**

```
You are analyzing a robotic manipulation sequence to determine the specific high-level action that caused a transition
between two observed states.
For each task, you will be provided with the following visual information:

- **Image 1 (Initial State)**: An image containing {num_views} camera viewpoint(s) that show the robot's state (e.g., arm
pose, gripper status) and objects' positions *before* performing an action.
- **Image 2 (Resulting State)**: An image containing {num_views} camera viewpoint(s) that show the robot's state (e.g.,
arm pose, gripper status) and objects' positions *after* performing the action.

### Your Task:
Carefully analyze the Initial State (Image 1) and the Resulting State (Image 2).
Determine which high-level (goal-oriented) action most plausibly caused the transition from the initial state to the
resulting state.
Focus on changes in object positions, object interactions, and overall scene context.

If none of the options are a perfect match, select the one that most closely describes the transformation.

### Options:
A) {A}
B) {B}
C) {C}
D) {D}

### CRITICAL OUTPUT FORMAT INSTRUCTIONS:
Please do step by step reasoning first, then give your final answer. For example, if you think the correct answer is 'A',
your response should be this format: '<think>(replace with your reasoning here)</think><answer>A</answer>'.

**You must choose the best possible option from A, B, C, or D. If none are perfectly correct, select the most likely or
closest answer.**

### Final Answer:
```

Figure 36: Chain-of-Thought prompt for answering high-level Action Inference questions

---

### Chain-of-Thought Prompt for Mid-Level Action Inference Questions

You are analyzing a robotic manipulation sequence to determine the specific mid-level action that caused a transition between two observed states.
For each task, you will be provided with the following visual information:

- **Image 1 (Initial State)**: An image containing {num_views} camera viewpoint(s) that show the robot's state (e.g., arm pose, gripper status) and objects' positions *before* performing an action.
- **Image 2 (Resulting State)**: An image containing {num_views} camera viewpoint(s) that show the robot's state (e.g., arm pose, gripper status) and objects' positions *after* performing the action.

### Reference Frame:
The coordinate system is **always defined relative to the base of the robot arm**.
This frame applies consistently, even if the robot's base itself is not fully visible in Image 1 (Initial State) or Image 2 (Resulting State).

To help visualize these axes, imagine yourself standing at the robot's base, oriented to look in its primary "forward" direction:
- **X-axis**: Represents this "forward" direction from the base (positive: forward, negative: backward).
- **Y-axis**: Extends to your **left** from this perspective (positive: left, negative: right).
- **Z-axis**: Points **upward** from the base where you are positioned (positive: up, negative: down).

### Interpreting Gripper Actions:
The state of the gripper is defined by its "proportional aperture," a value from 0.0 (fully closed) to 1.0 (fully open).
Action Type ("Open" vs. "Close"): The terms describe the direction of change between the initial state and the resulting state.
Action Value (Extent of Change): The value represents the magnitude of the change. It is calculated as the absolute difference between the starting and ending proportional apertures.
Let P1 be the proportional aperture in the initial state and P2 be the proportional aperture in the resulting state. The value is calculated with the formula:
value = absolute_value(P2 - P1)
- **Close gripper** means the gripper jaws are **more closed** in the resulting state than in the initial state.
- **Open gripper** means the gripper jaws are **more open** in the resulting state than in the initial state.

### Your Task:
Carefully analyze the Initial State (Image 1) and the Resulting State (Image 2).
Determine which of the provided mid-level action (Options A, B, C, or D) best describes the transformation that occurred from Image 1 to Image 2.

If you find that no option is a perfect match, select the one that represents the closest and most plausible transformation.

### Options:
A) {A}
B) {B}
C) {C}
D) {D}

### CRITICAL OUTPUT FORMAT INSTRUCTIONS:
Please do step by step reasoning first, then give your final answer. For example, if you think the correct answer is 'A', your response should be this format: '<think>(replace with your reasoning here)</think><answer>A</answer>'.

**You must choose the best possible option from A, B, C, or D. If none are perfectly correct, select the most likely or closest answer.**

### Final Answer:

Figure 37: Chain-of-Thought prompt for answering mid-level Action Inference questions

---

🔴🟡🟢 **Chain-of-Thought Prompt for Low-Level Action Inference Questions**

You are analyzing a robotic manipulation task involving a robot arm interacting with objects.
The primary goal of your analysis is to determine the correct final pose of the robot's gripper after a given action.
For each task, you will be provided with the following visual information:
- **Image 1 (Initial State)**: An image containing {num_views} camera viewpoint(s) that show the robot's state (e.g., arm pose, gripper status) and objects' positions *before* performing an action.
- **Images 2, 3, 4, and 5 (Candidate Resulting States)**: Four distinct candidate images. Each shows a possible state of the scene after the action. Each candidate image also contains {num_views} camera viewpoint(s) with the identical arrangement and properties as Image 1.

### Reference Frame:
The coordinate system is **always defined relative to the base of the robot arm**.
This frame applies consistently, even if the robot's base itself is not fully visible in Images 1-5.
## Linear Axes
To help visualize these axes, imagine yourself standing at the robot's base, oriented to look in its primary "forward" direction:
* **X-axis**: Represents the "forward" direction from the base (positive: forward, negative: backward).
* **Y-axis**: Extends to your left from this perspective (positive: left, negative: right).
* **Z-axis**: Points upward from the base (positive: up, negative: down).
## Rotations
The direction of rotation is defined from the perspective of looking **from the origin outwards** along the positive axis. All positive rotations follow the **right-hand rule**.
* **Roll (around +X axis):** Positive roll is a **clockwise** rotation, causing the top of the gripper to move to the right.
* **Pitch (around +Y axis):** Positive pitch is a **clockwise** rotation, causing the front of the gripper to move down.
* **Yaw (around +Z axis):** Positive yaw is a **counter-clockwise** rotation, causing the front of the gripper to move to the left.

### Your Task:
You are provided with:
- **Image 1 (Initial State)**.
- A low-level action: {action}.
- **Images 2, 3, 4, and 5 (Candidate Resulting States)**.

### Low-Level Action Format:
The low-level action you need to identify is a 7-element numerical vector. The options provided (A, B, C, D) will be such vectors. The format is: [delta_X, delta_Y, delta_Z, delta_roll, delta_pitch, delta_yaw, delta_gripper_state]
Where:
- `delta_X, delta_Y, delta_Z`: Represent the translational change of the robot's gripper endpoint in **meters** along the X, Y, and Z axes of the **base frame**, respectively.
- `delta_roll, delta_pitch, delta_yaw`: Represent the rotational change of the robot's gripper endpoint in **degrees** around the **base frame's** X-axis (roll), then Y-axis (pitch), then Z-axis (yaw) respectively.
- `delta_gripper_state`: This 7th element of the action vector specifies the **commanded change** to be applied to the gripper's current openness state. Gripper openness is represented by a normalized value, where **1.0 means fully open** and **0.0 means fully closed**.
    - A positive `delta_gripper_state` value from the action vector commands the gripper to become more **open** (i.e., its openness value moves towards 1.0).
    - A negative `delta_gripper_state` value commands the gripper to become more **closed** (i.e., its openness value moves towards 0.0).
    - A `delta_gripper_state` value of zero commands no change to the current gripper openness.

Your objective is to select the candidate image that most accurately depicts the scene resulting *strictly* from the given action, adhering to the considerations below.
**Important Considerations for Selection:**
1.  **Accurate Execution of Action & Gripper's Geometric Pose:**
    * The resulting scene must strictly and exclusively result from the successful execution of the specified action: {action}.
    * The robot gripper must achieve the appropriate final 3D geometric pose (position and orientation) as dictated by the action.
    * The change in the robot's pose must be the minimal and most direct transformation required by the command.
2.  **Overall Scene Consistency and Handling Imperfections:**
    * Choose the image that presents the most coherent and logically consistent result of performing the action.
    * If no candidate is flawless, prefer the one that best minimizes both errors in execution and any unjustified modifications to the environment.
Based on the **Initial State (Image 1)**, the given **low-level action** {action}, and the **selection considerations above**, choose the candidate image (from Images 2-5) that best represents the resulting state.

### Options:
A) Image 2, B) Image 3, C) Image 4, D) Image 5

### CRITICAL OUTPUT FORMAT INSTRUCTIONS:
Please do step by step reasoning first, then give your final answer. For example, if you think the correct answer is 'A', your response should be this format: '<think>(replace with your reasoning here)</think><answer>A</answer>'.

**You must choose the best possible option from A, B, C, or D. If none are perfectly correct, select the most likely or closest answer.**

### Final Answer:

Figure 38: Chain-of-Thought prompt for answering low-level Action Inference questions

# D    Error Case Analysis

To effectively categorize the wide range of errors we observed, we introduce a detailed taxonomy that classifies failures into two major groups: **perceptual grounding errors** and **reasoning errors**. This classification, summarized in Table 33, provides a structured framework for understanding the diverse ways in which models can fail, from misinterpreting visual inputs to making logical mistakes.

### Perceptual Grounding Errors

This category encompasses all failures where the model's understanding of the physical world, as presented through its visual inputs, is incorrect. These errors are fundamentally about a disconnect between what the model "sees" and what is actually happening in the environment. Perceptual grounding errors are further divided into two main types:

- **Negative Hallucination Errors:** These occur when the model fails to perceive an object, an event, or a property that is actually present in the visual input. The model essentially misses something that is there. This can range from failing to recognize a static object to misjudging the magnitude of a rotation or the state of a gripper.

- **Positive Hallucination Errors:** In contrast, positive hallucinations happen when the model perceives something that does not exist. The model "invents" an object, an action, or a state change. For example, it might hallucinate that an action's preconditions have been met or that a gripper has closed when it is still open.

### Reasoning Errors

Reasoning errors, on the other hand, occur when the model correctly perceives the visual input but makes a logical mistake in processing that information. These failures are not about what the model sees but rather how it interprets and acts upon that visual data. Reasoning errors are categorized as follows:

- **Spatiotemporal Reasoning Errors:** This subcategory includes failures related to understanding the dynamics of space and time. This could mean getting the sequence of events wrong (temporal) or misinterpreting the physical relationships between objects (spatial).

- **Commonsense Reasoning Errors:** These are failures where the model violates basic, intuitive principles about how the world works. This can involve making predictions that defy fundamental physics, assuming outcomes that aren't a direct result of an action, or failing to identify the most relevant aspect of a command.

By using this taxonomy, we can pinpoint the specific causes of model failures and better understand the limitations of current visual language models (VLMs) in complex physical environments.

Table 33: A Detailed Error Taxonomy with Examples

| Error Subcategory & Definition | Concrete Example |
|---|---|
| **Perceptual Grounding Errors** | |
| *Negative Hallucination Errors* | |
| **Static Perception Error:** The model fails to perceive an object or its properties that are currently present and static in the environment. | The VLM fails to recognize the black bowl in the scene, as illustrated in Figure 39 |
| **Dynamic Perception Error:** The model fails to detect or track a change, event, or motion when the gripper interacts with an object. | The VLM incorrectly recognizes the object it holds during physical interactions, as shown in Figure 8 |
| **Rotation Misquantification:** The model correctly perceives that the gripper has rotated but fails to accurately discern the direction and magnitude of rotation. | The VLM incorrectly recognizes the rotation direction around the Z-axis, as shown in Figure 42 |

    *Continued on next page*

Table 33: A Detailed Error Taxonomy with Examples – Continued

| Error Subcategory & Definition | Concrete Example |
|---|---|
| **Translation Misquantification:** The model correctly perceives the translational movement but fails to accurately sense the direction and the distance of the translation. | The VLM incorrectly recognizes the translation direction along the Z-axis, as shown in Figure 44 |
| **Gripper State Misclassification:** The model fails to accurately determine the current gripper state (e.g., open vs. closed, holding an object vs. not holding one). | The VLM fails to recognize that the gripper has grasped the black box, as shown in Figure 43 |
| *Positive Hallucination Errors* | |
| **Action Preconditions Hallucination:** The model incorrectly believes the necessary conditions for an action are met, leading it to attempt the next action. | The VLM hallucinates a key action precondition— grasping the orange—has been met, despite visual evidence of an empty gripper, as shown in Figure 40 |
| **Action Outcomes Hallucination:** The model invents a visual outcome that is not there to satisfy the desired action. | The VLM misinterpret a shadow as a "Wet Mark" to satisfy the command "Wipe the Plate", as illustrated in Figure 8 |
| **Dynamics Hallucination:** The model perceives an object or event that does not exist or fabricates a physical change that did not occur. | When a previously hidden chocolate becomes visible, the VLM hallucinates a "Placing" action rather than recognizing it was simply revealed, as shown in Figure 40 |
| **Rotation Hallucination:** The model perceives a rotation that did not happen or fabricates the rotation of a non-existent object. | The VLM generates a hallucination of rotation around the Z-axis where no such movement occurred, as illustrated in Figure 45 |
| **Translation Hallucination:** The model perceives a translation that did not happen or fabricates the movement of a non-existent object. | The VLM generates a hallucination of translation along the Y-axis where no such movement occurred, as illustrated in Figure 44 |
| **Gripper State Hallucination:** The model perceives a change in the gripper's state (e.g., opening or closing) that did not actually occur. | The VLM hallucinates that the gripper has closed to grasp the pot, but the gripper is still open, as illustrated in Figure 46 |
| **Reasoning Errors** | |
| *Spatiotemporal Reasoning Errors* | |
| **Temporal Reasoning Error:** The model fails to correctly process the order of sequential states. | The VLM believes that an apple was placed into a bowl when it was picked up, as illustrated in Figure 8 |
| **Spatial Reasoning Error:** The model makes a mistake in understanding spatial relationships, positions, or orientations. | The VLM misinterpreted a horizontal motion as "Lifting it up into the air, off the table", as illustrated in Figure 47 |
| **Reference Frame Grounding Error:** The model misinterprets or confuses different coordinate systems (e.g., robot-centric vs. world-centric frames). | The VLM describes "Move along negative Y direction" as moving to the left, but according to the pre-defined reference frame, this action should move the object to the right, as shown in Figure 48. |
| *Commonsense Reasoning Errors* | |

Table 33: A Detailed Error Taxonomy with Examples – Continued

| Error Subcategory & Definition | Concrete Example |
| --- | --- |
| **Physics Compliance Error:** The model plans or predicts an action that violates basic, intuitive principles of physics. | The VLM disregards the physical coupling that occurs when a gripper closes on a pot, as illustrated in Figure 50 |
| **Postconditions Overinference:** The model incorrectly assumes an action caused additional side effects beyond its direct outcome. | The VLM insists that a "Move" command must end with the object being released, as illustrated in Figure 8 |
| **Action Specificity Error:** The VLM correctly identifies the agent and patient of a causal event but fails to describe the most meaningful aspect of that event. | The VLM fails to identify the primary goal of moving the chip near the chocolate bar, instead suggesting the chip be placed on the table, as illustrated in Figure 8 |
| **Extraneous Action Error:** The model selects an image that depicts a state resulting from an action, but the action itself is not specified. | The VLM selects an image showing the gripper opening, even though the desired action does not specify a gripper state change, as shown in Figure 51 |

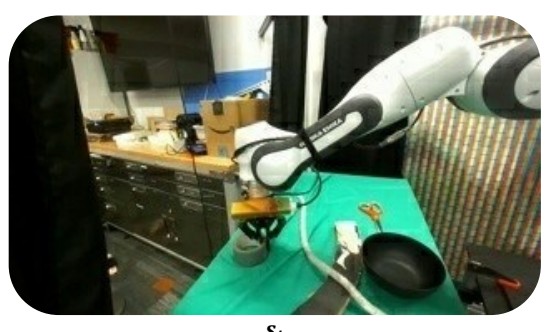 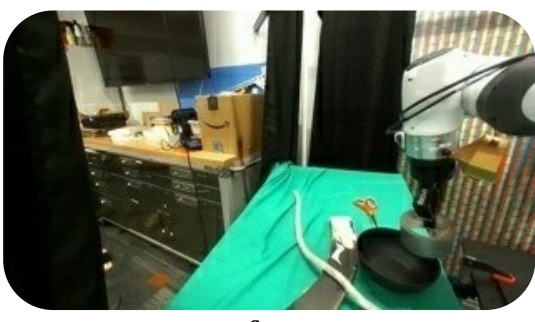

$s_t$          $s_{t+1}$

Given $s_t$ and $s_{t+1}$, what could be the action that translates $s_t$ to $s_{t+1}$?

**Gemini**

*   **Step 1: Analyze the initial state (Image 1).**
    In Image 1, a robotic arm is positioned over a work surface. The robot's gripper is holding a yellow sponge. On the table below the gripper is a roll of silver duct tape. Other objects on the table include a black pan, a pair of scissors, and other miscellaneous items. The robot appears to be about to interact with the silver tape.

*   **Step 2: Analyze the resulting state (Image 2).**
    In Image 2, the scene has changed. The robot's gripper is no longer holding the yellow sponge. Instead, it is now grasping the roll of silver duct tape. The robot arm has lifted the tape off the table. The yellow sponge is now resting on the table where the tape was previously.

*   **Step 3: Identify the action that caused the transition.**
    The most direct action that occurred between the two images is that the robot released the sponge, grasped the silver tape, and lifted it. The primary object being manipulated at the end of the sequence is the silver tape. The robot has picked it up from the table.

*   **Step 4: Evaluate the given options against the observed action.**
    The observed action is "pick up the silver tape". Let's see how the options relate to this. The options describe a complete, high-level task, and the observed action is likely a part of that task.

    *   **A) put the silver tape in the bowl:** This is a plausible high-level task. To put the tape in the bowl (or pan), the robot must first pick it up, which is what we see happening.
    *   **B) move the bowl to the other side of the table:** This is incorrect. The robot is interacting with the silver tape, not the bowl/pan.
    *   **C) place the silver tape on the table:** This is the opposite of what happened. The robot picked the tape *up from* the table.
    *   **D) put the silver tape in the pan:** This is also a plausible high-level task. The black object can be described as a small pan. To put the tape in the pan, the robot must first pick it up. This aligns with the observed action.

*   **Step 5: Select the best option.**
    Both A and D describe a plausible goal that starts with the action shown in the images (picking up the tape). The black object has a distinct handle, making "pan" a more accurate description than "bowl". Therefore, option D is the most accurate and plausible description of the high-level action that would include the observed state transition. The robot has picked up the tape as the first step towards putting it in the pan.

<answer>D</answer>

🔍 **Error Analysis**

The first error was a simple object misclassification, where the model identified an object held by the robot as a "yellowish sponge", likely due to a color tint from the camera, which highlights the model's sensitivity to image quality. The second, more significant error occurred when the model failed to recognize a pair of scissors, instead perceiving them as the handle for an adjacent black bowl. This conceptual failure caused the model to incorrectly identify the two items as a single "pan", which in turn led it to completely misinterpret the robot's overall task. Ultimately, these errors reveal that the VLM is vulnerable to both surface-level visual artifacts and deeper reasoning failures where it incorrectly groups separate objects, leading to a cascade of flawed conclusions.

**Negative Hallucination – Static Perception Error**

Figure 39: Representative Case of a Static Perception Error

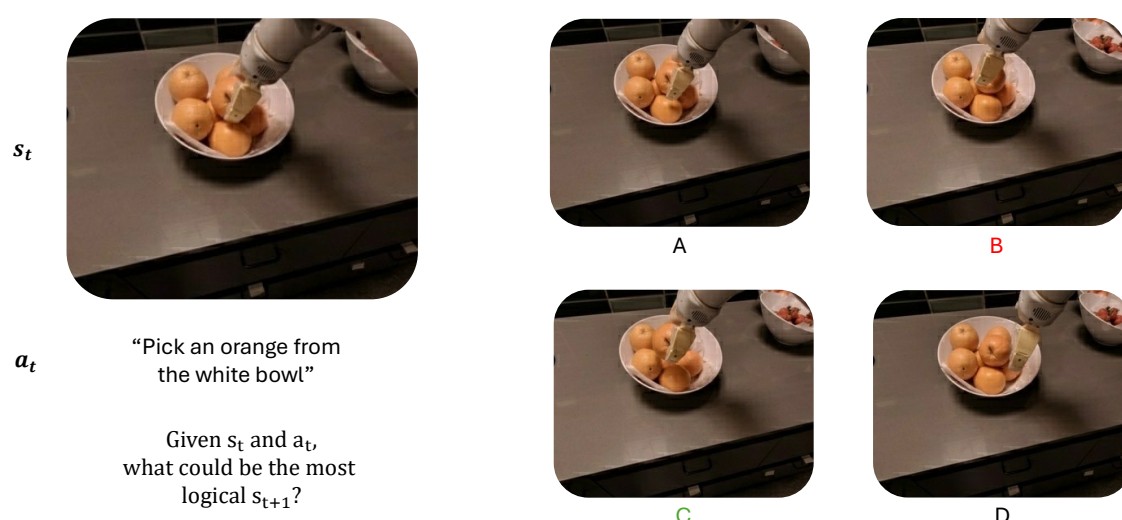

$s_t$

$a_t$

"Pick an orange from the white bowl"

Given $s_t$ and $a_t$, what could be the most logical $s_{t+1}$?

A      B

C      D

**Gemini**

* **Step 1: Analyze the initial state.** Image 1 shows a robot arm with an open gripper positioned over a white bowl containing several oranges. The robot is poised to perform an action.

* **Step 2: Understand the action.** The action is "pick orange from white bowl". A successful "pick" action involves three main sub-steps: moving to the object, grasping the object, and lifting the object from its original location.

* **Step 3: Evaluate Candidate Image 2.** In Image 2, the robot's gripper has closed around an orange. However, the orange is still inside the bowl and has not been lifted. This represents the "grasp" part of the action but not the "lift" part. It's an incomplete "pick" action.

* **Step 4: Evaluate Candidate Image 3.** In Image 3, the robot arm has moved upwards, and its gripper is closed, successfully holding an orange. The orange has been lifted clear of the bowl. This image correctly depicts the successful completion of the entire "pick orange from white bowl" action.

* **Step 5: Evaluate Candidate Image 4.** In Image 4, the robot's gripper is closed, but it is empty. It has not grasped an orange. This represents a failed attempt to pick an orange.

* **Step 6: Evaluate Candidate Image 5.** Similar to Image 2, Image 5 shows the gripper closed around an orange that is still resting in the bowl. This is an intermediate state of the pick action, not the final result.

* **Step 7: Conclude the best option.** Comparing the candidates, Image 3 is the only one that shows the successful completion of the entire "pick" sequence, including lifting the orange out of the bowl. Therefore, it is the most accurate representation of the resulting state.

<answer>B</answer>

## 🔍 Error Analysis

The target action is "pick an orange from the white bowl." A successful execution of this high-level action can be broken down into a sequence of sub-actions, each with its own preconditions and postconditions:
**move_to(orange):**
*Action*: Position the gripper over the target orange.
*Postcondition*: Gripper is aligned with the orange.
**grasp(orange):**
*Action*: Close the gripper to hold the orange.
*Precondition*: Gripper is aligned with the orange.
*Postcondition*: Gripper is closed and securely holding the orange.
**lift(orange):**
*Action*: Move the gripper vertically upwards.
*Precondition*: The state is_grasping(true) must be met.
*Postcondition*: The orange is successfully lifted from the bowl.

In the Image 3, The robot arm correctly performed the lifting motion, but because the orange was not being held, the gripper moved upwards while remaining open and empty. This demonstrates a clear violation of the action sequence's logic. The precondition for the lift action was not met, yet the action was executed anyway. The VLM selected Image 3 showing a failed attempt because it hallucinated that a key precondition—successfully grasping the orange—had been met, despite visual evidence of an empty gripper.

**Positive Hallucination – Action Precondition Hallucination**

Figure 40: Representative Case of a Action Precondition Hallucination Error

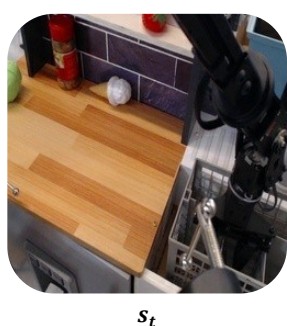 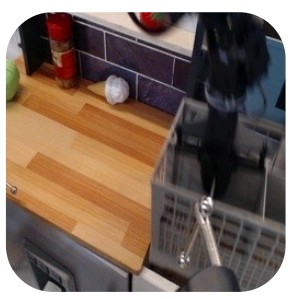

$s_t$          $s_{t+1}$

Given $s_t$ and $s_{t+1}$, what could be the action that translates $s_t$ to $s_{t+1}$?

**Gemini**

<think>
1. **Analyze Motion:** Compare the position of the robot arm and the object it's holding (the grey basket) in Image 1 and Image 2. In Image 1, the basket is higher up. In Image 2, the basket is lower down. The motion blur in Image 2 confirms a fast downward movement. In the given coordinate system, "up" is positive Z and "down" is negative Z. Therefore, the robot arm moved along the negative Z direction. This eliminates options A. Move along positive Z direction and C. Move along positive Z direction, Open gripper.

2. **Analyze Gripper State:**
   * In Image 1, the gripper is holding the basket. The jaws are open to accommodate the basket's frame.
   * In Image 2, the gripper is still holding the basket. It has not let go.
   * The action "Open gripper" is defined as the jaws becoming more open, which is typically associated with releasing an object. Since the robot is still holding the object, this is highly unlikely.
   * The action "Close gripper" is defined as the jaws becoming more closed. While we cannot visually confirm this change due to the blur and the nature of the grip (hooking rather than pinching), it is the logical alternative to opening. The robot is maintaining a "closed" grip on the object. It is not performing a release action.
   * Comparing the two remaining options, B. Move along negative Z direction, Close gripper and D. Move along negative Z direction, Open gripper, both have the correct movement direction (negative Z). Option D's "Open gripper" contradicts the fact that the robot is still holding the basket. Option B's "Close gripper" is more consistent with the action of securely holding and moving an object, even if a change in jaw distance isn't visible.

3. **Select Best Option:** The movement is clearly negative Z. The gripper action is maintaining a hold, which is functionally the opposite of opening to release. Therefore, "Close gripper" is the more plausible description of the gripper's role in this action, even if it's just maintaining a secure state. Option B is the best fit.
</think>
<answer>B</answer>

🔍 **Error Analysis**

The VLM compares the basket's position in both images. It perceives that the gripper has translational movement along the Z-axis but fails to infer the correct direction. So the core of the failure lies in the VLM's very first step: its assessment of the movement's direction. In the first state, the basket is deep inside a container. In the second state, the basket has been lifted up, closer to the top of the container. The motion is upward (positive Z). The VLM chose the wrong answer because its reasoning was based on a fundamental perception error. It incorrectly identified the direction of movement as downward instead of upward. This initial mistake had a cascading effect: The incorrect "downward" (negative Z) assessment caused it to immediately discard the correct options that described an "upward" (positive Z) motion. It was then forced to choose between two incorrect options. Its subsequent logic for choosing "Close gripper" over "Open gripper" was sound, but it was applied to a pool of already-wrong choices.

**Negative Hallucination – Translation Misquantification**

Figure 41: Representative Case of a Translation Misquantification Error

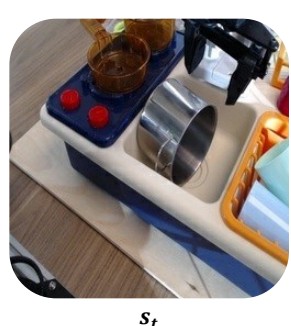 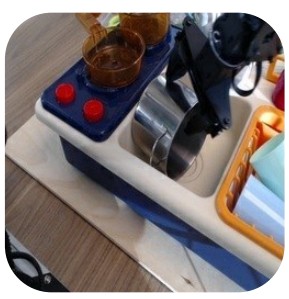

$s_t$           $s_{t+1}$

Given $s_t$ and $s_{t+1}$, what could be the action that translates $s_t$ to $s_{t+1}$?

**Gemini**

*   **Analysis of Tilt (Rotation around X-axis):**
    *   The X-axis points forward from the robot base.
    *   In Image 1, the robot gripper is relatively vertical, holding the pot in the sink.
    *   In Image 2, the gripper has tilted. The top part of the gripper has moved forward and down, while the jaws holding the pot have moved backward and up.
        *   Visualizing this from the perspective of the robot base looking forward along the X-axis, this is a **clockwise** tilt.

*   **Analysis of Rotation (Rotation around Z-axis):**
    *   The Z-axis points upward from the robot base.
    *   Observe the orientation of the gripper's main body/joint from a top-down perspective.
    *   In Image 1, the main joint of the gripper is positioned more towards the right side of the scene.
    *   In Image 2, the gripper has twisted so that this joint is now more towards the front (and slightly to the left).
    *   This change in orientation, when viewed from above, is a **counterclockwise** rotation.

*   **Conclusion:**
    *   The action consists of a clockwise tilt around the X-axis and a counterclockwise rotation around the Z-axis.
    *   This combination matches Option A. Tilt clockwise around X-axis, Rotate counterclockwise around Z-axis.

<answer>A</answer>

🔍 **Error Analysis**

The correct mid-level action should be **Tilt clockwise around X-axis, Rotate clockwise around Z-axis**. The VLM chose the wrong answer because it fundamentally misunderstood what a "rotation" command refers to in a robotic context. It correctly observed that a joint on the robot's arm moved along a counter-clockwise path, but it failed to realize that the gripper at the end of the arm simultaneously rotated in a clockwise direction.
This led to the following failure cascade: It correctly identified the first part of the action (clockwise tilt around the X-axis), but incorrectly identified the second part of the action as a counter-clockwise rotation around the Z-axis, based on the movement of the wrong part of the robot. It then selected the answer that matched its flawed analysis. The error highlights a critical gap in the VLM's inability to differentiate between the movement of an arm and the final orientation of the tool it is holding.

**Negative Hallucination – Rotation Misquantification**

Figure 42: Representative Case of a Rotation Misquantification Error

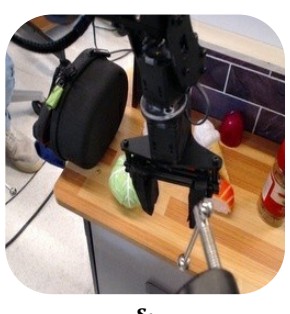
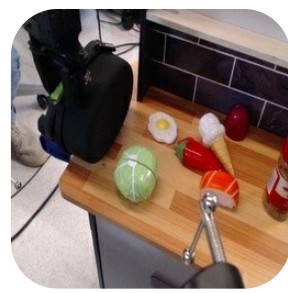

$s_t$                                          $s_{t+1}$

Given $s_t$ and $s_{t+1}$, what could be the action that translates $s_t$ to $s_{t+1}$?

**Gemini**

* **Step 1: Analyze the robot's directional movement.**
    * In Image 1, the robot's gripper is positioned directly above a green, cabbage-like toy on the wooden surface.
    * In Image 2, the robot's arm (specifically the part with the attached black bag, which is just behind the gripper) has moved significantly to the left side of the image, away from the toy cabbage.
    * According to the provided reference frame, movement to the left corresponds to the **positive Y direction**.
    * Therefore, the primary horizontal movement is along the positive Y axis. This eliminates options A. Move along negative Y direction, Close gripper and D. Move along negative Y direction, Open gripper

* **Step 2: Analyze the gripper's action.**
    * We are left with options **B. Move along positive Y direction, Close gripper** and **C. Move along positive Y direction, Open gripper**
    * In Image 1, the gripper is in a fully open state, poised to grasp the object below it.
    * In Image 2, the gripper itself is not clearly visible, as the arm has retracted. We must infer the action based on the context.
    * The robot is moving away from the object it was about to interact with. This is a retraction or disengagement action.
    * The definition of "Open gripper" is that the jaws are more open in Image 2 than in Image 1. Since the gripper was already open in Image 1, an "Open gripper" command would ensure it remains open or opens to its maximum extent. This is a very common action when disengaging from a task or resetting, to make sure nothing is accidentally grabbed or dragged.
    * The definition of "Close gripper" is that the jaws are more closed in Image 2. While a robot might close its gripper when moving to a neutral position, in the context of aborting a grasp and moving away, opening the gripper is a more logical and safer action to ensure the workspace is clear.
    * Given that the robot is moving away from the object without picking it up, the most plausible accompanying action is to ensure the gripper is open.

* **Step 3: Combine the analyses.**
    * The movement is in the positive Y direction.
    * The most logical gripper action is "Open gripper".
    * This corresponds to option C.

<answer>C</answer>

🔍 **Error Analysis**

The VLM correctly identified the robot arm's movement in the positive Y-direction and successfully eliminated two of the four answer choices. Then, it attempted to locate the gripper in the second image but failed. Because the black gripper was positioned directly in front of the black box, the lack of color contrast made it impossible for the model's vision system to segment the two objects, even though it is totally perceptible by human eyes. The model interpreted this inability to see the gripper as evidence of its absence, incorrectly concluding that the gripper had "retracted" or moved out of the scene. Based on the false premise that the robot had retracted, the VLM inferred that the robot was aborting the task. It then correctly reasoned that when aborting a grasp, the safest and most logical action is to ensure the gripper remains open to avoid accidental contact. The root cause of this error is a fundamental failure in visual segmentation. The model's logic and high-level reasoning were sound, but they were applied to a scenario that did not exist in reality. The VLM's vision component could not solve the "figure-ground problem" in the second image. The black gripper (figure) blended seamlessly with the black box (ground), causing the model to perceive only the background box. Thus, The VLM chose the wrong answer because its vision system failed to distinguish the black gripper from the black background object it was approaching. This segmentation failure led the model to incorrectly believe the gripper was not in the scene, causing it to assume the robot was retracting from the task. Based on this false assumption of a "retraction," it then logically but incorrectly concluded that "Open gripper" was the most plausible action, completely missing the robot's true intent to perform a grasp.

**Negative Hallucination – Gripper State Misclassification**

Figure 43: Representative Case of a Gripper State Classification Error

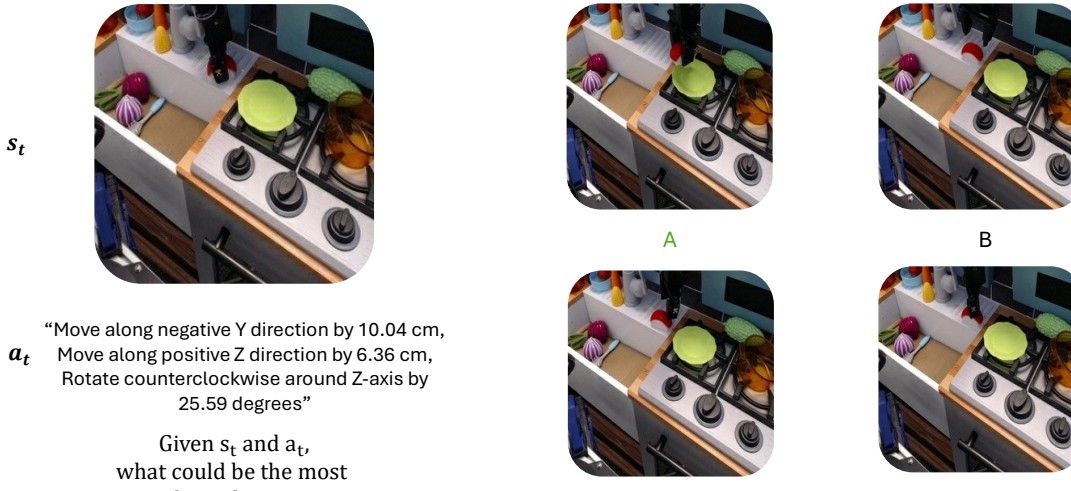

$s_t$

$a_t$

"Move along negative Y direction by 10.04 cm,
Move along positive Z direction by 6.36 cm,
Rotate counterclockwise around Z-axis by 25.59 degrees"

Given $s_t$ and $a_t$,
what could be the most
logical $s_{t+1}$?

A

B

C

D

### Gemini

* **1. Analyze the Initial State (Image 1):** The robot's gripper is holding a red and white object (resembling a toy sushi or a brush) above the white draining board area of a toy kitchen set. The gripper's "fingers" are oriented roughly parallel to the front edge of the counter.

* **2. Deconstruct the Action:** The action has three components:
    * **'Move along negative Y direction by 10.04 cm':** Based on the reference frame (positive Y is left), this means the gripper should move to the **right** by approximately 10 cm. This would shift it from its position over the draining board towards the stovetop.
    * **'Move along positive Z direction by 6.36 cm':** This means the gripper should move **upward** by approximately 6 cm.
    * **'Rotate counterclockwise around Z-axis by 25.59 degrees':** When viewed from above, the gripper should rotate counterclockwise by about 25 degrees. The red part of the object, initially pointing towards the front, should rotate to point more towards the back-left of the scene.
    * **Gripper State:** The action does not involve opening or closing the gripper, so it should remain in its initial state, continuing to hold the object.

* **3. Evaluate the Candidate Images:**
    * **Image 2:** The gripper has moved to the right and up, and it has rotated counterclockwise. However, it is no longer holding the red and white object; the object has been dropped onto the draining board. Since the action did not include opening the gripper, this image is incorrect.
    * **Image 3:** The gripper has moved slightly, but not by the specified amount. It has not rotated. Most importantly, it has dropped the object. This image is incorrect.
    * **Image 4:** The gripper has moved significantly to the right (negative Y) and upward (positive Z), consistent with the specified distances. It is now positioned over the boundary between the draining board and the stovetop. The gripper has also rotated counterclockwise by an angle that appears to be around 25 degrees. Crucially, it is still holding the red and white object. This image correctly reflects all components of the given action.
    * **Image 5:** The gripper has moved to the right and upward, similar to Image 4. However, its orientation has not changed; it has not performed the counterclockwise rotation. It is still holding the object. Since it misses the rotation component, this image is incorrect.

* **4. Conclusion:** Image 4 is the only candidate that accurately depicts the result of all three parts of the specified action: the translation in the Y and Z directions, the counterclockwise rotation, and maintaining the grip on the object.

<answer>C</answer>

### 🔍 Error Analysis

In Image 4, the gripper only has a noticeable rotation movement around the Z-axis. Compared to Image 1, it does not have a noticeable translational movement along the negative direction of Y-axis. But VLM has hallucinated this translation along Y-axis, which leads to incorrect conclusion.

**Positive Hallucination – Translation Hallucination**

Figure 44: Representative Case of a Translation Hallucination Error

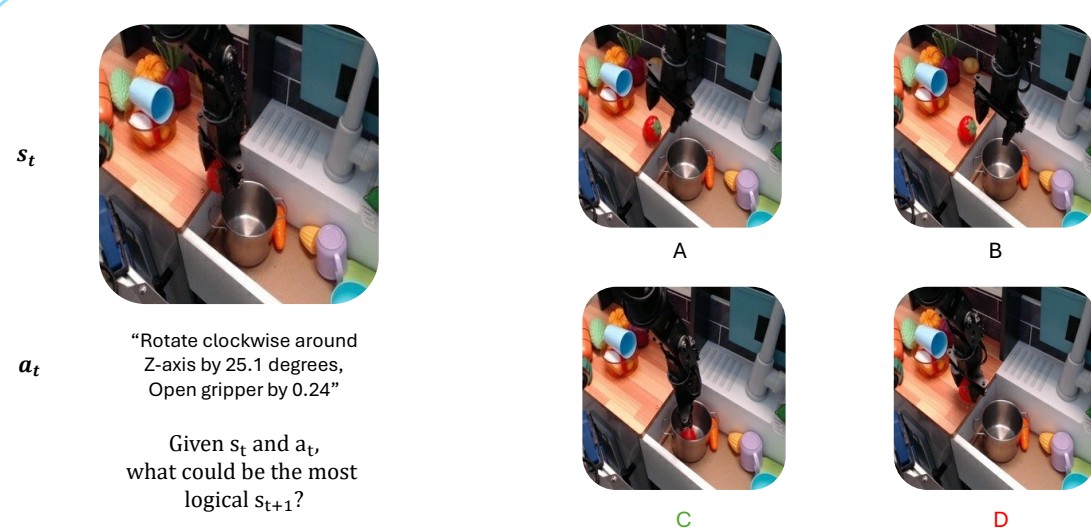

$s_t$

$a_t$

"Rotate clockwise around Z-axis by 25.1 degrees, Open gripper by 0.24"

Given $s_t$ and $a_t$, what could be the most logical $s_{t+1}$?

A

B

C

D

**Gemini**

<think>
The user wants me to select the image that shows the result of the action 'Rotate clockwise around Z-axis by 25.1 degrees, Open gripper by 0.24'.

1. **Initial State (Image 1):** Robot holds a red strawberry over a pot in a sink. Gripper is closed.
2. **Action Analysis:**
   * `Rotate clockwise around Z-axis by 25.1 degrees`: The arm's end-effector should rotate in a clockwise arc when viewed from above. From the current view, this means moving to the right.
   * `Open gripper by 0.24`: The gripper should open, releasing the strawberry.
3. **Predicted Outcome:** The arm rotates to the right, and the gripper opens, dropping the strawberry into the pot below.
4. **Candidate Evaluation:**
   * **Image 2:** Arm rotated, gripper is open. But the strawberry is on the counter. This is physically implausible. The robot would have had to place it there, not drop it. Incorrect.
   * **Image 3:** Similar to Image 2. Arm rotated, gripper is open, but the strawberry is on the counter. Incorrect.
   * **Image 4:** Arm rotated, but also moved down significantly (not in the command). The gripper is still closed (violates 'Open gripper' command). Incorrect.
   * **Image 5:** Arm has rotated clockwise. The gripper is open. The strawberry is gone. The most logical conclusion is that it was released and fell into the pot, becoming hidden from view. This matches the action command and physical reality. Correct.

Therefore, Image 5 is the best choice.
</think>
<answer>D</answer>

🔍 **Error Analysis**

In Image 5, the gripper only has a translational movement along the positive Y-axis, without any rotational movement. But VLM has hallucinated the clockwise rotation around Z-axis, which directly leads to the incorrect conclusion.

**Positive Hallucination – Rotation Hallucination**

Figure 45: Representative Case of a Rotation Hallucination Error

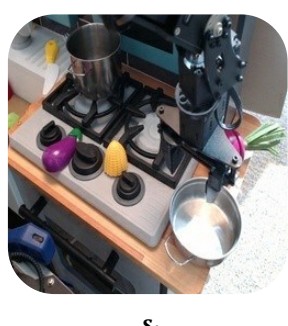 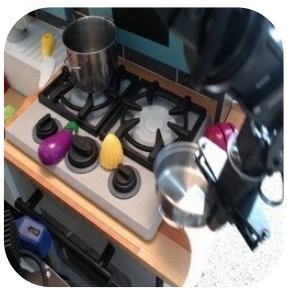

$s_t$          $s_{t+1}$

Given $s_t$ and $s_{t+1}$, what could be the action that translates $s_t$ to $s_{t+1}$?

**Gemini**

* **Step 1: Analyze the gripper action.**
  * In Image 1, the robot's gripper is holding a small silver pot by its handle. The gripper jaws are closed around the handle.
  * In Image 2, the robot's gripper is still holding the same pot in the same manner. The jaws remain closed.
  * According to the instructions, "Open gripper" means the jaws are more open in Image 2, and "Close gripper" means they are more closed. Since the robot is holding the object throughout the action, it must maintain a closed grip. Opening the gripper would cause the pot to be dropped. Therefore, the gripper action is not "Open gripper". The most logical associated action is maintaining a closed state, which aligns better with "Close gripper" than "Open gripper" in the context of the given options.

* **Step 2: Analyze the robot arm's movement.**
  * The reference frame is relative to the robot's base. The Z-axis points upwards. A rotation around the Z-axis is a horizontal swing of the arm.
  * In Image 1, the pot is held to the right of the toy stove.
  * In Image 2, the pot has been moved and is now positioned over the right burner of the stove.
  * To move the pot from the right side of the stove to a position over the stove, the robot arm had to swing towards the left (from the robot's perspective, which is on the right).
  * Imagine looking down from above (along the negative Z-axis). A swing from the right towards the left is a **counter-clockwise** rotation.

* **Step 3: Combine the analyses and evaluate the options.**
  * The action involves a **counter-clockwise rotation around the Z-axis** and maintaining a **closed gripper**.
  * Let's check the options:
    * A) Rotate clockwise around Z-axis, Close gripper: Incorrect rotation direction.
    * B) Rotate clockwise around Z-axis, Open gripper: Incorrect rotation and gripper action.
    * C) Rotate counterclockwise around Z-axis, Close gripper: This matches our analysis. The rotation is counter-clockwise, and the gripper remains closed.
    * D) Rotate counterclockwise around Z-axis, Open gripper: Incorrect gripper action.

* **Step 4: Final Conclusion.**
  * The action that transforms the scene from Image 1 to Image 2 is a counter-clockwise rotation around the Z-axis while the gripper remains closed. Option C is the only one that correctly describes this transformation.

<answer>C</answer>

🔍 **Error Analysis**

The VLM's very first step is based on a hallucination. It states, "In Image 1, the robot's gripper is holding a small silver pot by its handle. The gripper jaws are closed around the handle." In reality, the gripper is wide open and positioned above the pot in Image 1, not holding it. Based on this false premise, the VLM correctly reasons that to continue holding the pot, the gripper must remain closed, thus logically eliminating the "Open gripper" option. The VLM performs this step perfectly. It correctly observes the arm's swing from right to left and accurately identifies this as a "counter-clockwise" rotation around the Z-axis. The core of this failure is not a flaw in the VLM's logic but a catastrophic failure in its initial perception. It did not just misclassify an object; it invented an entire interaction ("holding a...pot by its handle") that is directly contradicted by the visual evidence. Thus, the VLM chose the wrong answer because it began its reasoning by hallucinating that the robot was already holding the pot with a closed gripper in the first image. This completely false starting point led it to logically but incorrectly conclude that the gripper must remain closed throughout the action. Although it correctly identified the robot's rotational movement, this correct piece of analysis was combined with the flawed conclusion about the gripper, resulting in the selection of the wrong final answer.

**Positive Hallucination – Gripper State Hallucination**

Figure 46: Representative Case of a Gripper State Hallucination Error

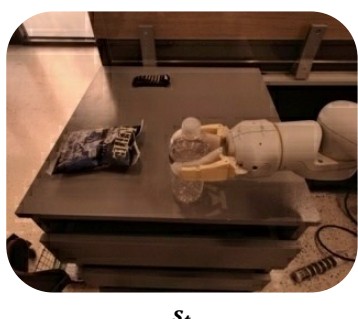 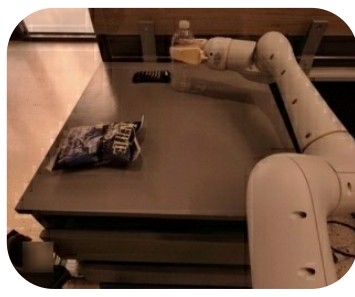

$$s_t \qquad\qquad\qquad\qquad\qquad\qquad\qquad s_{t+1}$$

Given $s_t$ and $s_{t+1}$, what could be the action that translates $s_t$ to $s_{t+1}$?

**Gemini**

<think>The user wants me to identify the high-level action that caused the transition from Image 1 to Image 2.

1. **Analyze Initial State (Image 1):** The robot gripper is holding a water bottle that is resting on a grey table. To the left is a blue snack packet. Behind the bottle is a remote control.
2. **Analyze Resulting State (Image 2):** The robot gripper is still holding the water bottle, but it has lifted it up into the air, off the table. The other objects have not moved.
3. **Identify the Transformation:** The robot has picked up the water bottle from the table. The bottle moved primarily vertically upwards.
4. **Evaluate Options:**
   * A) `place water bottle on table`: This is the opposite of what happened. The bottle was lifted *from* the table. Incorrect.
   * B) `move water bottle towards blue snack packet`: The primary motion is a lift. However, observing the horizontal position, the bottle in Image 2 is slightly to the left of its position in Image 1. Moving left is moving "towards" the blue snack packet. This is a plausible description of the horizontal component of the motion, which could indicate the overall goal.
   * C) `move water bottle near rxbar chocolate`: This is the same as option B, just with a more specific (and unconfirmed) name for the snack. It's equally plausible as B.
   * D) `put water bottle next to remote control`: The bottle starts near the remote and moves up and away from it (slightly to the left). This is the opposite of the observed horizontal movement relative to the remote. Incorrect.
5. **Conclusion:** Since the action is clearly a "pick up," and none of the options match perfectly, I must find the "best fit." Options A and D are factually incorrect regarding the direction of motion. Options B and C correctly describe the subtle horizontal component of the motion. The robot lifts the bottle and, in the process, moves it slightly towards the snack packet. This suggests the overall high-level goal is to move the bottle in that direction. Therefore, B is the most plausible choice among the given options.</think>
<answer>B</answer>

🔍 **Error Analysis**

The VLM first fails at the most fundamental level: geometry and motion. It cannot correctly distinguish the horizontal and vertical movement in the 3D space, as it misinterpreted a horizontal robot motion as "lifting it up into the air, off the table." This initial error invalidates its entire reasoning process. Since it believes the action is a "lift," it concludes that none of the provided options (which describe horizontal movement) are correct. Forced to find a "best fit" from a set of what it considers flawed options, its reasoning becomes unstable. It latches onto the horizontal component of the movement described in the options but does so under the false premise of a primary "lift." In this confused state, its object identification also falters, leading to the misidentification of the chocolate bar.

**Spatiotemporal Reasoning Error – Spatial Reasoning Error**

Figure 47: Representative Case of a Spatial Reasoning Error

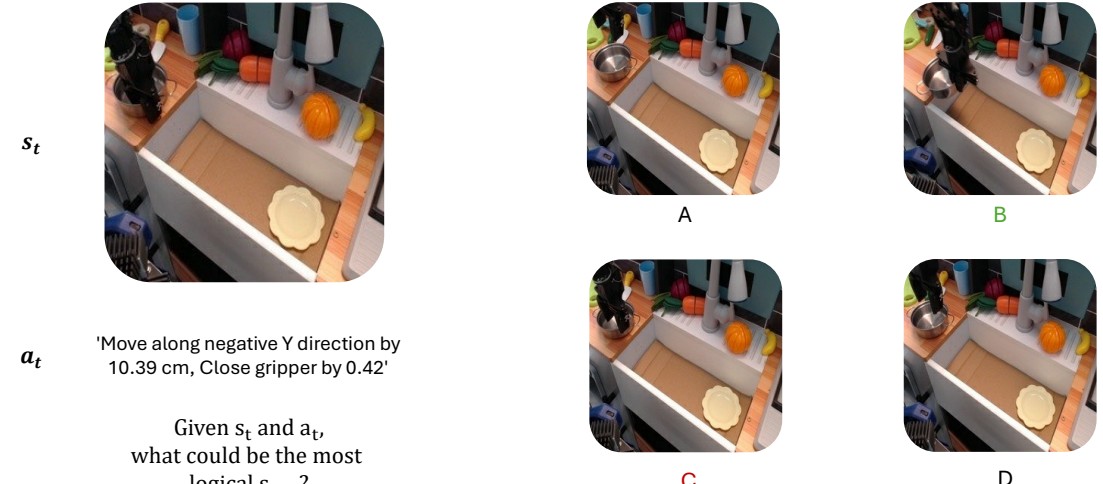

$s_t$

$a_t$    'Move along negative Y direction by 10.39 cm, Close gripper by 0.42'

Given $s_t$ and $a_t$,
what could be the most
logical $s_{t+1}$?

**Gemini**

* **Step 1: Analyze the Initial State and Reference Frame.**
    * In Image 1, the robot arm is on the left side of the scene, with its gripper open and positioned above a small metal pot.
    * The robot's base is off-camera to the left. Following the standard reference frame, the positive Y-axis points to the left from the robot's perspective. In the camera's view, this means moving from the left edge of the image towards the center/right would be a positive Y movement. Conversely, moving from the center of the image towards the left edge would be a negative Y movement.

* **Step 2: Deconstruct the Action Command.**
    * The action is 'Move along negative Y direction by 10.39 cm, Close gripper by 0.42'.
    * **Movement Component:** 'Move along negative Y direction by 10.39 cm'. Based on our reference frame analysis, the robot gripper should move further to the left within the image frame. It starts over the pot and should end up further towards the left edge of the image.
    * **Gripper Component:** 'Close gripper by 0.42'. The gripper starts in a wide-open state in Image 1. It should become significantly more closed, but not completely shut (a change of 0.42 out of a 1.0 range).

* **Step 3: Evaluate Each Candidate Image.**
    * **Image 2:** The robot arm is completely absent from the scene. This does not match the specified action of a 10.39 cm move.
    * **Image 3:** The robot arm has moved downwards and slightly to the right (in the image frame), placing the gripper inside the pot. This movement is in the negative Z and positive Y directions, which contradicts the specified negative Y movement. While the gripper has closed, the positional change is incorrect.
    * **Image 4:** The robot arm has moved to the left in the image frame, from its initial position over the pot to a new position further towards the left edge. This is consistent with a move in the negative Y direction. The gripper is also visibly more closed than in Image 1, which matches the 'Close gripper by 0.42' command. This image correctly represents both components of the action.
    * **Image 5:** Similar to Image 2, the robot arm has been removed from the scene. This is incorrect.

* **Step 4: Final Selection.**
    * Images 2 and 5 are incorrect as the robot arm is gone.
    * Image 3 is incorrect because the movement direction is wrong.
    * Image 4 is the only candidate that shows the robot arm moving in the correct direction (negative Y, or left in the image) and closing its gripper as specified.

<answer>C</answer>

🔍 **Error Analysis**

The VLM fails to ground its spatial analysis on the pre-defined reference frame. Given the reference frame, 'Move along negative Y direction by 10.39 cm' means that the gripper should move further to the right within the image frame. But VLM's analysis reveals that the robot gripper should move further to the left within the image frame, which directly leads to the wrong conclusion.

**Spatiotemporal Reasoning Error – Reference Frame Grounding Error**

Figure 48: Representative Case of a Reference Frame Grounding Error

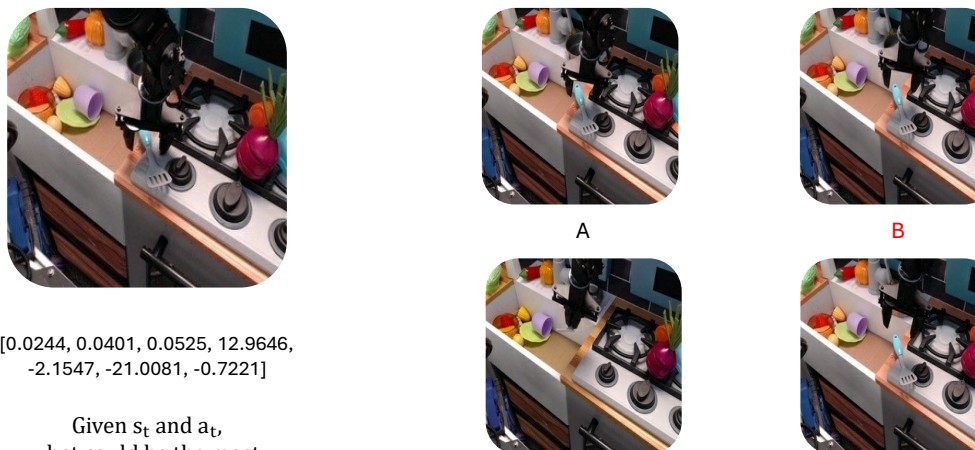

$s_t$

$a_t$

[0.0244, 0.0401, 0.0525, 12.9646, -2.1547, -21.0081, -0.7221]

Given $s_t$ and $a_t$, what could be the most logical $s_{t+1}$?

**Gemini**

*   **Initial State Analysis (Image 1):** The robot gripper is holding a light blue toy spatula. The gripper is positioned above the countertop, between the toy sink on the left and the toy stovetop on the right. The spatula is held in a mostly vertical orientation. The gripper is partially closed.

*   **Action Vector Analysis:** The action is `[0.0244, 0.0401, 0.0525, 12.9646, -2.1547, -21.0081, -0.7221]`.
    *   **Translation (`delta_X, delta_Y, delta_Z`):**
        *   `delta_X = 0.0244`: A small positive movement along the X-axis (forward).
        *   `delta_Y = 0.0401`: A small positive movement along the Y-axis (to the left).
        *   `delta_Z = 0.0525`: A significant positive movement along the Z-axis (upward).
        *   **Combined Translation:** The gripper should move up, to the left, and slightly forward. The most noticeable change should be the upward movement.

    *   **Rotation (`delta_roll, delta_pitch, delta_yaw`):**
        *   `delta_roll = 12.9646`: A positive roll of about 13 degrees around the base X-axis (counter-clockwise tilt).
        *   `delta_pitch = -2.1547`: A small negative pitch of about 2 degrees around the base Y-axis (tilting the gripper's "nose" slightly down).
        *   `delta_yaw = -21.0081`: A significant negative yaw of about 21 degrees around the base Z-axis (a clockwise rotation when viewed from above).
        *   **Combined Rotation:** The gripper should rotate clockwise (yaw), tilt slightly to its left (roll), and pitch down a tiny bit. The most prominent rotation will be the clockwise yaw.

    *   **Gripper State (`delta_gripper_state`):**
        *   `delta_gripper_state = -0.7221`: A large negative value indicates a command to close the gripper. Since the gripper is already holding the spatula, this means it should tighten its grip, not release the object.

*   **Candidate Image Evaluation:**
    *   **Image 2:** The robot arm has moved, but the spatula has been released and is now lying on the stovetop. This contradicts the gripper command `(-0.7221)`, which is to close, not open. Therefore, Image 2 is incorrect.
    *   **Image 3:** The robot arm has moved upwards, to the left, and slightly forward, consistent with the translation components. The gripper has rotated clockwise (yaw) and also appears to have rolled, which aligns with the rotation commands. Crucially, the gripper is still holding the spatula, which is consistent with the negative `delta_gripper_state`. This image accurately reflects all components of the action.
    *   **Image 4:** The gripper is empty, meaning the spatula has been released. This contradicts the gripper command. Furthermore, the final position seems much higher and further back than specified by the action's small forward `delta_X`. Therefore, Image 4 is incorrect.
    *   **Image 5:** The robot arm has barely moved from its initial position. The spatula has been released onto the stovetop. This contradicts both the significant movement specified by the action vector and the "close" command for the gripper. Therefore, Image 5 is incorrect.

*   **Conclusion:** Image 3 is the only candidate that shows the robot arm executing the specified translation and rotation while also correctly following the gripper command to maintain its hold on the object.

<answer>B</answer>

### 🔍 Error Analysis

Even though we have stressed the use of "from the origin outwards" approach to define the rotation directions, VLM still fails to transform its perspective from the traditional "looking toward the origin" approach. Given the viewpoint we defined in the prompt, a positive roll around the base X-axis corresponds to a clockwise tilt and a negative pitch around the base Y-axis corresponds to tilt the gripper's nose up.

**Spatiotemporal Reasoning Error – Perspective Taking Error**

Figure 49: Representative Case of a Perspective Taking Error

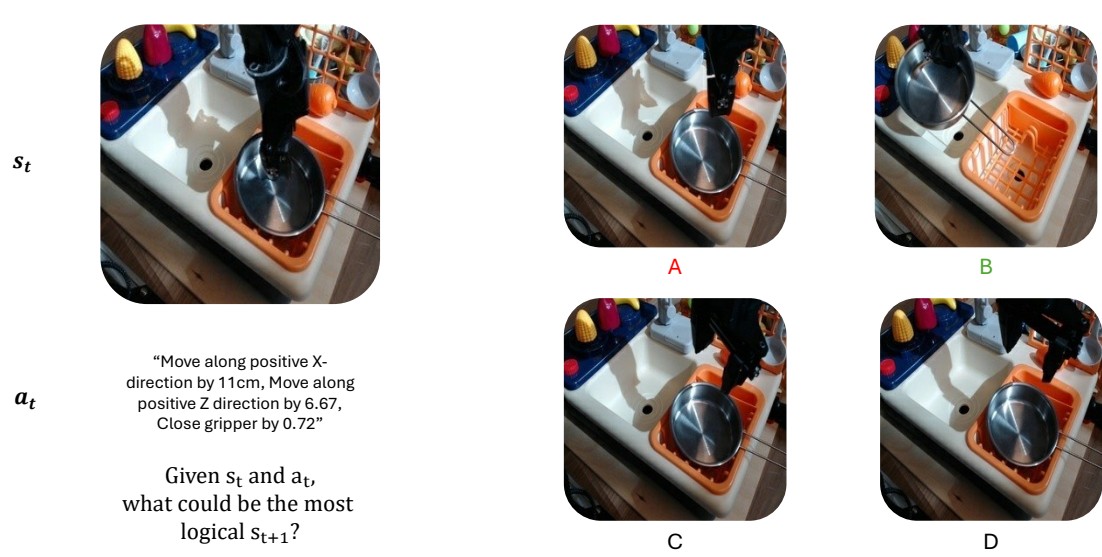

$s_t$

$a_t$

"Move along positive X-direction by 11cm, Move along positive Z direction by 6.67, Close gripper by 0.72"

Given $s_t$ and $a_t$, what could be the most logical $s_{t+1}$?

A    B    C    D

### Gemini

1. **Analyze the initial state (Image 1):** The robot gripper is open, holding a black object over a silver pan in an orange dish rack. It seems to be dropping something into the pan. The scene is a toy kitchen sink area.
2. **Analyze the action:** `Move along positive Y direction by 11.1 cm, Move along positive Z direction by 6.67 cm, Close gripper by 0.72`.
   * **Reference Frame:** Based on the typical setup, the robot base is likely to the right of the image. So, +Y is to the left (towards the sink), and +Z is up.
   * **Movement:** The gripper should move significantly to the left and up.
   * **Gripper:** The gripper should close. In Image 1, it's open. In the final image, it should be closed.
3. **Evaluate the candidates:**
   * **Image 2:** The gripper has moved to the left (from over the pan to over the divider) and has moved up. The gripper is now closed. The pan and other objects are undisturbed. This matches all components of the action.
   * **Image 3:** The gripper has moved left and up, but it has also moved the pan into the sink. The action did not specify moving the pan. This is an incorrect outcome.
   * **Image 4:** The gripper has closed and moved up slightly, but it has not moved to the left. It is still over the dish rack. This fails the "+Y movement" instruction.
   * **Image 5:** Very similar to Image 4. The gripper has closed and moved up slightly, but there is no significant movement to the left. This also fails the "+Y movement" instruction.
4. **Conclusion:** Image 2 is the only one that correctly shows the gripper moving left (+Y), moving up (+Z), and closing, without affecting other objects in the scene. The other images either fail to perform the specified movement or introduce uncommanded changes to the scene. Therefore, Image 2 is the correct answer.
<answer>A</answer>

### 🔍 Error Analysis

This example demonstrates a critical failure where a Vision-Language Model (VLM) made a prediction that violates basic laws of physics. The VLM explicitly rejected the correct outcome (Image 3), stating that "the action did not specify moving the pan." This conclusion is non-compliant with the laws of physics. The model failed to infer that the Close gripper action, when performed on an object, creates a state of physical coupling. It did not apply the principle that a force applied to the gripper would necessarily be transferred to the object it is firmly holding. The VLM treated the pan as a static background element, completely unaffected by the direct and forceful interaction of the gripper, which is a physically impossible scenario. This error reveals the model's inability to understand the physical principle of force transfer—that closing a gripper on an object couples the two together.

**Commonsense Reasoning Error – Physics Compliance Error**

Figure 50: Representative Case of a Physics Compliance Error

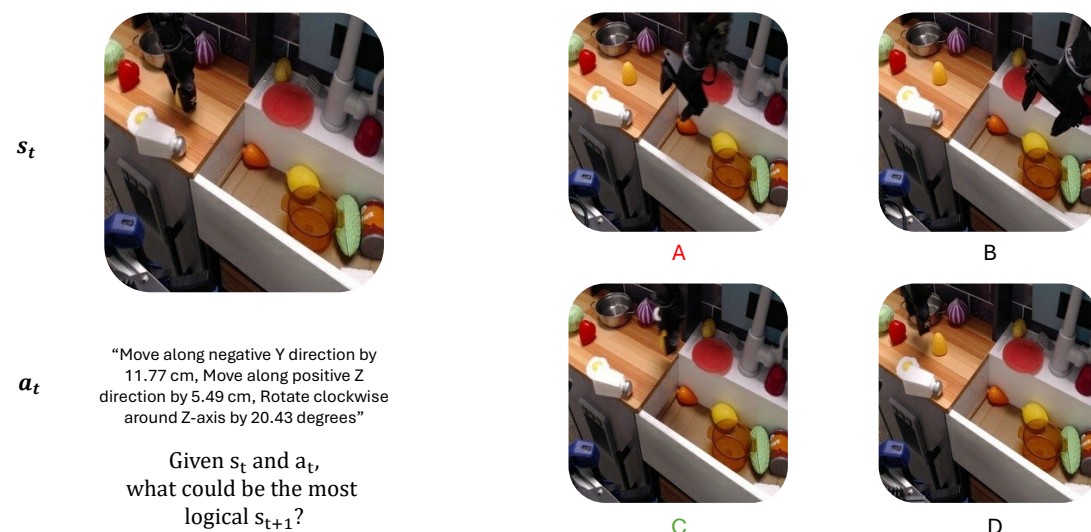

$s_t$

$a_t$

"Move along negative Y direction by 11.77 cm, Move along positive Z direction by 5.49 cm, Rotate clockwise around Z-axis by 20.43 degrees"

Given $s_t$ and $a_t$, what could be the most logical $s_{t+1}$?

### Gemini

* **Initial State Analysis (Image 1):** The robot's gripper is holding a yellow toy corn. It is positioned above the wooden countertop part of a toy kitchen set. The base of the robot appears to be to the lower-left of the frame.

* **Reference Frame Interpretation:**
    * **X-axis (forward/backward):** Points from the robot base (lower-left) into the scene (towards the upper-right).
    * **Y-axis (left/right):** Points from the right side of the image to the left. Positive Y is to the left, and negative Y is to the right.
    * **Z-axis (up/down):** Points upwards, perpendicular to the floor. Positive Z is up.

* **Action Decomposition:** The action is 'Move along negative Y direction by 11.77 cm, Move along positive Z direction by 5.49 cm, Rotate clockwise around Z-axis by 20.43 degrees'.
    1. **Move along negative Y direction by 11.77 cm:** This means the gripper should move to the **right** by approximately 11.77 cm. This should move it from its position over the wooden counter to a position over the white sink area.
    2. **Move along positive Z direction by 5.49 cm:** This means the gripper should move **upwards** by approximately 5.49 cm, increasing its distance from the countertop/sink.
    3. **Rotate clockwise around Z-axis by 20.43 degrees:** Looking down from above, the gripper should rotate clockwise. In Image 1, the gripper's jaws are roughly parallel to the front edge of the counter. A clockwise rotation would turn the front part of the gripper towards the backsplash of the kitchen set.

* **Candidate Image Evaluation:**
    * **Image 2:** The gripper has moved significantly to the right (negative Y), from over the counter to over the sink. It has also moved up (positive Z). The orientation has changed; the gripper has rotated clockwise, with its front jaw now angled towards the back. All three components of the action appear to be correctly executed.
    * **Image 3:** The gripper has moved up (positive Z) but has not moved to the right (negative Y). It is still in the same horizontal position as in Image 1. The rotation is also missing. This is incorrect.
    * **Image 4:** The gripper has moved to the right (negative Y) and up (positive Z), similar to Image 2. However, the rotation is **counter-clockwise**. The front jaw is angled away from the backsplash, towards the viewer. This is the opposite of the commanded rotation. This is incorrect.
    * **Image 5:** The gripper has moved up (positive Z) and slightly to the right (negative Y), but the rightward movement is minimal. It is still mostly over the wooden counter, not over the sink as a ~12 cm move would suggest. The translation in the Y-direction is insufficient. This is incorrect.

* **Conclusion:** Image 2 is the only candidate that accurately reflects all parts of the specified action: the rightward translation, the upward translation, and the clockwise rotation.

<answer>A</answer>

### 🔍 Error Analysis

Even though we have emphasized in the prompt that the resulting scene must strictly and exclusively result from the successful execution of the specified action, VLM still chooses Image 2 (its gripper is open) even though the mid-level action does not specify desired gripper state changes (which means the gripper state should remained unchanged relative to Image 1).

**Commonsense Reasoning Error – Extraneous Action Error**

Figure 51: Representative Case of an Extraneous Action Error

# E   Limitations and Future Directions

## E.1   Benchmark Design and Scope

**EQA vs. Direct Policy Execution.**   ACTIONEQA uses an Embodied Question Answering (EQA) format to evaluate a VLM's understanding of action cause and effect. While this choice allows for granular, diagnostic analysis of internal reasoning failures, it does not directly test the model's ability to *execute* a policy in a closed-loop setting. Future work could leverage the ACTIONEQA structure to train hierarchical policy models, using our Mid-Level Actions $a_t^{\mathrm{mid}}$ as symbolic sub-goals to bridge the gap between high-level language and low-level motor commands.

**Constrained Action Space Granularity.**   The Mid-Level Action tier, while essential for dissecting the semantic-to-physical gap, relies on a predefined set of 14 customized action labels detailed in Table 6. This is an abstraction necessary for structured evaluation and is not as rich as the infinite, continuous space of human language. Expanding the mid-level action taxonomy through generative modeling or finer-grained human annotation would further enhance the diagnostic power of this layer.

**Approximation in Low-Level Action Representation.**   Our definition of Low-Level Actions $a_t^{\mathrm{low}}$ as the direct element-wise subtraction of 7-DoF end-effector states $[\Delta x, \Delta y, \Delta z, \Delta \mathrm{roll}, \Delta \mathrm{pitch}, \Delta \mathrm{yaw}, \Delta g]$ aligns with common practice in recent vision-language-action model literature (Brohan et al., 2022; 2023; O'Neill et al., 2024; Belkhale et al., 2024; Kim et al., 2024). This approach provides a simple, fixed-size regression target suitable for end-to-end learning. However, we acknowledge that this represents a first-order approximation of the true rigid body transformation. The direct subtraction of Euler angles (roll, pitch, yaw) is mathematically ill-defined for representing changes in orientation on the non-Euclidean SO(3) manifold, which can introduce ambiguities from phenomena like gimbal lock. While this simplification directly tests a VLM's ability to reason about the action representations prevalent in the field, a compelling future direction is to extend ACTIONEQA with a more physically-grounded low-level action space, such as predicting quaternions for rotation, to investigate if more robust representations improve geometric reasoning.

**Dataset Scope and Mobile Base Actions.**   ACTIONEQA is grounded in three large-scale, real-world manipulation datasets. Although DROID, BridgeData V2, and RT-1 offer substantial diversity, our low-level tasks primarily focus on *arm manipulation* (7-DoF end-effector motion). Specifically, to maintain the integrity of our fine-grained analysis, we purposefully excluded the confounding variables of mobile base movements for the mid-level and low-level tasks in RT-1. Future iterations could extend the low-level hierarchy to explicitly model and evaluate the geometric and physical reasoning for simultaneous arm and mobile base actions, enabling a more holistic assessment of mobile manipulation VLMs.

## E.2   Future Research Directions

**Expanding Granularity through Subtask Decomposition.**   While our three-tiered hierarchy successfully identifies a performance bottleneck at the mid-level, the direct transition from high-level intent to mid-level motion descriptions remains coarse. A promising direction for future iterations of ACTIONEQA is the introduction of an intermediate layer focused on **subtask decomposition**. This layer would sit between the high-level goal and the mid-level motion descriptions, focusing on the logical breakdown of a complex task into symbolic sub-goals or semantic primitives (e.g., decomposing "Clean the table" into "Reach for sponge," "Wipe surface," and "Discard debris"). Incorporating this layer would allow for a more precise diagnosis of the mid-level bottleneck: it would help decouple failures in *hierarchical planning* (the ability to logically sequence sub-tasks) from failures in *spatial grounding* (the ability to map those sub-tasks to 3D geometric trajectories). Such a four-tiered approach would further clarify whether a model's reasoning collapses during the conceptual decomposition of a goal or during the transition from symbolic intent to physical execution.

**Targeting the Mid-Level Rotational Bottleneck.**   Our most significant finding is the performance bottleneck at the mid-level, driven primarily by failures in grounding compositional language into 3D spatial rotations. This result highlights a critical need for VLMs to develop more robust internal representations of 3D geometry and coordinate systems. Future work should focus on architectural designs or training data augmentation (e.g., synthetic rotational data) that explicitly force models to learn the perspective-invariant rules of 3D rotation and translation.

**Principled Multi-View Fusion.** Our ablation studies revealed a crucial trade-off: richer visual input from multiple views often imposes a "reasoning burden" on VLMs, leading to performance degradation. This suggests that current fusion mechanisms struggle to synergistically integrate distinct perspectives and instead suffer from a "clash of perspectives." Designing novel, principled architectures for selective or attention-based multi-view fusion, where the VLM learns to prioritize the most informative viewpoint for a given action level (e.g., egocentric for low-level, exocentric for high-level), remains an open and vital challenge. ACTIONEQA offers the necessary diagnostic framework to rigorously evaluate models along this axis.

