# OpenReview forum: "ActionEQA: Action Interface for Embodied Question Answering"
_TMLR — Accepted by TMLR_

### Review · Reviewer_GoPh · 2025-11-26

**Summary Of Contributions:**

The submission introduces ActionEQA, a benchmark designed to diagnose how well modern vision–language models bridge the semantic–physical gap in embodied tasks. The benchmark is constructed from real-world robotic manipulation datasets (BRIDGE, DROID, RT-1) and evaluates models across a three-tiered action hierarchy (high-, mid-, and low-level actions), and bidirectional reasoning tasks (state prediction vs. action inference). The authors evaluate many VLMs and report several findings, including a mid-level bottleneck, stronger performance on action inference than prediction, challenges associated with multi-view inputs, and a shift in failure types from perceptual to geometric/physical reasoning along the action hierarchy.

Strengths:
1. Novel and timely diagnostic framing for embodied reasoning.
2. Real-world datasets add realism lacking in simulation-heavy benchmarks.
3. Comprehensive evaluation across many SOTA VLMs.
4. Insightful qualitative error taxonomy.

Weaknesses:
1. Mid-level action labels are unvalidated and may be noisy.
2. Distractor generation lacks controlled quality assessment.
3. Claims regarding the mid-level bottleneck and “reasoning burden” are likely overstated.
4. Missing statistical rigor (no confidence intervals, effect sizes, or significance tests).
5. Human performance measurement is insufficiently documented.
6. Dataset biases and action-distribution imbalances are not analyzed.

**Additional Comments:**

This is a promising and potentially influential benchmark. However, the manuscript requires significant additional analyses and experiments to solidify its claims and rule out dataset-induced artifacts. The core ideas are strong, and with the requested improvements, this could be a high-impact contribution to the embodied AI and VLM evaluation fields. I would recommend a **leaning accept** and expect that the author(s) could provide a more comprehensive version before making the final decision for the reviewers.

**Audience:**

Yes

**Audience Explanation:**

TMLR’s readership includes researchers in embodied AI, robotics, multimodal learning, and vision-language-action modeling, all of whom would find the work directly relevant. The benchmark targets a central open problem in these communities: how to diagnose and evaluate the mapping from semantic language instructions to physical robot actions. The paper’s findings—such as the mid-level bottleneck, the asymmetry between action inference and state prediction, and the limitations of multi-view reasoning: offering insights that are timely and valuable for advancing embodied agents. Consequently, at least some of TMLR’s audience would have a clear interest in the results.

**Broader Impact Concerns:**

1. The possibility that action inference capabilities could be misused for surveillance or reconstruction of human activities.
2. Biases inherited from the underlying real-world robotic datasets, which may reflect skewed distributions of objects, environments, or human demonstrations.
3. Safety implications of deploying VLMs in robotic systems when predictive physical reasoning remains weak, as emphasized by the benchmark results.

**Claims And Evidence:**

No

**Claims Explanation:**

While the benchmark is thoughtfully designed and the empirical results are extensive, several central claims lack sufficiently robust support. The conclusion that mid-level action reasoning is the primary bottleneck may reflect artifacts of label noise, taxonomy design, or distractor construction, rather than a fundamental limitation of VLMs. The reported decline in performance when adding more exocentric views may stem from calibration issues, timestamp mismatch, or concatenation artifacts, not cognitive overload. Key results are reported without confidence intervals, effect sizes, or statistical significance tests, making it difficult to assess the strength of the observed differences. The human evaluation protocol is also under-described, undermining the interpretation of the human–model gap. Overall, although suggestive and valuable, the evidence does not yet convincingly support the strongest claims.

**Requested Changes:**

1. Provide a human validation study of mid-level action labels, including agreement rates and confusion matrices.
2. Add confidence intervals, effect sizes, and statistical tests for all comparisons across action levels, bidirectional tasks, and view conditions.
3. Conduct a distractor calibration analysis with human ratings of plausibility and ambiguity.
4. Add a controlled multi-view ablation using aligned or synthetic calibrated views to separate viewpoint inconsistency from model reasoning burden.
5. Fully document the human performance evaluation procedure (annotator count, protocol, inter-rater reliability).
6. Revise textual claims to avoid over-interpretation, clearly distinguishing empirical observations from causal hypotheses.

---

> ### Author Response · Authors · 2025-12-20
> **Response to Reviewer GoPh**
>
> ## Response to Reviewer GoPh
> We thank Reviewer GoPh for the constructive and detailed review, and for recognizing ActionEQA as a "novel and timely diagnostic framing" with the potential to be a "high-impact contribution." We particularly appreciate the push for greater statistical rigor and data validation; addressing these points has significantly strengthened the empirical foundation of our work. We address each requested change below and carefully updated our manuscript accordingly.
> ### Requested Changes
>
> #### [R1, W1] Provide a human validation study of mid-level action labels.
> ### Response:
>
> We thank the reviewer for raising this critical point. We agree that it is critical to ensure the validity of mid-level action labels. We respectfully point the reviewer to **Appendix C.3**, where we have already included an **Inter-Annotator Agreement (IAA)** study decomposed by action tier on a representative subset of 120 QAs pairs.
> | Action Level | Raw Agreement | Cohen's Kappa ($\kappa$) |
> | :--- | :---: | :---: |
> | High-Level | 97.5% | 0.967 |
> | **Mid-Level** | **97.5%** | **0.967** |
> | Low-Level | 92.5% | 0.900 |
> | **Overall** | **95.8%** | **0.944** |
> * Human annotators achieved a **Raw Agreement of 97.5%** and a **Cohen’s Kappa ($\kappa$) of 0.967** specifically on Mid-Level tasks. which signifies "almost perfect" agreement.
> * This **high level of human consensus** empirically demonstrates that the mid-level action labels are **valid and not noisy**.
>
> #### [R2, W4] Add confidence intervals, effect sizes, and statistical tests for all comparisons across action levels, bidirectional tasks, and view conditions.
> ### Response:
>
> We thank the reviewer for pointing out the need for formal statistical validation of our core findings. To address this, we have added four new subsections to the Appendix to rigorously test our primary claims. We have also revised the manuscript accordingly in **Section 3.2, 3.3, and 3.4** to reflect statistical rigor.
> * **1. Significance of the Mid-Level Bottleneck (Appendix C.11)**
>     We tested the "V-shaped" performance curve by comparing Mid-Level performance against both High-Level and Low-Level performance using paired t-tests.
>     *   **High vs. Mid:** Difference $+18.93$%, $p < 0.001$.
>     *   **Low vs. Mid:** Difference $+8.66$%, $p < 0.001$.
>     *   **Conclusion:** The mid-level bottleneck is a statistically robust phenomenon, not a random fluctuation.
> * **2. Significance of Action Inference vs. State Prediction (Appendix C.12)**
> We verified the claim that models are better at Action Inference (AI) than State Prediction (SP) using a **one-sided paired sample t-test** across the 34 models.
>     *   **Results:** The mean difference is $+4.31$%.
>     *   **Significance:** $p < 0.001$.
>     *   **Conclusion:** The asymmetry is statistically significant and substantial.
> * **3. Significance of Rotational Errors (Appendix C.13)**
>     To confirm that rotational errors drive the mid-level bottleneck, we tested normalized error rates for Rotation vs. Translation on BridgeData V2 across 4 view configurations.
>     *   **Results:** Rotational errors are consistently higher by an average of $3.27\%$.
>     *   **Significance**: $p = 0.013$ ($<0.05$).
>     *   **Conclusion:** Rotation is a statistically distinct source of failure compared to translation.
> * **4. Significance of Egocentric View Dominance (Appendix C.14)**
>     We validated the dominance of the Egocentric view over the Exocentric view on DROID using a **one-tailed Two-Proportion Z-Test**.
>     *   **State Prediction:** Gap $+4.10$%, $Z=2.20$, $p=0.014$.
>     *   **Action Inference:** Gap $+8.44$%, $Z=4.76$, $p < 0.00001$.
>     *   **Conclusion:** The advantage of the egocentric view is statistically significant across both reasoning tasks.

---

> ### Author Response · Authors · 2025-12-20
> **Response to Reviewer GoPh**
>
> #### [R3, W2] Conduct a distractor calibration analysis with human ratings of plausibility and ambiguity.
> ### Response:
>
> We agree with the reviewer that ensuring high-quality distractors is fundamental to the benchmark's validity. To address this concern, we have conducted an additional rigorous **Distractor Calibration Analysis** detailed in a **Appendix C.4**.
> * We randomly sample 150 QA pairs, proportionally across all three action tiers and reasoning tasks. Three professional annotators evaluated these samples using two 5-point Likert scales[1]:
>     1.  **Ambiguity (1-5, 1 is the best):** Checks if any distractor could be considered valid. (1 = Strictly Unique ground truth).
>     2.  **Plausibility (1-5, 5 is the best):** Checks if distractors are realistic. (5 = Highly realistic and physically valid actions).
>
> | Action Level | Avg. Ambiguity Score ($\downarrow$) | Avg. Plausibility Score ($\uparrow$) |
> | :--- | :---: | :---: |
> | High-Level | 1.08 | 4.42 |
> | Mid-Level | 1.21 | 4.51 |
> | Low-Level | 1.15 | 4.73 |
> | **Average** | **1.15** | **4.55** |
>
> * These results provide strong evidence that the distractors are of **high quality**:
>     *   **Low Ambiguity (1.15/5.00):** This confirms that label noise is negligible and the ground truth is strictly unique.
>     *   **High Plausibility (4.55/5.00):** This confirms that distractors represent realistic physical actions that require robust reasoning to distinguish from the correct answer.
>
> [1] Sullivan, Gail M., and Anthony R. Artino Jr. "Analyzing and interpreting data from Likert-type scales." Journal of graduate medical education 5.4 (2013): 541.
>
> #### [R4, W3] Add a controlled multi-view ablation using aligned or synthetic calibrated views to separate viewpoint inconsistency from model reasoning burden.
> ### Response:
> * We thank the reviewer for this insightful suggestion. We have conducted a new ablation study using DROID's hardware-synchronized and rectified stereo pairs (see **Appendix C.5**).
> * As shown in **Table 23**, even with perfectly calibrated data that eliminates timestamp and alignment issues, we still observe significant performance degradation in top-tier reasoning models like Gemini-2.5-Pro.
> * This provides additional empirical evidence that the bottleneck is indeed the model's inefficiency in action understanding, not view inconsistency.
>
> #### [R5, W5] Fully document the human performance evaluation procedure.
>
> ### Response:
> We thank the reviewer for emphasizing the need for a rigorous human baseline to validate the benchmark's quality. We would like to clarify that a comprehensive human performance protocol including annotator details, tooling, and statistical reliability measures was already documented in **Appendix C.3** of the submission.
> * We highlight the following data points already present in the manuscript:
>     *   **Annotators:** We recruited **two professional annotators** to complete the evaluation items.
>     *   **Protocol:** We utilized a custom-built Gradio interface to ensure human annotators received the exact same visual context, instructions, and multiple-choice options as the VLMs.
>     *   **Inter-Rater Reliability (IAA):** To ensure the benchmark is free of ambiguity, we calculated agreement metrics on a representative subset of 120 items.
>         *   **Raw Agreement:** The annotators achieved a **95.8% agreement rate** (95% CI: [92.5%, 98.3%]).
>         *   **Cohen’s Kappa ($\kappa$):** We achieved a $\kappa$ of **0.944**, which is statistically interpreted as "almost perfect" agreement.
> * We acknowledge that placing this critical information in the Appendix may have caused it to be overlooked. We have moved a summary of the human evaluation protocol and the IAA statistics into **Section 3.1** to explicitly validate the quality of the dataset.

---

> ### Author Response · Authors · 2025-12-20
> **Response to Reviewer GoPh**
>
> #### [R6, W3] Revise textual claims to avoid over-interpretation, clearly distinguishing empirical observations from causal hypotheses.
> ### Response:
> * We have included an additional statistical significance test in **Appendix C.11** to support the claim of mid-level performance bottleneck
> * We have revised the textual claims in **Section 3.3** related to the reasoning burden to avoid over-interpretation, explicitly separating empirical observations from causal hypotheses.
>
>
> #### [W6] Dataset biases and action-distribution imbalances are not analyzed.
> ### Response :
>
> We agree with the reviewer that a thorough audit of dataset characteristics is essential for interpreting model performance. In response to this request, we have conducted a comprehensive statistical analysis of the benchmark’s distributions and added  **Appendix B.7** and **Appendix B.8**, with key insights summarized below.
> * **High-level actions reflect the natural distribution of robotic manipulation tasks**
>
> | Action  | Count | Percentage (%) |
> |--------:|------:|---------------:|
> | put     | 1954  | 44.98          |
> | pick    | 916   | 21.09          |
> | place   | 722   | 16.62          |
> | move    | 400   | 9.21           |
> | remove  | 144   | 3.31           |
> | fold    | 92    | 2.12           |
> | take    | 86    | 1.98           |
> | flip    | 30    | 0.69           |
> | **Total** | **4344** | **100.00** |
>
> * **Mid-Level actions maintain a generally balanced distribution across translation, rotation, and gripper action**
>
>
> | Action Type | Count | Percentage (%) |
> |------------:|------:|---------------:|
> | Translation | 1821  | 35.87          |
> | Rotation    | 1726  | 34.00          |
> | Gripper     | 1530  | 30.14          |
> | **Total**   | **5077** | **100.00** |
>
> * **Low-Level actions cover a diverse range of spatial orientations**
>
>
> | Dataset        | Magnitude Type     | Mean   | Std    | Median | Q25    | Q75    | Min    | Max     |
> |---------------:|--------------------|-------:|-------:|-------:|-------:|-------:|-------:|--------:|
> | DROID          | Translation (m)    | 0.1603 | 0.0757 | 0.1554 | 0.0983 | 0.2103 | 0.0014 | 0.4160  |
> |                | Rotation (°)       | 28.0662| 18.9103| 22.1797| 19.2778| 27.8144| 3.8514 | 126.5623|
> |                | Gripper            | 0.3715 | 0.3406 | 0.2600 | 0.0000 | 0.7004 | 0.0000 | 1.0000  |
> | BridgeData V2  | Translation (m)    | 0.1218 | 0.0502 | 0.1189 | 0.0901 | 0.1459 | 0.0074 | 0.3173  |
> |                | Rotation (°)       | 20.9946| 12.8826| 19.3699| 11.7905| 25.5753| 1.6132 | 96.7925 |
> |                | Gripper            | 0.3484 | 0.2908 | 0.3600 | 0.0010 | 0.5065 | 0.0000 | 0.9574  |
>
> * **ActionEQA is free from answer position artifacts, as evidenced by the Chi-square test: χ² = 0.66, p = 0.8826**
>
> | Answer Key | Count | Percentage (%) | Expected | Deviation |
> |-----------:|------:|---------------:|---------:|----------:|
> | A          | 2190  | 24.90          | 2198.8   | -8.8      |
> | B          | 2179  | 24.78          | 2198.8   | -19.8     |
> | C          | 2230  | 25.36          | 2198.8   | +31.2     |
> | D          | 2196  | 24.97          | 2198.8   | -2.8      |
> | **Total**  | **8795** | **100.00** |          |           |

---

### Review · Reviewer_Z5PF · 2025-12-05

**Summary Of Contributions:**

This work introduces a new benchmark for evaluating "embodied question answering" for vision-language models (VLMs). This work evaluates open-sourced as well as proprietary VLMs across two categories: (1) state prediction where the VLM is queried the resulting state by providing a state (image) and an action (language); (2) action prediction where the VLM is queried with two states (images) and queried for the action that transitions the prior state to the latter state. Each major category is further divided into three categories by granularity. The authors claim that a bottleneck affects the VLMs reasoning at the middle layer of action inference where the VLM comparatively fails to reason direction/rotation around which axis caused the resulting state change.

This work also evaluates how VLMs perform when provided with multiple views as opposed to a subset of views and find that more information does not always necessitate better performance from the VLM. In certain cases, egocentric views perform better than images from other view points.

The final contribution is that how the nature of the failure changes with the granularity of the action changes. At higher level, errors are perceptual, whereas at lower level, the errors are related to incorrect geometric reasoning.

**Audience:**

Yes

**Audience Explanation:**

Yes, I believe this work is relevant to the greater TMLR community as it evaluates state-of-the-art open-sourced and proprietary VLMs on state-prediction and action-prediction tasks.The paper highlights some key shortcomings on the prediction tasks, and also highlights certain key areas where the VLMs fails to reason.

This work also highlights how different viewpoints influence the performance of the model on a given prediction task, and goes over how certain viewpoints perform better on a task.

I believe these findings would be relevant for the interests of the broader TMLR community.

**Broader Impact Concerns:**

A broader impact statement is not present and not required.

**Claims And Evidence:**

No

**Claims Explanation:**

While this work performs a thorough evaluation of the VLMs across two categories with three types of granularity for each category, I am not certain if the bottleneck in the mid-level action inference prediction is statistically significant. There are many instances where the mid-level action inference prediction has performed better than low-level action prediction, so it is possible that the result is not statistically significant.
I believe many such interesting insights would be visible after a thorough statistical quantitative analysis.

The authors claim that the mid-level bottleneck in the action hierarchy is caused by incorrect understanding of rotational actions, however, the error rate in rotational actions was marginally greater than the error rate for translational actions. Unsure if this claim is also statistically significant. It would also be interesting to see if humans also follow a similar trend where the rotational error is higher than the translational error.

The authors perform extensive evaluation in three levels of granularity for state prediction, however, do not reveal any trends other than - 'VLMs consistently demonstrate a higher aptitude for Action Inference over State Prediction'. I believe a quantitative statistical analysis will also greatly benefit the state prediction evaluation.

Lastly, it is also possible that many of the findings are purely circumstantial, or are peculiar to the datasets used for training the VLMs. The work would greatly benefit by having additional insights on why a particular model (or group of models) failed at a particular task - by either reasoning about the datasets used to train the model, or going over the architecture used. Whether the dataset used for evaluation in this work were used for training the VLM would also greatly influence the reasoning capacity of the model. This type of analysis will only be feasible for open-sourced models, nevertheless, it will provide a valuable insight on the reasoning capacity of said models. For e.g., if the models were never trained with the mid-level granularity, it is unlikely that they would be adept at reasoning about it.

**Requested Changes:**

Statistical significant of mid-level action inference:
Requested change: Include statistical significance testing (e.g., confidence intervals, hypothesis testing, effect sizes) to help determine whether the performance drop is meaningful or purely circumstantial.

More justification for rotational action errors:
Requested change: Additional information regarding the statistical evidence that the rotational action errors are higher would strengthen the work. If not found significant, reconsider the premise on rotational action errors. Evaluation on whether humans also exhibit similar patterns would help justify the higher amount of rotational errors.

Need additional insights regarding state inference:
Requested change:  As the title suggests, several insights about the state inference are missing and would help the reader understand the errors at different granularity for state prediction.

A deep-dive on model specific failure modes:
Requested change: A deeper (qualitative as well as quantitative) analysis on the model failure modes, by investigating the training datasets or architectural choices of the open-sourced models will provide key insights to the evaluations performed in this work. For open-source models, discuss whether the evaluation datasets overlap with their training data, and if the absence of the mid-level granularity in the training dataset is the reason for the failure of those models.

---

> ### Author Response · Authors · 2025-12-20
> **Response to Reviewer Z5PF**
>
> ## Response to Reviewer Z5PF
> We sincerely thank Reviewer Z5PF for the detailed and constructive review, and for recognizing the relevance of ActionEQA to the TMLR community. We appreciate the reviewer’s emphasis on statistical rigor and the need for deeper insights into model failure modes. These suggestions have pushed us to significantly strengthen the analytical depth of our work. We address each requested change below and carefully updated our manuscript accordingly.
> ### Requested Changes
> #### [R1] Statistical Rigor of Mid-Level Action Reasoning Bottleneck.
> ### Response:
> We thank the reviewer for pointing out the need for formal statistical validation of the observed mid-level action understanding bottleneck. We have added a statistical significance test to support the claim regarding the mid-level bottleneck in **Appendix C.11**
> * We tested the "V-shaped" performance curve by comparing Mid-Level performance against both High-Level and Low-Level performance using paired t-tests.
>     *   **High vs. Mid:** Mean Difference $+18.93$%, $p < 0.001$.
>     *   **Low vs. Mid:** Mean Difference $+8.66$%, $p < 0.001$.
>     *   **Conclusion:** The mid-level bottleneck is a statistically robust phenomenon, not a random fluctuation.
>
> #### [R2] Statistical Rigor of Rotational Errors.
> ### Response:
>
> We thank the reviewer for highlighting the need for formal statistical validation of the dominance of rotation errors in mid-level QAs. In response, we have added a statistical significance test to substantiate the claim regarding the mid-level rotational bottleneck in **Appendix C.13**.
>
> * To confirm that rotational errors drive the mid-level bottleneck, we tested normalized error rates for Rotation vs. Translation on BridgeData V2 across 4 view configurations.
>     *   **Results:** Rotational errors are consistently higher by an average of $3.27\%$.
>     *   **Significance:**  $p = 0.013$ ($<0.05$).
>     *   **Mean Difference.**  $+3.27$%
>     *   **Conclusion:** Rotation is a statistically distinct source of failure compared to translation.

---

> > ### Author Response · Authors · 2025-12-20
> > **Response to Reviewer Z5PF**
> >
> > #### [R3] Need additional insights regarding state inference.
> > ### Response:
> > We thank the reviewer for this constructive suggestion. We agree that a deeper dive into the specific failure modes of State Prediction adds significant value to the diagnostic utility of the benchmark. We have conducted a more fine-grained analysis across all three levels of the action hierarchy for State Prediction in **Appendix C.6**, with key insights summarized below:
> > * High-Level: Actions requiring complex geometric transformations (e.g., flip) or resulting in ambiguous visibility states (e.g., remove, take) suffer higher error rates than standard translation tasks, as shown in **Table 24**.
> > * Mid-Level: Downward motion along the negative Z-axis, especially when interactions with tables or objects introduce occlusion and collisions, creates substantial challenges for robust action understanding, as shown in **Table 25**.
> > * Low-Level: At larger motion magnitudes, inference performance is higher for translational states than for rotational states, as shown in **Table 26 and Figure 21**.
> > #### [R4] A deep-dive on model-specific failure modes.
> > ### Response:
> > We appreciate the reviewer’s concern regarding data contamination and the influence of training distributions. We have addressed this in **Appendix F** as follows:
> > * While the raw images from DROID, BridgeData, and RT-1 are public, the tasks, questions, and action labels across all levels are entirely novel and not open-sourced yet. Thus, the bottleneck at the mid-level is not a failure to recognize the data, but a failure to reason across the semantic-physical interface in ActionEQA.
> > * All 34 VLMs were evaluated in a strict zero-shot setting. The fact that even the most advanced reasoning models like Gemini-2.5-Pro, which likely have the most extensive training exposure to these datasets, still experience a massive performance drop at the mid-level of the action hierarchy strongly suggests the bottleneck is inherent to the reasoning process, not just a lack of data familiarity.
> > * The lack of 3D inductive biases in standard VLM architectures (which primarily use 2D ViT backbones) likely contributes to the mid-level failure. Without an internal representation of 3D coordinate frames, models struggle to map mid-level compositional language-based motion description to visual transformations, regardless of how much 2D robotic data they have seen.

---

### Review · Reviewer_RYf8 · 2025-12-06

**Summary Of Contributions:**

The authors present a benchmark that evaluates the ability of VLMs to predict action outcomes and infer actions that led to given outcomes at three levels of granularity (tasks, motions, and motor commands). The hierarchical structure of the evaluation enables honing in on the level at which VLM failures occur.

The authors assess 34 VLMs and find that:
- Most failures occur at the level of understanding motion descriptions, particularly rotational commands.
- VLMs are better at inferring actions than predicting action outcomes.
- Performance tends to degrade with the number of camera views
- Performance tends to be higher on first-person tasks than on third-person tasks.

**Additional Comments:**

Most models seem to do best with 4 views, gemini 2.5 flash being an exception. However, the paper suggests that most models degrade in performance as the number of views increases. Maybe I am misreading the paper, I would appreciate clarification.

Nits:
- The error categorization plot is interesting. Using a consistent layout would improve readability, currently the circle slices swap from left to right in two places.
- Some parts read as quite LLM-generated. This is mostly fine but some words appear meaningless in their context, causing mild confusion. An example is "follows a meticulous five-stage pipeline" - it's unclear what meticulous means here.
- I think this may be tautological, or maybe I'm not understanding the nuance:
> The nature of model failures shifts with the action hierarchy. At the high level, errors are predominantly perceptual. As actions become more granular, the failure source shifts to flawed geometric and physical reasoning, which accounts for the majority of errors at the low level.

**Audience:**

Yes

**Audience Explanation:**

- fantastic & concrete details of failure modes in VLMs
- great model coverage
- in-depth analysis of several observations, including the analysis of when adding additional views fails.

**Broader Impact Concerns:**

None.

**Claims And Evidence:**

Yes

**Claims Explanation:**

Yes, aside from the mid-level performance drop claim where the evidence could be more convincing.

**Requested Changes:**

This change (or a strong explanation of why it's unnecessary) would secure my recommendation for acceptance:

Use a statistical test to measure the strength of the non-monotonic trend where mid-level tasks are most difficult. Table 1 is too large to easily read so this claim is currently hard to verify. I counted the on- and off-trend models in the first column and found 21 on-trend and 13 off-trend models which seems reasonable, but could be affected by some model families being over-represented in the table.

According to table 4 the U-shaped performance trend only holds for action inference - the decline in state prediction performance is mostly monotonic.

Relevant text:
> As evidenced in Table 1, while VLMs demonstrate strong high-level goal-oriented reasoning and regain some footing at low-level kinematic interpretation, their performance plummets when tasked with understanding mid-level semantic motion descriptions. This trend is exemplified by the top-performing model Gemini-2.5-Pro, which scores 70.4% on high-level tasks and 61.5% on low-level tasks. However, its performance drops sharply to just 43.4% on mid-level tasks

This non-critical change would strengthen the work:

Add another layers of granularity to help understand the loss in performance at the mid-level. Perhaps there could be a layer between the goal and the motion description focused on decomposing the goal into subtasks.

---

> ### Author Response · Authors · 2025-12-20
> **Response to Reviewer RYf8**
>
> ## Response to Reviewer RYf8
> We sincerely thank Reviewer RYf8 for the positive assessment of our work, particularly for describing the failure mode analysis as "fantastic & concrete" and praising the depth of our model coverage. We appreciate the reviewer’s recommendation for acceptance and the constructive feedback regarding statistical verification and clarity. In response to your thoughtful suggestions, we have addressed each requested change and refined the manuscript accordingly to ensure our claims are statistically robust and clearly communicated.
> ### Requested Changes
> #### [R1] Statistical Rigors for mid-level bottleneck and Action Inference vs. State Prediction.
> ### Response:
> We appreciate the constructive feedback regarding the statistical verification of the mid-level bottleneck and asymmetry in bidirectional reasoning tasks. To address this, we have added these 2 subsections in Appendix to rigorously test our claims. We have also revised the manuscript accordingly to ensure statistical rigor.
> * **1. Significance of the Mid-Level Bottleneck (Appendix C.11)**
>     We tested the "V-shaped" performance curve by comparing Mid-Level performance against both High-Level and Low-Level performance using paired t-tests.
>     *   **High vs. Mid:** Mean Difference $+18.93$%, $p < 0.001$.
>     *   **Low vs. Mid:** Mean Difference $+8.66$%, $p < 0.001$.
>     *   **Conclusion:** The mid-level bottleneck is a statistically robust phenomenon, not a random fluctuation.
> * **2. Significance of Action Inference vs. State Prediction (Appendix C.12)**
> We verified the claim that models are better at Action Inference (AI) than State Prediction (SP) using a **one-sided paired sample t-test** across 34 models.
>     *   **Results:** The mean difference is $+4.31$%.
>     *   **Significance:**  $p < 0.001$.
>     *   **Conclusion:** The asymmetry is statistically significant and substantial.
> #### [R2] Add another layer of granularity.
> ### Response:
> * We thank the reviewer for this insightful suggestion. We agree that a layer focused on subtask decomposition would significantly enhance the diagnostic power of the benchmark. Specifically, it would allow us to decouple failures in hierarchical planning (the logical breakdown of a goal) from failures in spatial grounding (mapping those subtasks to 3D geometry), thereby providing a clearer explanation for the mid-level bottleneck we observed.
> * While our current 3-tier hierarchy focuses on the semantic-to-physical gap, we believe it is a extremely vital direction for refining our ActionEQA. Thus, we have added a section to **Appendix E.2** to discuss how a 4th Planning/Subtask tier could further isolate if action understanding failures stem from logic/decomposition or from geometric grounding.

---

> ### Author Response · Authors · 2025-12-20
> **Response to Reviewer RYf8**
>
> ### Additional Comments
> #### [C1] Clarification on View Performance.
> ### Response:
> We appreciate the opportunity to clarify our textual claim on multi-view performance:
> * We found a divergence between Open-Weights and Proprietary models. In Table 2, models like InternVL3-14B demonstrate monotonic improvement with more views. However, our "reasoning burden" claim refers specifically to advanced reasoning models like Gemini-2.5-Flash, which show a peak at 1 view or 2 views followed by a decline as more views are added.
> * We have added a controlled multi-view ablation using aligned calibrated views of DROID as additional empirical evidence (e.g. advanced reasoning models like Gemini-2.5-Pro also showed a consistent degrading performance with additional views despite the absence of temporal and geometrical noise in the visual data) in **Appendix C.5** to show the cognitive overhead with multiple views.
> * We have revised the textual claims to more clearly emphasize the empirical observations and reduce potential confusion.
> #### [C2] Error Categorization Plot Layout.
> ### Response:
> * We have standardized the layout of **Figure 7** so the circle slices are consistent across three subfigures for easier side-by-side comparison.
> #### [C3] Phrasing Issue.
> ### Response:
> * We apologize for the overuse of meticulous and other superlative adjectives. We have conducted a thorough edit of the manuscript to ensure a more objective tone.
> #### [C4] Tautology Issue.
> ### Response:
> * We appreciate the reviewer’s perspective on this point. We recognize that this shift may appear tautological at first glance. To clarify the nuance, it is theoretically possible for a model to fail a high-level task due to a breakdown in logic or commonsense reasoning even with perfect perception (e.g., identifying all objects correctly but choosing an action that violates physical constraints). Conversely, a model could fail a low-level task due to a perceptual deficit (e.g., being unable to locate the gripper in the frame) rather than a failure to process the 7-DoF geometry.
> * Our qualitative analysis of Gemini-2.5-Pro’s CoT reasoning process revealed:
>     * At the high level, models usually reason correctly about the goal but fail because they misidentify or hallucinate objects (Perceptual Grounding Error).
>     * At the low level, models often see the scene correctly but fail to map the numerical vectors to the 3D coordinate system (Reasoning Error).
> * We have revised the text in **Section 4** to better emphasize that these findings are based on our systematic categorization of 150 error cases and to clarify the distinction between failure modes for high-level and low-level action understanding.

---

> > ### Comment · Reviewer_RYf8 · 2025-12-22
> > **Response to Authors**
> >
> > Thank you very much for your response, and for bearing with me as I get to grips with this domain.

---

### Decision · Action_Editor_KZGh · 2026-04-14

**Recommendation:** Accept with minor revision

**Additional Comments:**

Two of three reviewers initially marked the evidence as unconvincing, and the third requested additional statistical support; post-rebuttal, all converged on acceptance. The benchmark isolates the semantic-to-physical interface rather than measuring end-to-end task success, which is useful to the community. I recommend acceptance.

**Audience:**

Yes

**Audience Explanation:**

The paper sits at the intersection of embodied AI, VLM evaluation, and robotics, which are all areas of interest to the TMLR community. The finding that mid-level action reasoning is the primary bottleneck (rather than high-level planning or low-level control) could potentially be a useful diagnostic insight for anyone building or evaluating embodied agents. All three reviewers agreed on audience relevance.

**Claims And Evidence:**

Yes

**Claims Explanation:**

Post-revision, the evidence is convincing. The authors added the statistical tests that were missing in the original submission, and all ended up showing that the results were significant at reasonable thresholds. This was the main sticking point for two of the three reviewers, and all three were satisfied after rebuttal. The human agreement numbers are strong and the distractor quality checks look good. Causal claims about why the mid-level bottleneck exists are still somewhat speculative, but the authors have appropriately toned down the language. Overall, the empirical results seem to be on firm ground.

---

> ### Author Response · Authors · 2026-04-16
>
> We greatly appreciate the action editor and all the reviewers for the constructive feedback throughout the review process! We are very glad to hear that the additional statistical tests and language adjustments effectively addressed the reviewers' initial concerns. For the camera-ready version, we will ensure that all changes discussed during the rebuttal are fully incorporated, particularly the added statistical tests, the clarified presentation of results, and the toned-down causal claims. Thanks again for your valuable guidance in strengthening our work!